# All-optical strategies to minimize photobleaching in reversibly switchable fluorescent proteins

Guillem Marín-Aguilera[1,3], Francesca Pennacchietti [1,3] ✉, Andrea Volpato[1], Alessia Papalini[1], Abhilash Kulkarni[2], Niusha Bagheri[2], Guillaume Minet[1], Jerker Widengren [2] & Ilaria Testa [1] ✉

Photobleaching is a general hurdle of fluorescence-based techniques especially in high-resolution microscopy that relies on prolonged and complex illumination. Strategies to reduce photobleaching require chemical modifications of the cell medium, which often compromise physiological cellular conditions. Here, we outline an all-optical strategy to minimize photobleaching in reversibly switching fluorescent proteins (RSFPs), a class of probes used in super-resolution and protein-multiplexing imaging techniques. By identifying the photobleaching pathways, we develop imaging schemes to increase the number of on-off photoswitching cycles, either modulating the on-switching light or co-irradiating the RSFPs with light at longer wavelengths with respect to fluorescence excitation. We apply the optimized imaging scheme to achieve imaging multiplexing at high-spatiotemporal resolutions and to record longer time-lapse imaging of sub-cellular structures with both confocal microscopy and parallelized RESOLFT nanoscopy.

Reversibly switchable fluorescent proteins (RSFPs) have advanced the field of fluorescence microscopy by enabling new super-resolution imaging[1–7] and spectroscopic approaches[8,9], as well as allowing for kinetic-based imaging multiplexing[10–13]. The switching mechanism in RSFPs involves a light-induced transition between two distinct molecular species, an emissive (on-state) and a non-emissive (off-state). The nature of the photoswitching is generally associated with a cis/trans isomerization of the chromophore followed by a protonation/deprotonation[14–18]. Due to the stable off-state, it is possible to create enough sparsity to use RSFPs in single-molecule localization microscopy[5,19] and super-resolution optical fluctuation imaging (SOFI)[20,21], while the reversible nature of the switching mechanism makes them compatible with coordinate-targeted techniques, in particular, reversible saturable optical fluorescent transition[1] (RESOLFT) and non-linear structured illumination microscopy[3] (NL-SIM). The diversity in rates of the off-switching kinetics associated with different variants has been used in imaging multiplexing at the cellular and subcellular levels[11–13,22]. In fluorescence-based spectroscopic techniques, such as fluorescence anisotropy, the long lifetime of the on-off states of RSFPs enables the extension of the mass limit inherent to conventional fluorescence anisotropy and allows for measuring the rotational diffusivity of molecules and complexes with larger masses and hydrodynamic radii[8].

Regardless of the approach, the ability of RSFPs to withstand more than thousands of photoswitching cycles defines the resulting image quality. The progressive loss of fluorescence upon cycling is referred to as photoswitching fatigue and reflects the photobleaching profile of the fluorescent protein. In a broader sense, photobleaching results from the interplay between the local nano-environment of the probe and its complex photophysics upon excitation; thus, efforts to mitigate photobleaching in microscopy strived to optimize the conditions on those two fronts. Regarding the former, multiple studies have shown that refining the composition of the imaging solvent improved the photostability of the probe[23–25]. Fusing an enhancer

[1]SciLifeLab, KTH Royal Institute of Technology, Solna, Sweden. [2]Experimental Biomolecular Physics, Dept. Applied Physics, KTH Royal Institute of Technology, Stockholm, Sweden. [3]These authors contributed equally: Guillem Marín-Aguilera, Francesca Pennacchietti. ✉e-mail: francesca.pennacchietti@scilifelab.se; ilaria.testa@scilifelab.se

nanobody to proteins of the rsGreen family increased their brightness and photostability[26]. At the same time, the development of RSFPs as fluorescent markers has been guided by recursively introducing mutations that favor faster and more robust photoswitchers[27–29].

Studies in enhanced green fluorescent protein (EGFP) identify the triplet state as the gateway for photobleaching, being the leading pathway for oxidative chemistry in the chromophore[30]. Recent studies have characterized the triplet lifetime of rsEGFP2 as well as its absorption spectrum. The properties of the triplet state can be leveraged to reduce the photobleaching of fluorescent proteins[31] (FPs). Modulation of the illumination light at a low repetition rate compared to the triplet lifetime efficiently extends imaging time[32]. Furthermore, given that the long-lived triplet state of FPs has a red-shifted absorption spectrum maximum compared to the fluorescent form, co-illuminating in this spectral window can promote the depopulation of the triplet state back to the ground singlet state. The major photobleaching pathway becomes less efficient, enabling longer time-lapse imaging[31]. Although optically-induced repopulation of the emissive state from the triplet has been known for years[32,33], and even used in imaging with organic dyes[34,35], this technique has not been applied to photobleaching reduction until recently[31]. NIR co-illumination has been demonstrated to reduce photobleaching up to 3–4 times for non-switchable FPs, such as EGFP, under widefield illumination (power densities between 1–100 W/cm² and illumination time of 15–50 ms).

Here, we characterize the main pathways underlying photoswitching fatigue in RSFPs and identify strategies to minimize photobleaching. A mechanistic understanding of the photocycle enables us to maximize the fluorescence output of rsEGFP2 by modulating the on-switching dose. Furthermore, by co-illuminating the probes with orange/NIR light, we demonstrate a reduction in photoswitching fatigue across a library of green-emitting RSFPs. Based on this photophysical investigation, we propose a strategy to extend confocal and super-resolution time-lapse imaging, as well as multiplexing approaches. The optimized imaging scheme, which incorporates patterned light at different wavelengths, enables prolonged time-lapse imaging of light-sensitive organelles.

## Results

### Characterization of photoswitching fatigue in rsEGFP2

To study the fatigue in RSFPs, we focused on the negative photoswitcher rsEGFP2[36]. This RSFP is a common marker for super-resolution imaging[36–38], multiplexed bio-imaging[11,12], as well as novel fluorescence anisotropy methods[8]. Illumination at 405 nm drives the off-to-on transition (on-switching), whereas 488 nm light excites rsEGFP2 and returns it to the off state (off-switching), inherently coupling illumination time and power density (Fig. 1a). As a first step, we quantified the loss of fluorescence as a function of illumination power density at these key wavelengths (405 and 488 nm; Fig. 1b, c, Supplementary Fig. 1).

We homogeneously illuminated a region ($20 \times 20\ \mu m^2$) of a thin layer of rsEGFP2 embedded in polyacrylamide (PAA) gel with consecutive on-and-off photoswitching cycles. The emitted fluorescence signal during each off-switching phase was recorded and the photoswitching fatigue curve was built by repeating the photocycle hundreds of times (Fig. 1a). The resulting fatigue curve for rsEGFP2 is characterized by two distinct components: (i) an initial drop, defined

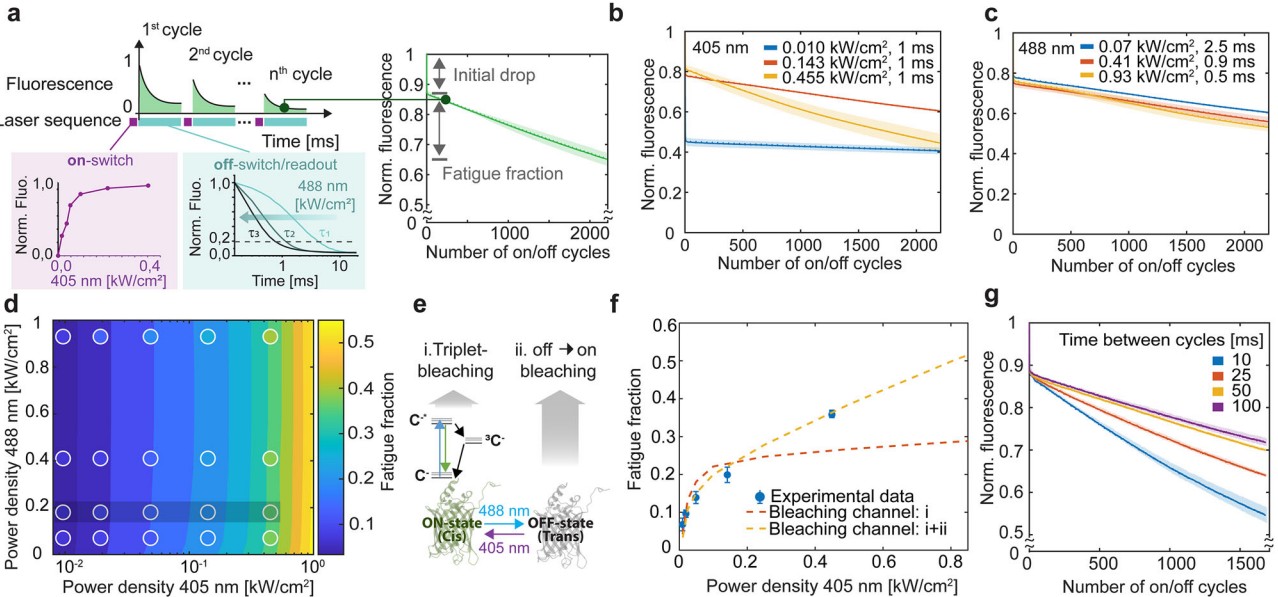

**Fig. 1 | Photoswitching fatigue characterization and kinetic modeling.**
**a** Schematic of the photoswitching fatigue experiment and the effect of 405 and 488 nm illumination on the on–off cycle. rsEGFP2 embedded in PAA gel was repeatedly illuminated with 1 ms pulses of 405 and 488 nm light to off-switch fluorescence to 20% of its initial level. The integrated fluorescence per cycle was plotted against the number of cycles to generate fatigue curves (mean ± σ, ≥3 measurements). **b** Photoswitching fatigue at increasing 405 nm doses. Fatigue curves were obtained by varying the 405 nm power density at fixed illumination time (1 ms). Each curve shows the integrated fluorescence signal per cycle (mean ± σ, ≥3 measurements). **c** Photoswitching fatigue at constant 488 nm illumination dose. The effect of varying 488 nm power density was tested while proportionally decreasing illumination time to achieve ~80% fluorescence off-switching per cycle. Integrated fluorescence per cycle is shown as mean ± σ from ≥3 measurements. **d** Dependence of fatigue on 405 and 488 nm illumination.

Experimental data points (mean of ≥3 measurements) are overlaid with the simulated fatigue fraction. The color map represents the fluorescence loss between the 4th and 2000th cycle for each illumination condition. **e** Simplified kinetic scheme for rsEGFP2. The main fatigue pathways are shown, with black arrows indicating thermally driven transitions, colored arrows light-induced transitions, and gray arrows photobleaching routes. **f** Comparison between experiment and kinetic model. Experimental data (mean ± σ, ≥3 measurements), gray region in panel (**d**) are compared with simulated trends, including only triplet-state-mediated bleaching (orange dashed line) or both triplet- and off-state-mediated fatigue (yellow dashed line). **g** Effect of thermal relaxation. Photoswitching fatigue was measured with increasing dark times between illumination cycles, showing reduced fatigue with longer relaxation intervals. Integrated fluorescence per cycle is plotted against the number of cycles (mean ± σ, ≥3 measurements).

as a sharp loss of the fluorescence signal in the first few cycles, and (ii) a steady decrease over thousands of cycles, which we refer to as the fatigue fraction. An increase in the 405 nm light dose decreased the observed initial drop, corresponding to a higher fraction of on-switched fluorophores per cycle (Fig. 1a, b). Similarly, the fatigue fraction also increased at higher 405 nm, as well as 488 nm, illumination when monitoring two orders of magnitude in power density for both illuminations ( ~ 0.01 to -1 W/cm², Supplementary Fig. 1). We chose to study the bleaching as a result of a complete photoswitching cycle, thus we tuned the off-switching illumination time to match 80% decrease of the fluorescence signal (Fig. 1a). This choice is grounded in the general use of RSFPs, where the system is typically reset at the beginning of each cycle, allowing a comparison across power density values and among different RSFPs. The role of 405 and 488 nm illumination on bleaching is fundamentally entangled and depends on the balance between the pathways they predominantly drive. The 488 nm excitation light defines the access to the triplet state and, consequently, bleaching from it. The 405 nm light contributes to photoswitching fatigue through both an indirect effect, as it increases the on-switched population and thereby promotes triplet-mediated photobleaching, and a direct effect through its own absorption, which leads to photobleaching.

The photocycle of rsEGFP2 has been extensively studied by means of spectroscopy and crystallographic methods[36,39], and the intermediate steps of the photocycle and their rates are well-reported in the literature[8,14–16]. Using this information, we built a minimal effective kinetic model for rsEGFP2 that could aid the mechanistic explanation of the observed photoswitching fatigue behavior (Supplementary Note 1). We considered the photocycle of rsEGFP2 as a 5-state kinetic model where the photoswitching reaction occurs from a light-induced isomerization followed by a protonation/de-protonation process. The thermodynamically stable *cis anionic* (C) form is the emissive form of the chromophore. From there, the on-to-off photoswitching occurs via an excited state isomerization (cis-to-trans) reaction, followed by a ground state process to the *trans neutral* (TH) form[40,41], with a characteristic time of ~50 μs at physiological pH as revealed by high-power time-resolved off-switching experiments[8]. To recover the C form of the chromophore, the off-to-on transition is efficiently triggered by violet light, typically 405 nm, from TH. Similarly, the excited state isomerization (trans-to-cis, in this case), to the *cis neutral* (CH) form, is followed by the deprotonation of the chromophore. As previously reported[14–16], the off-to-on photoswitching process can be described as a cascade of intermediates at different time scales, ranging from a few μs-to-ms. In the investigated experimental setting, we have identified an *on-like intermediate*, $C^{Int}$, with a lifetime of ~1 ms and therefore compatible with the time resolution explored in this study and generally used for advanced imaging application. The detectable fluorescence suggests it relates to an anionic form of the chromophore[41] and places it as the major gateway for bleaching pathways that stem from the on-switching process (Supplementary Notes 2, 3).

For rsEGFP2, and in general for all RSFPs, the off-switching provides a competitive channel to the triplet formation via intersystem crossing (ISC), which, combined with the spectral shift between on and off states, inherently shields the fluorophore from the adverse effects of prolonged excitation and decreases the probability of triplet-state accumulation. The triplet shielding introduces differences in the photobleaching profile of RSFPs and non-switchable fluorescent proteins. Given a constant 488 nm illumination, the former will reach a lower level of steady-state triplet population due to the competition with the off-switching process. RSFPs will converge back to the FPs photobleaching behavior in the case of constant and simultaneous 405 and 488 nm illumination, where a substantial steady-state on-population can be reached.

We incorporated the triplet-mediated photobleaching pathway in our kinetic modeling of rsEGFP2 (i in the simplified scheme of Fig. 1e)

and simulated the fatigue fraction response over the explored experimental conditions. As shown in Fig. 1f, the experimental data are well supported by the simulation for 405 nm power densities below 300 W/cm² and diverge hereafter. The experimental behavior suggests an additional photobleaching pathway stemming from the off-to-on transition (ii in Fig. 1f, e, Supplementary Note 4). The need for an additional photobleaching channel is further supported by the dependence of the fatigue on the waiting time between sequential cycles (Fig. 1g). Despite leaving time for the triplet population to relax, the fatigue can never be recovered fully, indicating that not all the irreversible fluorescence loss stems from a single mechanism (i.e., the triplet-based photobleaching). Increasing the dwell time from 10 to 100 ms resulted in ~20% photoswitching fatigue recovery after 1600 cycles (Fig. 1g). The time dependence of such relaxation exceeds the expected triplet state lifetime, both reported[42] and measured in our data, suggesting that the triplet state acts as a precursor to longer-lived photoproducts, which lead to reversible and irreversible photobleaching (Supplementary Note 5). Increasing the interval between successive photocycles prevents the accumulation of these intermediate long-lived dark states.

## Maximizing on-off cycles in rsEGFP2

The global photon budget of each RSFP is set by the combination of the initial drop of the fluorescence and the number of cycles the probe can undergo. Supported by the outlined kinetic model, we investigated strategies to maximize the detected fluorescence photons by modulating the illumination.

The initial drop in the fluorescence signal arises from the kinetics of the on-switching mechanism in rsEGFP2[43]. At low power, this drop is a consequence of the 405 nm absorption of $C$[43]. However, under the investigated power densities, both the kinetics and absorption properties of the intermediate state $C^{Int}$ account for an additional loss (Supplementary Note 3). At pH 7.5, rsEGFP2 is expected to have ~20% of the concentration of the fluorophore trapped in TH (Supplementary Notes 2, 3). The formation of the TH state can be mitigated by extending the on-switching times at a fixed total energy, which reduces $C^{Int}$ accumulation and lowers the probability of $C^{Int}$-to-TH conversion, a transition promoted by 405 nm absorption. Experimentally, a constant dose of 64 mJ/cm² was delivered by progressively increasing the illumination time (0.3–20 ms) while proportionally decreasing the 405 nm power density (200–3.3 W/cm²) as shown in Fig. 2a. Under these conditions, the fluorescence loss was reduced to ~7% when the activation dose was stretched to 20 ms. Figure. 2a, b shows the enhancement for both rsEGFP2 embedded in the PAA layer and linked to actin in eukaryotic cells. Our simulations confirm that this is compatible with preventing the quasi-equilibrium among the on-switching intermediates (Supplementary Note 6) and, therefore, a more efficient turnover between the on and off-states upon 405 nm irradiation. Altogether, we found that the initial drop observed in the photoswitching fatigue curves is not irreversible and can be efficiently recovered by modulating the way the on-switching light dose is delivered.

Reverse inter-system crossing (RISC), i.e., the backward process from the triplet state to the singlet manifold, was shown to be optically modulated and thereby effectively increasing the brightness of organic dyes[32] and preventing photobleaching in EGFP[31]. Since the protein scaffold shields the chromophore from the immediate local environment, the triplet lifetime in FPs is relatively long ( ~ 5 ms for EGFP[30]) and RISC transition can be triggered with lower light doses compared to dyes (0.1–40 MW/cm² for Eosin Y[32], compared to -kW/cm² for GFP-like proteins[31,44]). We measured the time evolution of a dark state relaxing in the μs-ms window by monitoring the increase in the integrated fluorescence signal between two 488 nm pulses separated by a variable delay. We observed a build-up in the signal of the second pulse with a raising time of ~2.5 ms (Fig. 2c, Supplementary Note 7). Since the

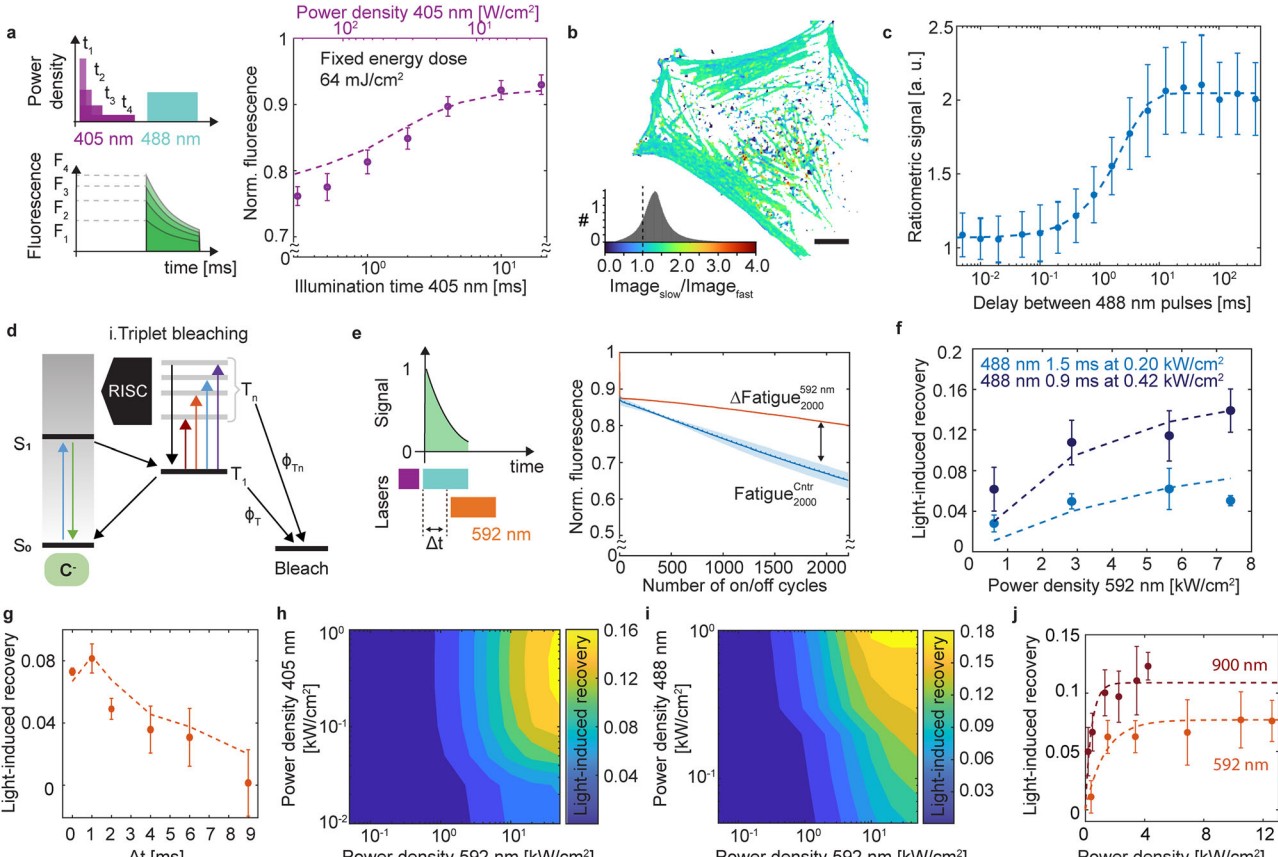

**Fig. 2 | Optical strategies to minimize photobleaching in RSFPs. a** Schematic and experimental data for the on-switching modulation at fixed energy dose of 64 mJ/cm² (mean ± σ of 4 replicas on rsEGFP2-embedded PAA gel). The simulation (dashed lines) reports on the evolution of the kinetic model over the experimental pulse schemes. **b** Ratiometric image of Chromobody-rsEGFP2 in U2OS cells for slow (405 nm for 20 ms at 0.014 kW/cm²) and fast (405 nm for 0.25 ms at 1.05 kW/cm²) on-switching acquisition. The histogram corresponds to the calculated ratiometric pixel intensity from N = 8 cells. Scale bar, 5 µm. **c** Relaxation time of the triplet state back to the emissive state ($\tau_1 = 2.46 \pm 0.32$ ms), overlapped to an exponential fit rise in dashed lines. The data is represented as mean ± σ of at least 3 measurements. **d** Schematic of the proposed triplet photobleaching pathway. **e** Photoswitching fatigue curves with and without 592 nm co-illumination. To parametrize the recovery effect of the additional wavelength, we compared the fluorescence signal after 2000 frames with 592 nm to a control curve for each experiment. Each curve shows the integrated fluorescence signal (mean ± σ, ≥3 measurements). **f, g** Fatigue recovery dependence at increasing 592 nm power density (**f**) at two different 488 nm light doses and delay between 488 and 592 nm pulses (**g**). The experimental data (mean ± σ of at least 3 measurements) are plotted together with a simulation of the kinetic model (dashed lines). **h** Simulated light-induced recovery for a range of 405 and 592 nm power densities. For this simulation, the 488 nm dose was kept constant at 1.2 ms illumination time and 200 W/cm². **i** Simulation of fatigue recovery as a function of the 488 and 592 nm power densities. The simulations were carried out at a constant 488 energy dose, power density times illumination time. **j** Light-induced recovery dependence to different wavelengths in the NIR spectral region: 915 nm vs 592 nm at increasing power density. The experimental data (mean ± σ of at least 3 measurements) are plotted together with the exponential fit rise (dashed lines, $k_{915} = 0.42$ kW/cm² and $k_{592} = 1.33$ kW/cm²).

thermal recovery for rsEGFP2 (thermal relaxation from *TH* to *C*) has a timescale of several hours[39], the increase in the concentration of the fluorescence emissive state is likely to originate from the relaxation of a dark state created upon excitation (Fig. 2d). The timescale of such relaxation is comparable to the reported triplet state lifetime in rsEGFP2 at 260 K[42].

To see if and to which extent the optically-induced RISC could help mitigate the photoswitching fatigue of RSFPs, we used the same experimental framework of the previous section with a modified pulse scheme which included 592 nm co-illumination (Fig. 2e). At 592 nm the absorption cross-section for the triplet state of rsEGFP2 was reported to be ~20% of the peak at 900 nm at 100 K[42]. We examined the effect of 592 nm co-illumination as a function of power density (~ 0.5–8 kW/cm²; Fig. 2f) and of the delay time relative to the 488 nm excitation pulse (Fig. 2g). Introducing 592 nm co-illumination into the pulse scheme led to a reduction in photoswitching fatigue (Fig. 2e), with recovery increasing by up to ~15% at higher power densities. When 592 nm co-illumination was applied, fluorescence signals were restored to similar levels regardless of the 488 nm power density

(Supplementary Note 7), indicating that only the triplet-mediated photobleaching fraction is optically recoverable. We also observed a correlation between the magnitude of recovery and the 488 nm excitation power, with light-induced recovery being more pronounced at higher 488 nm power densities (Fig. 2f). Since RSFPs are off-switched with a near-constant energy dose (power density × illumination time), a higher 488 nm power density shortens the exposure time, thereby maximizing the efficiency of 592 nm-mediated recovery. Regarding the delay between the 488 nm and 592 nm pulses (Fig. 2g), we observed that the magnitude of the recovery decays over a timescale of milliseconds, consistent with the reported lifetime of the triplet state[42].

To provide further insights into the photoswitching fatigue mechanism, we fitted our experimental data to the developed kinetic model (Supplementary Note 1), which covers the switching mechanism of rsEGFP2, the photobleaching pathways described in the previous section, as well as the spectroscopic properties of the triplet state, parametrized from the reported data in this manuscript and published studies[42]. Similarly to recent literature on light-induced RISC in EGFP[31] and rsEGFP2[45], we rationalize the repopulation of the emissive state via

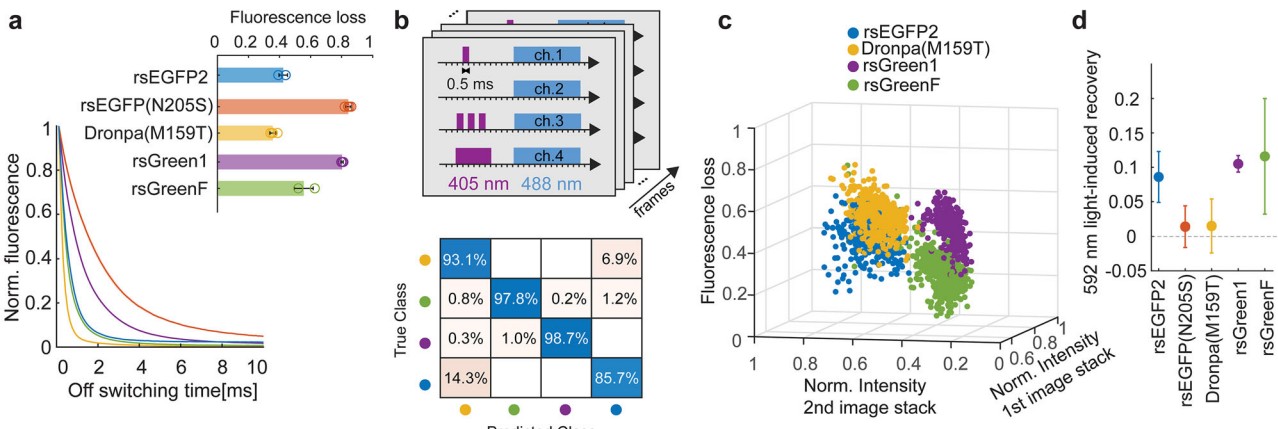

**Fig. 3 | Photoswitching fatigue for multiple green-emitting rsFPs. a** Off-switching and photobleaching behavior of green negative photoswitchers. Off-switching curves were recorded at 405 nm (160 W/cm², 1 ms) and 488 nm (200 W/cm²). Fluorescence loss after 2000 switching cycles is shown for several green negative photoswitchers. The on-switching illumination dose for all RSFPs was 405 nm, 1 ms at 160 W/cm², while the 488 nm off-switching doses were: 0.9 ms at 300 W/cm² for rsEGFP2; 3.7 ms at 200 W/cm² for rsEGFP(N205S); 0.6 ms at 220 W/cm² for Dronpa(M159T); 1.5 ms at 190 W/cm² for rsGreen1; and 1.0 ms at 240 W/cm² for rsGreenF. All data represent the mean ± σ of at least three independent measurements per sample. **b** Multiplexing scheme and unmixing analysis. The upper panel shows the 405 nm pulse sequence used for multiplexed imaging. Each pixel was illuminated with four sequential 405 nm doses. To generate unmixing frames, samples were scanned at 90 nm steps, yielding four images in which feature intensities varied with 405 nm exposure. This procedure was repeated 15 times to collect 60 frames. Every fourth frame was used to normalize the preceding three, and each four-image set was averaged to create a 3D unmixing space. Fluorescence decay due to photobleaching was added as a fourth unmixing dimension. The lower panel presents the confusion matrix showing the classification accuracy of the unmixing predictions for four RSFPs. Cluster centroids for each protein were identified, and the dataset was reanalyzed using a Gaussian mixture model containing the parameters of the isolated clusters. **c** Spatial distribution and classification. The distribution of segmented bacteria expressing individual RSFPs is shown in the 3D unmixing space. The combination of this spatial information with the normalized intensity of the third image stack enabled the identification of individual bacterial populations. **d** Light-induced recovery behavior. Recovery effects were analysed for different green negative photoswitchers when a 592 nm pulse followed the 488 nm pulse by 1 ms. The recovery was calculated as the difference in mean fluorescence loss between conditions with and without 592 nm illumination. Errors correspond to propagated standard deviations from at least three independent measurements per condition.

RISC from the triplet excited state ($T_n$) to a vibronic state of the singlet manifold (Supplementary Note 1 and 8). All the different wavelengths explored (405, 488, 592, 915 nm) will excite the triplet state[42] stimulating both photobleaching and RISC. Upon fluorescence excitation, both $T_1$ and higher triplet states contribute to photobleaching (Fig. 2c), and co-illumination with a 592 nm decreases the probability of photobleaching by reducing the triplet state population. The proposed kinetic model incorporates effective photobleaching quantum yields for ground and excited triplet states, $\Phi_{T1}$ and $\Phi_{Tn}$, respectively (Fig. 2d). The addition of a photobleaching path from $T_n$ was necessary for reproducing the experimental fatigue curves and not to underestimate the experimental photobleaching. Such a pathway is seemingly relevant since the triplet state of rsEGFP2 presents an absorption peak at 488 nm[42], and the triplet state absorption has been identified as a photobleaching promoter in EGFP[31].

Overall, the kinetic model is consistent with the data, further supporting that the photoswitching fatigue recovery stems from an optically-induced depopulation of the triplet state. We used the simulation tool to expand the observation over an extended interval of 405, 488 and 592 nm power density (Fig. 2h-i, Supplementary Note 9). In the simulated photoswitching fatigue experiments, the 592 nm illumination time was kept constant at 1 ms and overlapped in time with the 488 nm illumination pulse. Our simulations suggest that the triplet state can be depopulated effectively by co-illuminating with 592 nm (~kW/cm²), with the recovery being more evident at higher 405 nm power densities since a relevant fraction needs to be on-switched to yield enough $C$ concentration and accumulate triplet state (Fig. 2h). Similarly, the photoswitching fatigue recovery is more pronounced at higher 488 nm excitation power densities. In rsEGFP2 at high 488 nm power densities, the off-switching pulse length can be much shorter than the triplet lifetime (< 0.5 ms for 488 nm power densities > 600 W/cm², top half of Fig. 2i). In such conditions, the transient triplet population is effectively depleted during the 592 nm

only period, generating the most photoswitching fatigue recovery. The absorption of 592 nm light will not generate additional triplet via ISC but mainly depopulate the transient $T_1$ as shown in the scheme in Fig. 2d. Nonetheless, the photoswitching fatigue recovery reaches a plateau at 488 nm illumination power densities higher than 600 W/cm² and co-illumination of 592 nm power densities above 10 kW/cm² which indicates that the triplet state formation and RISC processes reach saturation under the experimental illumination parameters.

Shifting the red co-illumination toward the triplet-state absorption peak at 900 nm enhanced recovery (Fig. 2j). Furthermore, probing across the near-infrared range revealed that recovery efficiency is wavelength-dependent and closely follows the triplet absorption spectrum (Supplementary Note 10). Notably, 810 nm, where the triplet absorption coefficient is similar to that at 592 nm[42], resulted in less recovery than 900 nm, comparable to the difference observed between 592 nm and 915 nm. The increased recovery at ~900 nm is attributed to both stronger triplet-state absorption and reduced crosstalk with other photobleaching pathways, such as photobleaching from the on-switching intermediates (Fig. 2g, Supplementary Note 10).

## Photoswitching fatigue in other green-emitting RSFPs

We extended our investigation of the fatigue resistance to other RSFPs, focusing on green negative photoswitchers widely used in multiplexing[10–13] and super-resolution microscopy[28,38,46,47]. We observed a correlation between the off-switching speed of the RSFP and the experienced photoswitching fatigue (Fig. 3a, Supplementary Note 11), with slower photoswitchers (in particular, rsEGFP(N205S) and rsGreen1) being more prone to photobleaching upon cycling than their faster switching counterparts (rsEGFP2 and rsGreenF, respectively). The differences in photoswitching fatigue among the screened RSFPs can be rationalized by a less efficient off-switching process, which leads to longer illumination times per cycle and, consequently, a

higher probability of triplet-state formation. Assuming the triplet absorption spectrum is similar to that of rsEGFP2, the strong absorption of the $T_1$ state at the excitation wavelength results in an increase of photobleaching (Supplementary Note 11).

When comparing RSFPs, each exhibits a unique switching fatigue profile, therefore we set to explore whether fatigue can serve as an additional multiplexing dimension[10–13]. Multiplexing strategies generically refer to any method that aims to distinguish RSFPs that emit in the same spectral range by analysing fluorescence signal changes under specific illumination sequences. Methods like LIGHTNING[11] and TMI[12] resolve the kinetic rates of light-induced transitions, while approaches such as exNEEMO[10,13] capture fluorescence intensity changes in response to varying on-switching sequences. Bringing both approaches to high spatiotemporal resolution (e.g., confocal imaging with excitation ~100–500 W/cm$^2$) is difficult, since these techniques rely on the ability to clearly distinguish the off-switching kinetics, either as a fully sampled decay for LIGHTNING and TMI, or as integrated fluorescence for exNEMO (Supplementary Note 12).

To overcome this limitation, we tested photobleaching dynamics as an additional dimension to aid multiplexing strategies, resolving ambiguities under high irradiance. We designed a pulse scheme with four acquisitions per pixel, varying the on-switching energy dose to produce distinct fluorescence responses for each of the four proteins (Fig. 3b). At each pixel, the sample undergoes the pulse sequence in Fig. 3b, with the 4$^{th}$ frame normalizing the recorded intensity of the first three, similar to exNEEMO[10,13]. To account for photobleaching, 60 frames, 15 unmixing images, were collected for differently labeled bacteria, and the fluorescence loss was calculated. Data for each RSFP were fitted to a Gaussian model in a 4-dimensional space (Fig. 3c), including normalized intensities from the first three image stacks and fluorescence loss. To evaluate the unmixing method, all identified bacteria were reanalyzed using a Gaussian mixture model containing the information of the individual proteins. This pulse scheme is compatible with a parallelized confocal microscope, allowing the acquisition of unmixing images with ~ 0.5 Hz temporal resolution and over an extended field of view (FOV) ~ 40 × 40 μm$^2$, distinguishing four RSFPs with high accuracy (Fig. 3b). This approach demonstrates that tracking the photoswitching fatigue in samples with multiple RSFPs enables accurate identification of individual markers.

We then investigated if the reduction of fatigue by 592 nm co-illumination is a general phenomenon across various RSFPs variants. Our data show that rsGreen1 and rsGreenF had a significant recovery of photoswitching fatigue upon co-illumination with 592 nm light (Fig. 3d). We also measured the recovery as a function of the delay between 488 nm and 592 nm illumination pulses (Supplementary Note 11). The recovery magnitude decreased as the delay between the pulses increased, indicating that the effect stems from triplet-state depopulation, consistent with the behavior observed in rsEGFP2. Furthermore, fatigue recovery due to thermal relaxation was reduced in slower photoswitchers (Supplementary Note 11), suggesting that the maximum recoverable fraction is lower than that of faster variants. The differences in photoswitching fatigue behavior are likely due to diminished triplet-state shielding and, consequently, enhanced triplet-mediated photobleaching.

**Prolonged time-lapse imaging at high spatiotemporal resolution with RSFPs**

Motivated by our findings in a controlled environment, such as PAA gel or bacteria expressing the different RSFPs, we investigated the strength of the phenomenon in living cells to achieve prolonged time-lapse imaging in different modalities.

Taking advantage of our parallelized confocal microscope[37,48] we imaged the mitochondrial outer-membrane-protein-25 (OMP-25) labeled with rsEGFP2 in U2OS cells. In this configuration, a spatially patterned 405 nm illumination created by a micro-lens array (MLA) will

on-switch fluorophores at multiple focal spots in the xy-plane of the sample while the fluorescence is interrogated with another identical and spatially overlapped pattern of 488 nm light. To form an image, the sample is scanned between consecutive focal spots with a step size tuned depending on the target resolution. Sampling according to the Nyquist criterion, in a confocal image, the average cycling of every protein is less than 10 times. We introduced the 592 nm co-illumination simultaneously with the fluorescence readout step, co-aligning a 592 nm illumination with MLA to trigger the optically-induced depopulation. Our data show a decrease in photobleaching after 500 frames with ~10 kW/cm$^2$ of 592 nm co-illumination (Fig. 4a). As shown in Fig. 4a, the features of the mitochondrial network are better preserved after hundreds of imaging frames when the 592 nm pulse is added. Moreover, we compared photobleaching of timelapses with similar initial signal-to-background ratio (mostly dependent on the transfection efficiency) and observed that photobleaching was slower with the 592 nm co-illumination (Supplementary Note 13). Our results show that optically-induced depopulation of the triplet state can prolong time-lapse optical imaging of living cells.

To explore the benefit of red co-illumination in multiplexing imaging, we imaged bacteria expressing two RSFPs in confocal modality, rsGreen1 and rsGreenF. The nature of the protein can be identified by resolving the off-switching decay of the proteins, like in TMI[12] (Fig. 4b), and the global fluorescence is monitored over 50 frames. Without red co-illumination, the weaker RSFPs set the absolute number of frames of acquisition, while the addition of the 592 nm co-illumination extends the absolute observation window for the multiplexing imaging by 20% to 30% (Fig. 4b). Our data show that 592 nm co-illumination can be a powerful aid to extend in time high-resolution single-channel multiplexed imaging with RSFPs in combination with existing unmixing methods.

We further tested the feasibility of the method for super-resolution imaging recordings, specifically in parallelized RESOLFT nanoscopy[37,48]. In the parallelized confocal geometry, the nanoscale spatial confinement is achieved by off-switching the RSFPs located in the periphery of each focal spot with a grid-like pattern generated by interfering waves. The parallelization of the imaging scheme minimizes the illumination intensities without compromising the overall frame time. As with any other RESOLFT or NL-SIM imaging scheme, photoswitching fatigue represents the ultimate limit to the maximum number of frames which can be recorded.

Similarly to the parallelized confocal imaging experiments, the 592 nm illumination is delivered simultaneously and co-aligned to the fluorescence readout step (Fig. 4c). As the majority of the triplet population will reside in the center of each focal spot, the optically-induced depopulation will be more effective in this position. We observed a decrease in the photobleaching fraction after 20 frames in U2OS cells endogenously expressing vimentin-rsEGFP2 with an increasing 592 nm illumination (Supplementary Note 13) as well as in U2OS cells expressing LifeAct-rsEGFP2. Using the same pipeline, we observed ~20% reduction in the photobleaching fraction after 20 frames (Fig. 4c) by adding the 592 nm co-illumination, which competes with reported chemically-based strategies for reducing photobleaching[25,26]. As shown in Fig. 4c, co-illumination with 592 nm helped in maintaining the signal-to-noise ratio (SNR) during the time-lapses. Detailed features of the actin network of the cells were still visible after 20 frames (Fig. 4c). Similarly, we transfected U2OS cells with LifeAct tagged to other green negative photoswitchers and evaluated the effect of the 592 nm co-illumination in the same manner (Fig. 4d). Our data in cells correlates well with our findings in the PAA gel for the different RSFPs, showing that 592 nm co-illumination can be a general strategy for reducing photoswitching fatigue in other reversibly switchable fluorescent proteins.

Additionally, we monitored whether the 592 nm co-illumination induced any phototoxicity in the cells. An assay using the

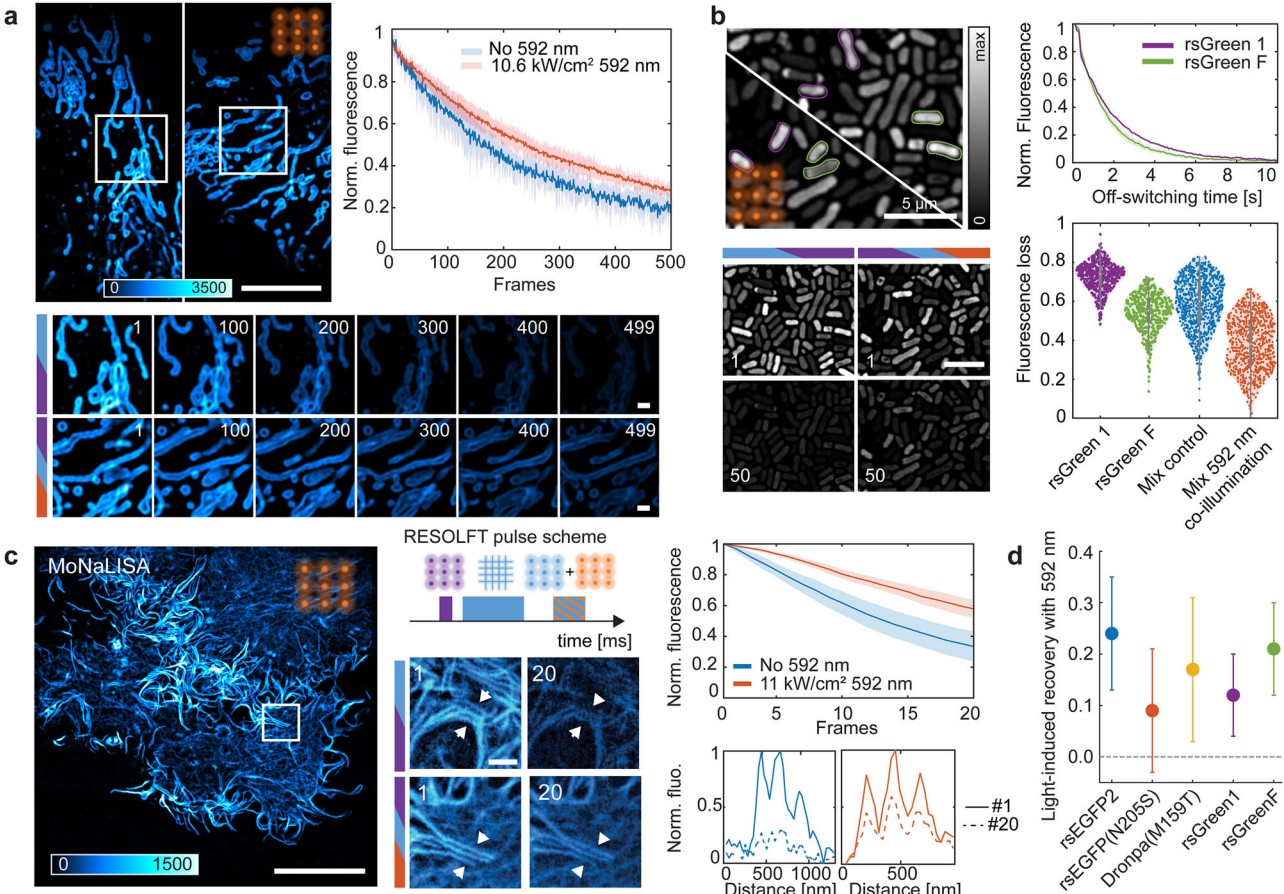

**Fig. 4 | Minimizing photoswitching fatigue in advanced live-cell imaging.**
**a** Parallelized confocal imaging of OMP-25-rsEGFP2 U2OS cells. Fluorescence signal evolution is shown for cells imaged with and without 592 nm co-illumination. Each curve represents the mean of photobleaching traces from 4–5 cells per condition. The shaded area indicates the standard deviation from the mean ($\pm \sigma$). Scale bars, 10 μm (overview) and 3 μm (zoom). **b** Identification of photoswitchable proteins in bacterial mixtures. A bacterial sample expressing rsGreen1 and rsGreenF was analyzed by fitting the off-switching kinetics during a confocal time-lapse (50 frames). Individual bacteria were classified as rsGreen1 (green) or rsGreenF (magenta). The violin plot shows the distribution of fluorescence loss under four conditions: rsGreen1 only ($N = 542$), rsGreenF only ($N = 462$), mixed sample ($N = 896$), and mixed sample under 592 nm co-illumination ($N = 777$). In the box plots, the center line marks the median, box limits denote the interquartile range (IQR), whiskers extend to $1.5 \times$ IQR, and individual measurements are shown as dots. Scale bar,

5 μm. **c** MoNaLISA imaging with reduced photobleaching. The 592 nm light dose was delivered simultaneously with 488 nm readout to reduce switching fatigue. Representative LifeAct-rsEGFP2 U2OS cells demonstrate photobleaching mitigation: actin filaments remained visible after 20 frames with 592 nm co-illumination, whereas they disappeared without it. The graph shows averaged fluorescence decay for 4–5 cells per condition; shaded regions represent $\pm\sigma$. Scale bars, 10 μm (overview) and 1 μm (zoom). **d** Light-induced recovery across green negative photoswitchers. Recovery after 20 imaging frames was quantified for U2OS cells expressing LifeAct tagged to different photoswitchable proteins under 592 nm co-illumination. Power densities were 11 kW/cm² for rsEGFP2; 4.8 kW/cm² for rsEGFP(N205S), rsGreen1, and rsGreenF; and 6.5 kW/cm² for Dronpa(M159T). Recovery was computed as the difference in mean fluorescence loss with and without 592 nm illumination. Errors represent propagated standard deviations from at least three independent measurements per condition.

DNA-repairing protein XRCC1 (X-Ray cross-complementary factor 1)[49] did not report additional DNA photodamage from the 592 nm illumination in the cells (Supplementary Note 14). Similarly, the mobility of mitochondria was assessed as a phototoxicity parameter. Both datasets, with and without 592 nm co-illumination, were inspected for spontaneous events, such as branching, fission, fusion, long stretching and long retraction, and a time stamp of their occurrence was annotated. Our data show the similarity in the cumulative distribution of events with and without 592 nm co-illumination (Supplementary Note 14), suggesting no adverse effects arise from the additional 592 nm illumination regarding mitochondria mobility.

## Discussion
We reported an all-optical solution for maximizing the number of photoswitching cycles in reversibly switchable fluorescent proteins. By studying the fluorescence signal loss during photoswitching at different illumination conditions and rationalizing our observation with an effective kinetic model, we identified two distinct contributions: an

initial drop due to the kinetics of the on-switching process, and a progressive decrease caused by photobleaching. We demonstrated that the former can be mitigated by modifying the delivery of the on-switching energy dose, yielding enhanced fluorescence output with longer illumination times and reduced power density, while the progressive photobleaching can partially be recovered by co-illumination at 592/900 nm light. The experimental characterization of the photoswitching fatigue of rsEGFP2, aided by the effective kinetic model, identified two main photobleaching pathways, one deriving from the triplet state of the on-state and another one linked to the dynamics of the on-switching mechanism of rsEGFP2. The experimental observations point to the optically-induced depopulation of the triplet state as a viable mechanism for photoswitching fatigue reduction at illumination intensities compatible with confocal and super-resolution imaging. The on-off switching partially shields the triplet population, making $T_1$ accessible upon exposure to light matching the triplet absorption spectrum, e.g., 592 and 915 nm. Our results showed that acting on the identified intermediate species involved in the

photocycle of RSFPs can lead to prolonged and brighter live cell imaging experiments. Further expansion of the kinetic modeling, for example, by incorporating additional long-lived dark states (e.g., radicals) or extended pH equilibria, would generalize the kinetic model to a wider range of experimental schemes beyond live cell imaging.

These strategies are particularly suited for microscopy modalities using repetitive light exposure and fast frame rate, such as parallelized confocal microscopy, NL-SIM and parallelized RESOLFT. The on-state modulation can be implemented in any microscopy modality by simply extending the illumination time; however, the longer dwell times required may hinder capturing rapid dynamics. On the other hand, co-illuminating the sample with 592 nm to reduce photobleaching is readily implementable in fast imaging acquisition, with respect to 900 nm illumination, where specific NIR compatible optics would be required. We translated the characterization and simulations in a pulse schemes for confocal and RESOLFT nanoscopy to effectively mitigate photobleaching. Given that the phenomenon is conserved among other green negative photoswitchers, we provide an imaging solution that advances image multiplexing via RSFPs kinetics. In these experiments, the choice of illumination typically favors one specific variant, for example, a fast switcher, at the cost of higher fatigue for the slower switcher or vice versa. By interleaving red light, it is possible to equalize the fatigue curve for different RSFP variants, resulting in improved imaging contrast. We showed that with the proposed imaging scheme, the SNR of fine sub-cellular structures is maintained across tens of imaging frames.

There is a trade-off between spatiotemporal resolution and photostability in any live-cell imaging application. In this context, all-optical approaches to reduce photobleaching offer advantages, as such methods do not interfere with optimized protocols in sample preparation and can be complementary to additional chemically based strategies designed to further improve photostability. By using 592 nm light, a wavelength already available on most commercial systems, to drive triplet-state recovery, this strategy can be seamlessly integrated into diverse imaging modalities. However, because 592 nm is a common wavelength for the excitation of orange- and red-emitting fluorophores, this wavelength may impair multicolor acquisitions. This limitation can be addressed by exploring other wavelengths within the triplet absorption spectrum of rsEGFP2, which peaks around 900 nm[42]. Near-infrared co-illumination successfully extended the number of frames accessible to widefield imaging with EGFP without inducing additional photodamage[31]. In this regard, the finding of a light-induced depopulation mechanism of the triplet state in RSFPs would be impactful in all the applications that utilize photoswitching within biological imaging, from super-resolution imaging approaches such as RESOLFT, as presented here, and pcSOFI, to single protein tracking applications or novel spectroscopy techniques that have photoswitching of fluorescent proteins at their core.

## Methods

### RSFP characterization in PAA gel

All RSFP characterization experiments were done by embedding the protein of interest in PAA gel. To prepare a sample, a few μl (depending on the RSFP) of purified protein were diluted in phosphate buffer for a total volume of 42.0 μl. The protein solution was mixed with 30.0 μl acrylamide (Rotiphorese Gel 30, Roth, Germany), 0.75 μl 10% ammonium persulfate (Ammonium peroxydisulfate ≥ 98%, Roth, Germany) and 1 μl 10% TEMED (15524-010, Invitrogen). The final solution was thoroughly mixed by vortex, and a small volume (10–20 μl) was placed on a glass slide (# 1.5), and a coverslip was pressed onto the sample to produce a thin layer. After complete polymerization, the sample was sealed with silicon-based glue (Picodent Twinsil, Picodent, Germany) to prevent drying out.

Before each photoswitching fatigue experiment, characterization of the off- and on-switching kinetics of the sample was performed. The off-switching time parametrization was done by averaging 25 off-switching profiles recorded in a small FOV ($\sim 1 \times 10\,\mu m^2$) at different 488 nm illumination ($\sim 10$–1000 W/cm$^2$). For a given 488 nm power density, the characteristic off-switching time is such that the signal has decayed to 20% of the initial value. To characterize the on-switching response, a $20 \times 20\,\mu m^2$ FOV is consecutively photoswitched with increasing 405 nm illumination ($\sim 10$–1000 W/cm$^2$). At each level of 405 nm light, the camera is exposed for the calibrated 488 nm off-switching time, and the measurement is repeated 9 times. The mean of the integrated fluorescence for each cycle, with its standard deviation, is measured and plotted against the 405 nm power density.

To record a photoswitching fatigue experiment, a $20 \times 20\,\mu m^2$ FOV underwent multiple photoswitching cycles ($\sim 1600$–2200 depending on the experiment). The camera exposure time was matched to the 488 nm pulse width, and the integrated fluorescence signal was collected for that photoswitching cycle. At the onset of the new cycle, a 1 ms dose of 405 nm light reset the majority of the ensemble population back to the emissive state. Additionally, a 592 nm wavelength was incorporated into the pulse scheme. The typical pulse dwell time was 10 ms, but was adjusted depending on the 488 nm power density.

The nonlinearity of the on-switching process in rsEGFP2 was tested by modifying the delivery of the 405 nm illumination with different combinations of power density and illumination times while keeping a constant total energy of 64 mJ/cm$^2$. The total dose of 64 mJ/cm$^2$ was chosen by previously calibrating the sample to an on-switching ramp with a 1 ms 405 nm illumination time. The same area of $20 \times 20\,\mu m^2$ was repeatedly exposed to 25 on-off photoswitching cycles for every combination of 405 nm power densities ($\sim 3$–200 W/cm$^2$) and illumination times (20–0.3 ms). The fluorescence signal was captured by the camera during the 488 nm illumination. For a given experiment, the FOV is initially illuminated with only 488 nm; the fluorescence retrieved in this initial cycle is considered to represent the true concentration of fluorophores in that volume. The fluorescence output for every on-switching condition is normalized to the integrated signal in that initial photoswitching cycle for every FOV. Each data point was built by averaging 20 of the 25 photoswitching cycles for a given condition. The experiment was repeated in 4 different areas of an rsEGFP2-embedded PAA gel.

Moreover, to characterize processes with a higher temporal resolution, a confocal system equipped with an avalanche photodiode (APD) detection system was used. To examine the on-switching mechanism, a burst (5 μs) of high-power 405 nm light ($\sim 30$ kW/cm$^2$) triggered the on-switching transition, after a variable delay (5 μs – 50 ms), the subsequent fluorescence signal was read out with 488 nm illumination at ~40 kW/cm$^2$. The area below the off-switching curve was calculated and plotted against the variable delay. Similarly, we investigated the relaxation of a dark state formed upon 488 nm absorption from the emissive state. In this case, a 5 μs 405 nm burst ($\sim 30$ kW/cm$^2$) was used as the on-switching light dose, after a 3 ms delay—to ensure full relaxation to the emissive state—a first 488 nm dose (500 μs at ~40 kW/cm$^2$) was delivered to the sample, enough to decrease the observed signal down to 80–90% of the initial value. After that, a second 488 nm pulse with the same illumination dose was delivered. The delay between consecutive 488 nm pulses systematically varied between 5 μs and ~ 500 ms. The area below the second off-switching curve was compared to a control curve, where no delay existed between pulses, and plotted against the delay time. All the experimental parameters are reported in Supplementary Table 9.

### Optical set-ups for characterization experiments

The photoswitching fatigue experiments, as well as the characterization experiments, were carried out on a modified version of the set-up reported here[48] (Supplementary Fig. 28). In brief, 3 different wavelengths were combined in a WF microscope configuration with a high

numerical aperture (NA) objective (HCX PC APO 100×/1.40 NA oil-immersion; Leica Microsystems). Photoactivation and fluorescence excitation were carried out using modulated laser diodes, 405 nm (200 mW, 06-01, 405 nm; Cobolt, Hübner Photonics) and 488 nm (200 mW, 06-01, 488 nm; Cobolt, Hübner Photonics) respectively. To test the photoswitching fatigue recovery, two longer wavelengths were incorporated, 592 nm (DEMO-2RU-VFL-P-2000-592; MPB Communications) modulated by an acousto-optic tunable filter (AOTF, AOTF-nC-VIS-TN; AA Optoelectronics) and a modulated laser diode 915 nm (250 mW, 06-MLD, 915 nm, Cobolt). The emitted fluorescence was directed to the detection path with a dichroic mirror (ZT405/488//640/775RDC; Chroma) and later to a scientific CMOS (sCMOS) camera (ORCA Fusion C14440-20UP; Hamamatsu). In the microscope, the light doses were digitally modulated by transistor-transistor logic signals with an NIDAQ PCI 6371 (National Instruments) acquisition device. The NIDAQ instrument synchronized the pulse sequence and the camera readout. The length and delay of the laser pulses, the output power of the lasers, the exposure time of the camera, and all other relevant parameters can be controlled through ImSwitch[50]. For sample screening, we added a wide-field path, which we could alternate with the main WF illumination by flipping a motorized mirror. The power densities of the different beams were calculated by measuring the power at the objective back-aperture, and the area was computed by fitting a Gaussian function to the beam profile imaged on an Alexa-Fluor488 thin layer, for 405 and 488 nm beams, and Abberior STAR RED, for 592 nm beam.

Experiments in which higher temporal resolution was required were performed using the confocal microscope configuration shown here[8] (Supplementary Fig. 29). In brief, 405 and 488 nm lasers were combined and tightly focused to a diffraction-limited volume with a high NA objective (APO 100×/1.40 NA oil immersion; Leica Microsystems). Photoactivation and fluorescence excitation were carried out using modulated laser diodes, 405 nm (125 mW, 06-01, 405 nm; Cobolt, Hübner Photonics) and 488 nm (70 nm, 06-01, 488 nm; Cobolt, Hübner Photonics) respectively. The 405 nm polarization was controlled with achromatic $\lambda/2$ and $\lambda/4$ mounted on remotely controlled rotating stages (Thorlabs) and set to circular polarization. To record an experiment, the microscope incorporates a $xy$ galvo scanning system (Cambridge Technology) to scan the light beams across the FOV. The emitted fluorescence is later de-scanned and directed to a pair of APDs (Micro Photon Devices). The counts recorded in both detectors are summed together later in the data treatment. The microscope components were controlled using a field-programmable gate array card (FPGA, PCIe-7852, National Instruments) and custom-designed software in LabVIEW. The power densities of the different beams were calculated by measuring the power at the objective back-aperture, and the area was computed by calculating the FWHM of the beam's PSF from a gold-bead sample. The schematic of the microscopes, as well as a list of the optical components, is reported in Supplementary Note 15.

The experiments characterizing the photoswitching fatigue recovery effect of near-infrared (NIR) triplet absorption on rsEGFP2 photoswitching fatigue were performed on an epi-illuminated, commercial confocal laser scanning microscope (Olympus FV1200). The setup was integrated by a combination of elements from two previous works[51,52] (Supplementary Fig. 30). The PAA-embedded rsEGFP2 was on-switched by a train of 405 nm (LDH-D-C-405, PicoQuant GmbH, Berlin) picosecond pulses extending for 1 ms at a 80 MHz repetition rate and an average power density of ~300 W/cm² at the sample plane. After a 1 ms delay, the confocal volume was illuminated with a train of picosecond 488 nm (LDH-D-C-485, PicoQuant GmbH, Berlin) pulses spanning 1 ms at the same repetition rate as the 405 nm laser and an average power density of ~14 kW/cm² at the sample plane. The diode lasers were controlled via a laser driver (Sepia PDL 828) with a repetition rate of 80 MHz. Since the pulse width of the laser was less than

100 ps, a train of pulses constituted a larger pulse. The pulse trains obtained for the final measurements are as mentioned, along with the optical setup schematic in Supplementary Note 15. After a 10 ms period of no illumination, the photoswitching cycle was repeated again for ~5000 cycles.

The NIR co-illumination was provided by a tunable Ti: Sapphire laser (Mira 900, Coherent, pumped by Nd: Vanadate laser (Verdi™ V-10, Coherent)) and was tuned to both 810 and 900 nm for scanning the triplet spectrum. The Ti: Sapphire laser was operated in a continuous mode, and its input was controlled using a mechanical shutter. Contrary to the characterization experiments in the WF configuration, the NIR co-illumination was applied during the whole duration of the photoswitching fatigue experiment. The power of the Ti:Sapphire laser was varied to achieve power densities of ~1–50 kW/cm² at the sample plane for both wavelengths (810 and 900 nm). The fluorescence was collected back from the same objective (UPlanSApo 60 × 1.2w, Olympus) and then passed through a dichroic mirror (ZT405/488/635rpc-UF2, Chroma), after which the light was focused onto a pinhole of 100 µm. A slightly wider pinhole size was chosen in order to have a larger volume to average the deformities present at every focal plane of the gel. A bandpass filter (535/70) was used in the emission path, and finally, the light was collected by two APDs (Tau-SPAD, PicoQuant GmbH, Berlin). The signals were directed to a data acquisition system (Hydraharp 400, Picoquant, Berlin) and time traces of the signals were exported via Symphotime and analysed with a home-built code.

**rsEGFP2 photophysical kinetic modeling**

To provide an insight into the mechanisms involved in the photoswitching fatigue, we built a photophysics simulation tool. To simulate the response of rsEGFP2, we emulated the experimental framework and calculated the population evolution given a proposed kinetic scheme. The system of kinetic equations is solved in matrix form. Since the photophysical evolution of the RSFP chromophore can be specified as a linear system of ordinary differential equations, it is possible to find an analytical solution to propagate a set of initial populations for each species. The solution involves the diagonalization of the kinetic rate matrix[53], which collects all the rates connecting the different species to each other. In general, to simulate a complex pulse scheme, we divide it into time windows—instances in which the laser states are not changing, and thus the kinetic rates are fixed—and solve the rate equations for the length of such a window. For consecutive time windows, the populations of the species at the last simulation point of one window are the initial populations of the next window. A more detailed description of the theory behind the photophysical simulations can be found in Supplementary Note 16.

The proposed photocycle model includes the 4 main states responsible for the photoswitching in rsEGFP2 – $TH, CH^i, C, T^\dagger$ - and an intermediate species, as described in the literature[15]. Furthermore, to parametrize the observed photoswitching fatigue we included two bleaching pathways: (i), triplet states bleaching, and (ii), photo-destruction of the chromophore from the off-state. The complete photocycle scheme with the different transitions between states is shown in Supplementary Note 1. Attaining the different levels of complexity that the photocycle model includes, different pathways were activated or silenced depending on the scope of the simulation. The parameters input in the kinetic model are taken from literature, properly referenced, or by fitting the kinetic model to our data. The parameters used for each of the reported simulations can be found in Supplementary Table 2.

**Protein purification**

Transformed E. coli cells (TOP10 and BL21 strains, pBAD and pQE expression systems) were grown overnight in LB medium supplemented with ampicillin. The next morning, cells were diluted 1:100 together with ampicillin 1:1000 in LB medium and added to 50-ml

tubes so that the tubes were filled to the brim, and the tubes were then sealed and incubated until the optical density (OD) at 600 nm reached 0.6 at 37 °C. Then IPTG was added to a final concentration of 1 mM, and the tubes were tightly sealed again to restrict oxygen availability. After overnight incubation, the cells were collected by centrifugation in the same tubes. The resulting bacterial pellet was sonicated and re-concentrated by centrifugation. The remaining supernatant was filtered (MinSart RC Hydrophilic 25 mm 0.2 µm, Sartorius, Germany), and the proteins were purified by Ni-NTA affinity chromatography (His SpinTrap; GE Healthcare) according to the manufacturer's instructions. The purified proteins were concentrated by ultrafiltration and taken up in phosphate buffer, pH 7.5.

## RESOLFT and parallelized confocal microscopy

Live-cell RESOLFT and parallelized confocal imaging were performed on a modified version of the microscope set-up reported here[48], the same one used for the photophysical characterization at low illumination power densities (Supplementary Fig. 28). To achieve a spatial resolution below the diffraction limit, the light was spatially patterned in the sample plane by introducing optical elements in the laser paths. Photoactivation was carried out at 405 nm in combination with a micro-lens array (MLA, MLA-150-5C-M; Thorlabs) to create a multi-spot pattern with 625 nm periodicity. Illumination at 488 nm and 592 nm was combined using a dichroic mirror (zt502rdc; Chroma) and focused by an MLA to create a multi-spot pattern at the sample plane with periodicity 625 nm, and that was aligned to the 405 nm illumination. To create the light pattern necessary for fluorescence confinement, the collimated light from the 488 nm digitally modulated diode laser (200 mW, 06-01, 488 nm; Cobolt, Hübner Photonics) was directed to a half-wave plate and polarizing beam splitter (PBS). The half-wave plate was adjusted to equally split the light equally into P and S polarizations. After the polarizing beam splitter, the light was sent through custom-made phase-diffraction gratings of 437-nm-high $SiO_2$ lines with a 25 µm period (Laser Laboratorium Göttingen). After the gratings, the light paths were recombined with another polarizing beam splitter. The ±1 diffraction orders are selected at the back focal plane of the objective to obtain a final off-periodicity of 312.5 nm. A piezo scanning system (BPC303; Thorlabs) was used to apply pulse illuminations across a small region equal to the multi-foci periodicity at precise steps (typically 35 nm but adjusted to the desired optical resolution). The light was digitally modulated by transistor-transistor logic signals with a NIDAQ PCI 6371 (National Instrument) acquisition device. The FOV for MoNaLISA imaging was ~$40 \times 40$ µm². The typical imaging pulse scheme consisted of 4 illumination sequences for RESOLFT imaging— on-switching, confinement, readout and recovery, and 3 illumination sequences for parallelized confocal imaging—on-switching, readout and recovery. The NIDAQ instrument synchronized the pulse sequence, the camera readout, and the scanning. The readout laser was synchronized with the sCMOS (ORCA Fusion C14440-20UP; Hamamatsu) camera exposure to detect fluorescence emission. The length and delay of the laser pulses, the output power of the lasers, the exposure time of the camera, and all other relevant parameters were controlled through ImSwitch[50]. A schematic of the imaging microscope, as well as a list of the optical components, can be found in Supplementary Note 15. The imaging parameters are reported in Supplementary Table 10.

## Cell culture and transfection

Wild-type U2OS (ATCC HTB-96) cells and U2OS with endogenously tagged vimentin-rsEGFP2[54] were cultured in DMEM (Thermo Fisher Scientific; 41966029) supplemented with 10% (vol/vol) FBS (Thermo Fisher Scientific; 10270106) and 1% penicillin-streptomycin (Sigma Aldrich; P4333) and maintained at 37 °C and 5% CO2 in a humidified incubator. For transfection, $2 \times 10^5$ cells per well were seeded on coverslips in a six-well plate. After 1 d, cells were transfected using Lipofectamine LTX Reagent with PLUS reagent (Thermo Fisher Scientific; 15338100) according to the manufacturer's instructions. 24–36 h after transfection, cells were washed in PBS solution, placed with phenol-red-free DMEM or Leibovitz's L-15 medium (Thermo Fisher Scientific; 21083027) in a chamber, and imaged.

## Ratiometric image calculation

We tested experimentally if the fluorescence enhancement we observed in the rsEGFP2-PAA layer system in Fig. 2a was visible in rsEGFP2 labeled live cells. In multiple cells we recorded two images: one we deemed *slow image*, which consisted of 20 ms illumination at 405 nm and 14 W/cm² power density followed by a fluorescence readout step with 488 nm, the other, which we named *fast image*, consisted of a 0.25 ms 405 nm burst at 1.05 kW/cm² power density followed by the fluorescence readout step with 488 nm illumination. The images were taken sequentially in confocal mode in our WF microscope. To calculate the ratiometric image for every *slow/fast image* pair, we pre-processed the pair of images by subtracting the background and masked with an intensity threshold using Otsu's method from the scikit-image Python library. The per-pixel intensity was divided as *slow/fast* to generate the ratiometric image (Supplementary Fig. 14). A general quantification is shown in Fig. 2b by accounting for all the valid pixels from a pool of 8 ratiometric images. To avoid photobleaching bias from one of the on-switching doses, the first image of each *slow/fast images* pair was alternated across different cells, i.e., cell 1 was imaged with the slow image pulse scheme first, cell 2 was first imaged with the fast image pulse scheme instead.

## Live-cell single channel multiplexing

Multiplexing experiments were carried out on transformed E. coli cells prepared as mentioned above. Bacteria expressing single RSFPs were imaged after centrifugation, and mixed samples were prepared by combining bacteria expressing different proteins at suitable concentrations. Two different methods for unmixing were employed: recording of slow off-switching curves and intra-pixel 405 nm intensity modulation inspired by exNEEMO[10,13]. For the former, the bacteria in the FOV were off-switched using the 488 nm illumination (~5 µW at the sample plane) of the auxiliary WF shown in Supplementary Fig. 28 with ~100 ms exposure time. Afterwards, the same bacteria were imaged in the parallelized confocal modality in the microscope described above. The scanning step between adjacent multi-foci was set to 90 nm. Since the WF and parallelized confocal images have different coordinate systems, the image stacks were transformed and registered with the MultiStackReg[55] plug-in in ImageJ. Following the registration, a custom-written segmentation routine in ImageJ was used to identify single bacteria in both WF and parallelized confocal image stacks and the intensity traces were extracted. The data clustering is performed by a custom-written Matlab script based on a Gaussian mixture model (GMM) that fits the data into a given number of Gaussian components, extracting μ and σ.

On the other hand, the intra-pixel 405 nm intensity modulation method consisted of a series of 4-frame parallelized confocal stacks with modulated 405 nm illumination doses. On every pixel, the sample is consecutively illuminated with 4 different pulse sequences the integrated intensity is collected during the 488 nm excitation and the modulation of the fluorescence levels is given by the on-switching dose. The sample is scanned between adjacent foci with a step of 90 nm. The microscope control software ImSwitch[50] was adapted to acquire N-images per pixel. A custom-written Python script splits the acquired frames into 4, and our reconstruction algorithm returns a stack of 4 frames, each with a modulated pixel intensity given by the input illumination sequence, for every acquisition frame. To build a time-lapse recording, this procedure is repeated 15 times, resulting in a total of 60 frames. Following the acquisition, a custom-written script in ImageJ is used for image processing and segmentation of individual

bacteria. The intensity traces of the identified bacteria are extracted and processed in a custom-written MATLAB script. Similarly to exNEEMO[10,13], the intensity of every 4th frame is used to normalize the intensity of the three previous ones and the average of the normalized 4-frame stack intensity trace across the 60 frames time-lapse is computed. Simultaneously, the fluorescence signal loss due to photobleaching is computed as the difference in average raw intensity between the first 3 normalization frames and the last 3 normalization frames in the stack. The data clustering is performed by a custom-written MATLAB script based on a GMM. The first training step extracts $\mu$ and $\sigma$ for the individual proteins. Afterwards, the whole dataset is reanalyzed using the fitted GMM, and the confusion matrix is created.

### Statistics & reproducibility
Data were analyzed using custom-written scripts in MATLAB R2021b, R2022b and R2024b (64 bit) and custom pipelines in Fiji macros and Python scripts for image analysis. The code developed for simulating and fitting the different rsEGFP2 characterization experiments was written in Python. The light doses employed to achieve super-resolution in MoNaLISA were tailored to every experiment day guided by a calibration of the imaging parameters before the experiement. Sample sizes, means, and standard deviations are indicated in the relevant figure legends. The light-induced recovery effect in all samples is reported as the difference between the mean fluorescence loss without (control condition) and with 592/915 nm illumination (experimental condition). The error associated with the mean difference is calculated by propagating the error from the standard error (typically, $\sigma$) for both conditions. Detailed information on the experimental parameters is reported in Supplementary Table 9 for the characterization experiments and in Supplementary Table 10 for imaging experiments.

### Reporting summary
Further information on research design is available in the Nature Portfolio Reporting Summary linked to this article.

## Data availability
The data supporting the findings of this study are provided in the Source Data of the main text and supplementary information. The images of Fig. 4 can be found in the Zenodo repository: 10.5281/zenodo.17581690. Source data are provided with this paper.

## Code availability
The code for the photophysics simulation of rsFPs is available at the GitHub repository https://github.com/marinaguilera/photophysics_simulator.git and https://zenodo.org/records/17579513.

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

## Acknowledgments

We thank Rebecca Mordini for assisting with the reviewing process. This work was supported by EU ERC InSPiRe (101002490), Swedish Research Council VR (2022-04415) and CZI Chan Zuckerberg Initiative TESTA 2023 (2303-2508) to IT; Swedish Research Council VR starting grants (2021-04528) to FP; Swedish Research Council VR (2021-04556) to JW.

## Author contributions

I.T. designed the research. I.T. and F.P. co-supervised the research. F.P., A.P., and G.M.A. performed the experiments and analysed the data. A.V. and G.M.A. developed the kinetic model. J.W., N.B., A.K., and G.M.A. contributed to the spectral experiments in the near-infrared. G.M. and G.M.A. modified the microscope control software ImSwitch to acquire the multiplexing recordings. I.T., F.P., and G.M.A. wrote the manuscript with the help of all the authors.

## Funding

## Competing interests

The authors declare no competing interests.
