## [Transparent Peer Review file · Nature Communications]

All-optical strategies to minimize photobleaching in reversibly switchable fluorescent proteins

Corresponding Author: Dr Francesca Pennacchietti

Version 0:

Reviewer comments:

Reviewer #1

(Remarks to the Author)

Review of NCOMMS-25-05441, "All-Optical Strategies to Minimize Photo-Bleaching in Reversibly Switchable Fluorescent Proteins," by Marin-Aguilera et al.

This paper develops optimized methods for using light to remove photoswitchable fluorescent proteins from undesirable triplet states that trap population for long time periods. This strategy not only makes the proteins less likely to photobleach, but also makes these molecules available to fluoresce more often. The experimental work in this paper is both thorough and compelling, and the work will be of great value to the community of researchers who use these proteins. The authors demonstrate that this strategy works for other green-emitting fluorescent proteins, and for a number of imaging modalities. I feel that this paper will have a high impact among those in the advanced optical imaging community, and is ultimately worthy of publication in Nature Communications. There is one crucial issue that I feel needs to be addressed first, as well as a range of minor issues. Please do not take the length of the list of the issues I have identified as a negative, I would simply like to see this work have as much impact as possible.

To begin with my major concern, the model that the authors present is far from convincing to me; in fact, I am pretty certain that it is flawed, for several reasons. First, along with virtually all of the prior literature in the field of photodeactivation of triplet states, this study makes the unfounded assumption that RISC solely leads to the population of an excited singlet state. In most circumstances, a more likely outcome is the population of a highly vibrationally excited level of the singlet ground state, because the vibrational density of states of S0 at this energy is vastly higher than that of any singlet excited state (this behavior is challenging to prove, but it can be done; see, for instance, Ref. 28 in the paper and [dx.doi.org/10.1016/j.isci.2021.103600](https://doi.org/10.1016/j.isci.2021.103600); as this journal encourages reviewers to reveal their identities, I will say at this point that this publication is one of mine.). This process, which we in the field typically (and, I would argue, incorrectly) call internal conversion (IC) in the case of T1, explains why phosphorescence quantum yields are not typically more than a few tens of percents for organic molecules. Indeed, the typical way to attain a high phosphorescence quantum yield in such molecules is to have the T1 energy be well below that of S1, which not only prevents RISC to S1, but also lowers the probability of IC. The vibrational density of states of S0 grows exponentially with increasing energy, and so the higher the triplet state, the faster RISC to the ground state will be as compared to RISC to an electronic state with an origin of similar energy to the triplet state. That does not mean that no population will undergo RISC to excited singlet states, but in many, if not most, organic systems the branching ratio for RISC to excited singlet states is likely to be small. As an example, in Ref. 29 the authors had to invoke photobleaching from the triplet state that also causes repopulation of the singlet manifold to explain the fluorescence intensity does not depend linearly on the intensity of the deactivation wavelength, but rather saturates. This behavior is a natural consequence of RISC going to the S0 state directly, as can be shown by trivial, analytical kinetic modeling.

Second, the model, at least as shown schematically, treats all Tn states as having identical behavior. Given what the authors observe, this is manifestly impossible. As mentioned above, the higher the triplet state, the more likely RISC is to occur (and 488-nm light was indeed shown in Ref. 28 to cause RISC to S0, which is missing from the authors' model), but at the same time it may also be more likely that some sort of photobleaching event occurs. There is absolutely no reason to believe that the branching ratio for these phenomena (not to mention relaxation to lower triplet states) should be the same following triplet absorption with two disparate colors of light. The triplet absorption spectra in Ref. 27 (which predates Ref. 28 and is more relevant to the work presented here) make it quite clear that the different wavelengths used here go to different excited

triplet states.

Third, although I appreciate that the authors did their best to use literature values for rate constants, I do not trust much of the literature based on the points raised above. Their model contains so many processes and rate constants that it could reproduce almost anything, even though the model is missing the processes discussed in the previous two paragraphs. Indeed, in my view their model is not compatible with the results in Ref. 28. Developing a model that incorporates verifiable photophysics will take a considerable amount of additional experimental work, and I would suggest leaving that for a future paper. Others will likely have a different opinion than mine, but I feel that the modeling work detracts from the paper because, in my opinion, it is at best incomplete, and more likely is inaccurate. Although some of the experiments performed may have been motivated by the model, presenting the interesting results while having those results be motivated by a model that will elicit skepticism in readers like me proves a distraction. I suspect that this opinion will (understandably) not be popular with the authors at first (and maybe afterwards as well), but I urge them to consider my suggestions strongly. It is better to get the modeling right (or to prove the current modeling to be correct) in a subsequent paper (which I feel will also have high impact) than to dilute the impact of impressive experimental work with a model that cannot currently be substantiated.

Other, more minor points that need to be addressed, in no particular order of appearance or degree of concern:

1. The authors play fast and loose with the optical spectrum. 405-nm light is violet; there is no such color as "UV-vis." 592-nm light is orange (and barely so, it's almost yellow), not red. Calling light "red-shifted" is confusing, and makes it sound as if the authors are looking at a side-band of another laser they used. Please give this light a clear name, such as "light at the triplet deactivation wavelength."
2. The authors use a lot of vague language in the paper, for instance talking about "several" rounds of photoswitching, and using many unclear antecedents ("it" and "this") that made the paper hard to read for me in some passages because I had to go back and figure out the antecedents. As I teach my students, this problem can be fixed by avoiding the use of "it" to refer to something prior, and to always follow "this" at the start of a sentence with a noun.
3. Many of the figure panels were difficult for me to parse (e.g., Figure 1d, Figure 1e, Figure 2a, Figure 2h, Figure 2 i,j). Many of the graphs are deceptive, with y-axes not starting at zero, thus exaggerating small effects. At very least, broken scales should be used in such cases. In Figure 2a in the right panel, please add a second x-axis on top that gives the intensity. 90 mJ/cm² (note that the 2 is not an exponent in the figure) is not an energy. To say that the model reproduces the data in Figures 2e,f is really a stretch. Within your uncertainty, the data in Figure 2f have no dependence on t whatsoever. I also cannot read whatever is in the color bars for the heat maps in Figures 2i,j.
4. There are places in the paper where the authors refer to a spectrum without saying type of spectrum. An example is line 51, "...the...triplet state of PPs has a red-shifted spectrum..." I presume that they mean absorption rather than phosphorescence, but they need to be clear. Another example is in line 180.
5. There are quite a few places in which numbers that should be exponents are not.
6. In the first sentence of the paper, the abbreviation FP is used without definition. There are other similar problems throughout the manuscript (e.g., EGFP, SNR).
7. In the second paragraph of the results section, it would be quite helpful if the authors explained what they mean by an "excitation cycle." The meaning only becomes clear later on in the paper.
8. There are many "gratuitous" hyphens following words that start with "photo." For instance, the correct word is "photophysics," not "photo-physics." "Infrared" also does not have a hyphen.
9. "Data" is plural, not singular, and Nature correctly follows that usage.
10. Many people who might use the results presented here in their own work have commercial fluorescence microscopes that are not capable of simultaneous exposure at two different wavelengths and are not able to do the sort of multiplexing that the authors demonstrate. Are the authors able to test how much improvement can be achieved if only alternating scans of different colors can be used? If this idea is not easy to pursue, that is not crucial to the publication of this work.

To end on a positive note, this work is quite impressive, and I hope to see the paper published in this journal. Please know that all of my comments are meant in an entirely constructive manner; the topic of your work is highly complex, and many of the issues that I have brought up are not well understood or appreciated in the community. I would like to see you publish this work in a manner that is not only exciting, but also "bullet-proof."

(Remarks on code availability)

Reviewer #2

(Remarks to the Author)

In this manuscript, Marin-Aguilera et al provide an in-depth study of photobleaching mechanisms in reversibly switchable

fluorescent proteins (RSFPs) and introduce means of reducing photobleaching under conditions typical of RESOLFT super-resolution microscopy by the addition of red-shifted illumination during switching cycles or possibly dark periods between switching cycles. The paper is a natural follow-up of the recent work of the group of Agathe Espagne (ref 29) who showed that near infrared light is able to induce reversed intersystem crossing from the triplet state of standard green fluorescent proteins such as EGFP, shortcutting photobleaching pathways typically arising downstream this state. Extending the work to the case of RSFPs is important due to the numerous applications that these markers offer in advanced microscopy approaches.

The paper is divided into three sections: (i) a thorough experimental investigation of photofatigue decay of purified polyacrylamide-embedded rsEGFP2 as a function of illumination conditions, (ii) an experimental investigation of the level of recovery provided by red-shifted illumination, reaching up to ~15%, and finally (iii) practical realization in the context of biological samples, showing a moderate but significant effect for time-lapse experiments using either parallelized confocal or RESOLFT microscopy.

In addition, the authors introduce a novel technique for kinetic-based RSFP multiplexing based on differential photofatigue decays between different variants.

All the work is embedded in a conceptual framework based on an elaborated photophysical model of rsEGFP2, allowing the authors, through extensive simulations, to account for the experimental observations made and possibly predict optimized behavior of RSFPs in a variety of imaging experiments.

Overall, this represents a large body of interesting and timely work, clearly to the benefit of photo physicists and fluorescent protein developers, and potentially to the benefit of the large community employing RSFPs as labels in their biological experiments. The photophysical aspects are treated in the specific context of intense illuminations, typical of RESOLFT-like experiments, which is rarely addressed in the rich literature concerned with RSFPs photophysics. This makes the paper particularly interesting to photophysicists. Thus, in principle I am supportive of this work. However, I have many interrogations concerning the validity of the proposed model. To my opinion many issues remain to be solved, that may either stem from a certain lack of clarity (the paper is highly complex to read and properly digest) or be related to more serious inconsistencies. All these issues call for a major revision of the manuscript.

Major issues concerning the proposed photophysical model of Figure 1e and S1:

1/ In principle the cis anionic and the cis neutral chromophore should be in rapid equilibrium, depending on pH. This equilibrium is not represented in Figure 1e, but should be, as the cis neutral chromophore, even if it is not highly populated at physiological pH, is able to absorb 405 nm light, isomerize to TH, and probably photobleach under such illumination. I guess that in Figure 1e, CH is meant to represent the "cis on like neutral" chromophore of ref 16, not the real "CH". If this is correct, the "equilibrium" cis neutral chromophore is missing in the model.

2/ According to Figure S1, the triplet state is only accessed from C-, and does not absorb 405 nm light. Yet, the authors make a very strong statement at the beginning of the manuscript (P4-L97: "the data reports the 405 nm light to be the main driving force for increased photo-switching fatigue in rsEGFP2"). If the triplet state is indeed one of the main photobleaching mechanism induced by 405 nm light absorption, either it should be accessible by ISC from a 405 nm absorbing species (a protonated species) or the triplet itself should be able to absorb this light. Otherwise triplet state bleaching must arise from 488 nm light absorption, and in fact this is underlying the discussion along the whole manuscript. This is very confusing. For example, the initial rise in the fatigue fraction seen in Figure 1f relates to 488 nm induced bleaching from an increasingly high C- population due to increasing 405 nm light, which is an indirect effect. In view of this introductory sentence stating that photobleaching relates to 405 nm light, and also figure 1C showing little effect of the 488 nm light, all this needs strong clarification.

3/ In fact, the authors need to justify why T1 (Figure S1) would not absorb 405 nm light. This is by no means suggested by the experimental spectrum in ref 28, which shows at least as much absorption at 405 nm compared to 592 nm. It is in fact likely that T1 absorbs violet light. Have the authors considered the scenario where the off switching phase populates T1 and the immediately following on switching phase leads to reabsorption of 405 nm light by T1 ? Adding a delay between switching cycles would relieve this switching pathway, as observed.

4/ As mentioned by the authors, the data in Figure 1g suggests the presence of one (or several) additional dark state(s) with a lifetime significantly greater than that of T1 (P4-L122). Such additional dark states are indeed expected, and may be radical species. They, too, may absorb 405 nm light to promote either recovery or photobleaching. Can the model of the authors reproduce the data of Figure 1g ? I suspect not, because the long-lived dark states in question are not part of the model, although they are probably crucial to the behavior. The authors might want to consult the following paper: Duan et al, DOI: /10.1021/ja406860e.

5/ The authors, based on Supplementary Note 3, conclude that the Int intermediate absorbs light at 488 nm and is substantially fluorescent. How do they reconcile this finding with the fact that Int has been reported in several publications to be a protonated species, hence with little absorption at 488 nm and, in principle, very low fluorescence ? The authors report an extinction coefficient of Int at 488 nm of 28,000, based on their data and also based on ref 2 (supplementary table 2). This is very unlikely for a fully protonated species, and ref 2 does not support such value (the authors have probably misread table 1 in ref 2, the extinction coefficient of ~28000 of the fully neutral species is at 403 nm, not 488 nm). It is difficult, based on supplementary figure 3A, to understand how supplementary figure 3B is obtained: at early delays, the integrated

fluorescence from this figure is about half of the full signal at late delays. But this is not what is suggested by looking at panel A, where the area under the blue curve (5 μ s) seems much less than the area under the red curve (51 ms), not even considering the baseline due to residual background. So please explain how supplementary figure 3B is obtained. If a residual substantial amount of the on-state is present at the beginning of the second 488 nm pulse, this will contribute to the fluorescence signal in all curves of panel A, so how do the authors make sure that this amount is negligible? This being said, I agree that the microsecond rise in the fluorescence could point at Int being fluorescent. All in all, the authors should carefully clarify this point. If they maintain that indeed Int must be fluorescent, they should state it clearly in the main manuscript, as this is a very important finding in view of the available literature, and they should provide a possible mechanistic scenario. Furthermore, if Int is indeed fluorescent, why would it not undergo ISC to the triplet state in the proposed model? Is there any good reason to neglect this possibility?

6/ On which basis do the authors assume that C- may convert to Int as a ground state process? What is "tau_protonation-C" in Supplementary Table 1? This parameter is not reported in Supplementary Table 2. Is it used in the actual simulations?

7/ The authors invoke a direct bleaching pathway from TH, which apparently does not involve the triplet state. The authors should explain why the triplet state is not considered here? Is this because of the short lifetime of the nonfluorescent TH? This would make sense, but if the triplet state is not involved, then, photobleaching must arise from the excited singlet state manifold. The authors should provide more details here, notably as RISC is not going to help at all in this case, meaning that, according to the author's view, light induced recovery by red shifted light would not directly relate to 405 nm induced photobleaching (see point 2 above) ... in stark contradiction with the statement that photobleaching mainly arises from excitation at this wavelength. Could it still be that the triplet state and downstream radical species be reached from neutral chromophore species? The authors might be interested to read: Wulflele et al, DOI 10.1021/acs.jpcllett.2c00933.

8/ The authors find that increased power density 488 nm results in higher light induced recovery (figure 2e). This is unexpected in view of the fact that a high-level of 488 nm illumination should by itself promote RISC, decreasing the additional impact of red shifted light. In fact this was noted in refs 28 and 29. I feel that the data shown in figure 2e are misleading, as here the dose of illumination at 488 nm is kept constant, that is, the data at 0.42 kW/cm² are collected with a shorter pulse of 488 nm light than the data at 0.2 kW/cm², while at the same time, if I understand correctly, the 592 nm illumination is kept at 1 ms duration. In this situation, the observation can be easily explained, as, as soon 488 nm illumination terminates, the additional time until the next cycle becomes free for either thermal or light induced relaxation from T1 to S0. These aspects should be clarified to avoid this possible misunderstanding that the more 488 nm illumination the more 592 nm light induced recovery by RISC.

9/ As shown by the authors the RISC effect can as well be obtained by near-infrared light, at 810 or 900 nm. In fact, near infrared light should be the privileged choice, as absorption of the triplet state around 900 nm has been shown to be maximum in green fluorescent proteins. The authors have used 592 nm light probably because this was technically easier (already implemented on their microscope). Interestingly, they found that the achievable effect with 592 nm surpasses that at 810 nm, even with the huge power densities they used in the near infrared. It would be nice to give a precise interpretation of what might be going on. At the end of supplementary note 7, the authors claim that the comparison is difficult based on the fact that the "spatial distribution of light-induced kinetic processes is radically different". This statement is unclear to me, it would be nice if the authors could elaborate more. In all cases it looks like, from supplementary figure 13, that no matter what intensity is used, the level of light induced recovery at 900 nm cannot be reached at 810 nm. Would the authors have a clue whether this is true and if yes, why? At the end, using visible red light to promote RISC is nice, as this wavelength is available right away on many microscopes. However, the authors should comment on possible penalties e.g. for 2-color experiments requiring 592 nm light to excite a second red fluorescent marker.

10/ It would be nice to see the experimental data of figure 1F on a linear scale and without the fitted model data superimposed. It looks like these experimental data may follow a simple exponential growth on the used logarithmic scale, meaning a linear growth on a linear scale. Many readers could feel that using such a complicated photophysical model to fit essentially linear data is largely unjustified. Could the authors comment on this? What are the factors that prevent from only relying on a linear photobleaching effects by the 405 nm light?

11/ I am puzzled by supplementary figure 4: the data in this figure show the rapid growth of the integrated fluorescence which then levels off without any sign of further decay. On the contrary, the model predicts a decay, as Int is more and more redirected towards TH when the 405 nm power density is increased, and this decay is clearly visible in the fitted data. The experimental data here clearly seem to deviate from the model. Furthermore, the data of figure 1B seem to go in the same direction, as the initial point in the photofatigue decay curves appear to be significantly higher at 0.7 kW/cm² as compared to 0.22 kW/cm², as opposed to what the model would predict. In their experiment of figure S4, could the authors increase further the intensity of the 405 nm illumination to start seeing the expected decay (maybe using their confocal microscope)? Could it be that Int may also be photoconverted to C- by violet light, so that the relative amount of Int molecules sent to either C- or TH be relatively independent of 405 nm illumination? It is known that chromophore deprotonation in FPs can be accelerated by light. These aspects should be discussed.

12/ The scheme in Fig1e is difficult to read: the triplet state between C- and B should be shown in clear, notably as the trans states (T) could be confused by some readers with the triplet state (T1). The protonated chromophore state in equilibrium with C- should be shown as CH, and the short-lived intermediates preceding Int along the on switching pathway should be given a different name.

Overall, I strongly encourage the authors to revisit their photophysical model in depth. They should also much improve the clarity of their manuscript so that the essential message carried by this model can be captured by the white audience of

Nature Communication. At present, what I can extract from the manuscript is that: (i) the effect of 405 nm illumination on photofatigue either relates to the triplet state only via the amount of the on-state population at the beginning of off switching phases, or goes through a singlet state photobleaching mechanism that is nonrecoverable by RISC and (ii) recoverable photobleaching through T1 only occurs upon 488 nm illumination of the on-state.

One possible strategy that the authors could consider would be to investigate whether their photophysical model predicts a pH dependence of photofatigue and to carry out corresponding experiments to see whether these predictions are realized. For example, at low pH the fraction of protonated chromophores will increase, likely increasing direct bleaching by 405 nm light. It could also be expected that relaxation of Int to C- slows down, making the initial drop faster. In view of the complexity of the problem, those pH dependent experiments could help to narrow down the photophysical model.

Other major points

13/ The authors nicely apply their methodology to both prokaryotic and eukaryotic samples. Figure 4A and 4C suggest that there is a real benefit of applying 592 nm co-illumination. Yet, it is not clear whether the protecting effect shown in the photofatigue decay curves of figure 4A is as pronounced as that observed in vitro. Could the authors comment on this? Do they see a possible effect of polyacrylamide in augmenting the effect, maybe in relation to oxygen level and/or diffusion, and thus triplet state lifetime, and even maybe propensity for radical formation? In view of those photofatigue decay curves in figure 4A, which after all predict a relatively faint improvement, how representative are the series of images shown underneath? The authors may want to show the photofatigue decay of this very sample in the SI. In passing, the first frame of the "with co-illumination" series appears less intense than the first frame of the "without co-illumination" series. Does this correspond to a true, reproducible effect? If yes, what happens?

The Mona Lisa data are also nice and seem to show a significant gain. Would there be a way to more quantitatively assess the spatial resolution loss along multiframe data collection with and without co-illumination, rather than showing just one profile at a user chosen position (which is the classical old procedure often criticized for potential bias)? Possibly, a number (~10) of random locations across multiple fibers could be picked and the profiles presented in the SI?

More minor issues

1/ The authors investigated the effect of modulating the on switching light (essentially adding dark times, or reducing power density at constant dose, both resulting in increased data collection time) in the context of their photophysical studies, but they have not implemented such strategies for data collection of biological samples. Thus L19 in the abstract does not really reflect the content of the paper.

2/ P2-L45/46: this sentence calls for references

3/ P3-L64/65: maybe replace "dark state" by "triplet state": as is, it looks like the authors are not sure of the nature of the state, whereas their photophysical model clearly identifies the triplet state.

4/ P3-L75: the authors might be interested to read: Adam et al, <https://doi.org/10.1002/cphc.202200192> It would be interesting to know whether they noticed potential signs of the off-state heterogeneity described in this paper, in view of the ~kHz switching rates that they apply (e.g. non-mono exponential switching kinetics).

5/ P4-L96/97: there seems to be a contradiction between this statement and all the discussion in supplementary note 9 on the different RSFPs tested. In this note, unless I am mistaken, the difference in switching fatigue between the variants is not assigned to 405 nm induced photobleaching, rather to a balance between off switching and ISC induced by 488 nm light. Please explain and improve consistency of the text in this regard.

6/ P4-L122/125: increasing the dark time will not prevent T1-O2 interactions, resulting in ROS production. There are definitely several photobleaching mechanisms at play, please see Duan et al, DOI: /10.1021/ja406860e.

7/ P5-L136: "we investigated how the modulation of the on-switching dose, given a fixed total energy, influences the fluorescence output": this sentence does not make sense, the authors tend to make a confusion between power density and dose throughout the manuscript. Dose = total energy.

8/ P7-L195/201 (and Table S2): is it reasonable to assume that the quantum yield for photobleaching from the excited triplet is wavelength independent? As different levels of excited triplet states might be reached as a function of illumination wavelength, the fate could be very different, for example with blue shifted light generating more photobleaching and red shifted light generating more RISC. This is really an open question but the issue should be discussed, and at least the assumptions made clearly stated. A related question is: is it reasonable to assume a much weaker quantum yield for photobleaching from the excited triplet state than from the ground triplet state (supplementary table 2)? It would be expected indeed that the excited triplet state be more reactive than the ground triplet state.

9/ P9-L276/277: why only dim samples?

10/ P9-L282: "extends the absolute observation window": by how much? Also around 15%? Please discuss this point.

11/ P10-L294/295: it is not clear to me why co-illumination should only be applied during the readout step. Why not during

the switch off step, ideally using a doughnut-shaped red beam ? Isn't it during this step that most photobleaching occurs ?

12/ P10-L312/313: in fact, a protective effect could be expected, as shown in ref 29. Could such a decrease in phototoxicity be observed by looking at mitochondria mobility ?

13/ P10-L318/321: yes but no experimental demonstration was provided (see point 1 above). The downside of this strategy should also be mentioned (increase in data collection time).

14/ P11-L326: again, "a dark state" rather than the triplet state is invoked here (see point 3 above). This really reads bizarre, as the triplet state is clearly identified throughout the results section.

15/ P11-L346: the authors should recapitulate the benefits of using either visible red light or near infrared light. Have they tried to implement 900 nm illumination on their RESOLFT microscope ? This would be a very nice experiment to do, to check the pros and cons.

16/ The authors have developed the nice rsFusionRed proteins. Readers will naturally wonder whether they have assessed the potential photofatigue enhancement of those red rsFPs with near-IR light. I encourage the authors to answer this question.

17/ It would be nice in the discussion to consider the potential effect of the level of oxygen on the observed light induced recovery. Less oxygen is typically beneficial for photostability because of decreased ROS production, but also increases the triplet state lifetime (this was shown also in FPs). Under anaerobic conditions do the authors think that RISC by red shifted light will still help ?

18/ As RISC by red shifted light increases the singlet state population, not only photostability but also effective brightness should increase. Does this show up in the model and have the authors noticed this experimentally ? Discussing this point would be important.

19/ Using photofatigue as a distinguishing feature for multiplexing is a nice idea, but it should be better explained in what situation it is beneficial to use this technique rather than exNEEMO or TMI. Is this to collect the data faster ? What are the potential consequences in terms of photo/cytotoxicity ? Please discuss these issues.

20/ The provided software appears to work but is based on Python scripts without any manual. It will be very difficult to use unless such manual is made available. Ideally a Python GUI would be desirable to facilitate the use of the software. I am curious to know whether the authors considered using the SMS simulator (DOI: Bourgeois, 10.1038/s42003-023-04432-x) as an alternative to their software (and then the authors will not have much doubt about the identity of the reviewer ...) ?

21/ There are quite a few typos within the manuscript that should be checked ...

22/ Figure 2f is very noisy: please perform a statistical test (ANOVA) to check that the fitted decay is significant.

23/ Figure 2h legend: I cannot see any green shaded area.

24/ Figure 3A legend: fluorescence loss after how many cycles ? Why was the chosen off switching 488 nm dose different for each RSFP, while during a real demixing experiment, all RSFPs will be subjected to the same dose. Please explain. "Different multi-foci" -> "different foci" ? Note that some readers may confuse these "foci" with different "focal planes" and have a hard time to realize that they are all in the xy plane.

25/ Figure 4b, lower right panel: why showing the mix with and without 592 nm illumination ? It would be more informative to show the two RSFPs with co-illumination.

26/ Supplementary table 4: bleaching yields are in % ? Please check consistency with table S2.

27/ Supplementary figure 7b: what is the unit on the Y scale ?

28/ Supplementary table 6: bleaching quantum yields from ground and excited triplet seem inverted relative to those in table S2 and S4 ?

29/ I am not sure of some of the arguments presented in supplementary note 9: (i) that photobleaching by slower switching RSFPs is expected to be less recoverable by the RISC effect. If there is more triplet formed relative to off switching, the benefit of RISC should be higher, not lower ? Furthermore, non-switchable Green FPs have been shown to be nicely stabilized by RISC (ref 29); (ii) that the ratio between bleaching from the ground or excited triplet state is the key determinant: the ratio between RISC and photobleaching upon photon absorption by the triplet state is also a key determinant.

30/ supplementary figure 16: do these data support the notion that the long-lived dark state downstream the triplet is possibly also bleached by 405 nm or 488 nm light ?

(Remarks on code availability)

Below I paste point 20/ in my review:

The provided software appears to work but is based on Python scripts without any manual. It will be very difficult to use unless such manual is made available. Ideally a Python GUI would be desirable to facilitate the use of the software.

In conclusion for this part, at this stage I consider that the code is not a usable resource for the community.

Version 1:

Reviewer comments:

Reviewer #1

(Remarks to the Author)

Review of NCOMMS-25-05441A, "All-Optical Strategies to Minimize Photo-Bleaching in Reversibly Switchable Fluorescent Proteins," by Marin-Aguilera et al.

I thank the authors for their generally positive responses to my prior comments, which I feel have improved this interesting paper substantially. In particular, I applaud their approach of creating a generalized model that does not rely heavily on literature parameters. This model in general makes the paper considerably more convincing.

There is one major issue that I think still needs more work. First, as to whether the RISC drives the system to S₀ or a higher singlet state, I either misread reference 29 of the original paper or gave the wrong reference number. Rob Dickson's group has clearly found molecules for which RISC to an excited singlet state is the dominant pathway. That being said, in the work of his that the authors cite to substantiate this effect in fluorescent proteins, the protein that was used was specifically engineered to have this behavior, so I do not find this argument to be compelling. I am further concerned that the model used cannot differentiate between RISC to S₀ and a higher singlet state. These two processes must have different outcomes in the scheme of the authors. The fluorescence quantum yield of this protein is 0.3, according to Grotjohann et al. (DOI 10.7554/eLife.00248). Of course, some of the other 70% comes from nonradiative relaxation, but presumably a substantial fraction arises from ISC. If RISC repopulates the ground state, then no additional fluorescence can be generated until the next excitation pulse. If RISC populates a higher singlet state, then some population will be cycled back to the triplet state, where it can again have a chance to undergo RISC while the redder light pulse is on. I would therefore expect that the amount of fluorescence observed would be substantially higher for RISC to an excited singlet state than for RISC to the ground state. This picture supports what the authors propose, I think, and could be used as an argument for it. But the fact that their model does not indicate that the two different pathways could be distinguished is worrisome. It could simply be a related to their range of parameters. Still, the authors should at very least test some kinetic parameters that ought to show a difference and make sure that a difference is seen. For instance, with a 5% quantum yield for fluorescence and a 95% quantum yield for ISC, RISC to a higher singlet state should give far greater total fluorescence than should RISC to the ground state.

Some more minor points:

1. The authors continue to make errors of nomenclature that will potentially cause readers to doubt other aspects of the work unnecessarily. For instance, they still call 405-nm light "ultraviolet." This designation is an improvement over UV-vis, but 405-nm light is violet (or indigo, if one wishes). The UV spectrum starts at 400 nm. This designation should be fixed. They have also continued to misuse the term "red-shifted." I agree with them that this term is used often in spectroscopy, but it is used primarily to describe spectra, and secondarily to describe tunable lasers, and never to describe lasers with different wavelengths. It is not synonymous with having a longer wavelength. At a recent conference I queried a half dozen spectroscopists, in a neutral manner, regarding whether they would call a laser with a longer wavelength than another "red-shifted." The unanimous answer was no, and I got several puzzled looks for asking this question. I strongly recommend that the authors change this language. So the use in line 51 of the paper, for instance, is fine, but most of the other uses are not.

2. There are still numerous unclear antecedents in the paper that make for more difficult reading:

Line 29: "them," should be "RFSPs"

Line 55: "it" should be something like "this technique"

Line 58: "the one" do you mean "those?"

Some other small things I found as I was reading the manuscript:

3. Line 56: Please use something more accurate than "red-induced photobleaching"

4. The word "fluorescent" is used often when the correct word is "fluorescence." There is no such thing as a "fluorescent photon," for instance. Instead of "fluorescent signal," "fluorescence" is more accurate.

5. Line 78: "a decrease of 80%" in what?

6. Line 83: "fatigue fraction" is not defined

7. Line 84: "488 nm" should be "488-nm illumination."
8. Line 86: "405 nm light" should just be "405 nm"
9. Line 91: "crystallography" should be "crystallographic"
10. Line 115: "low" has no quantitative meaning, just say "for 405-nm power densities below 300 mW/cm²"
11. Line 127: "RFSP" not "RFSPs"
12. Line 128: What is "it?"
13. Line 134: "trapped in TH" not "trapped in the TH"
14. Line 136: Checking whether delivering the same dose over different times is known as a reciprocity study. What you are observing is a form of what is known as low-intensity reciprocity failure (LIRF), which is a clear sign of nonlinearity in a chemical system. I mention this in case you want to cite some relevant literature, but that is up to you.
15. Line 137: "delivered by progressively" not "delivered progressively"
16. Line 145: This sentence is hard to read. I'd recommend starting with "Reverse intersystem..." and ending with "by preventing photobleaching"
17. Line 149: "compared" not "respect"
18. Line 151: "in" rather than "on"
19. Line 153: Remove "rate" because you later in the sentence talk about a timescale, which is the inverse of a rate
20. Line 163: "increasing by up to" not "increasing up to"
21. Line 179: Laser lines by definition come from the same laser; for instance, an argon ion laser has several different lines. Use "all the different laser wavelengths explored" instead of all the different laser lines of the experiment"
22. Line 180: There is no such thing as the ground state of a triplet unless the ground state of the molecule is a triplet (molecular oxygen, for instance). Say something like "both T1 and higher triplet states contribute"
23. Line 200: "reaches a plateau at 488 nm of more than 600"
24. Line 201: "power densities of more than 10"
25. Line 215: "This" what?
26. Line 217: Doesn't this sentence say the same thing twice? Strong absorption promotes excitation? I would remove "promotes triplet-state excitation and" and replace "strong triplet absorption" with "strong absorption of the T1 state" for a little bit of additional clarity
27. Line 226: What is "they?"
28. Line 230: "illumination irradiance" is redundant. The adjective "high" goes better with irradiance, but if you want to use illumination couple it with "strong"
29. Line 239: Define FOV as field of view, i.e. "over an extended field of view (FOV)"
30. Line 248: What is "their?"
31. Line 249: "This" what?
32. Line 256: Replace "such" with "this"
33. Line 260: "criterion" not "criteria"
34. Line 265: What is a photobleaching constant? If this was defined I missed it.
35. Line 267: I do not know what "Attaining these results" means
36. Line 272: Remove "a timelapse of"
37. Line 274: "extends the absolute observation window for the multiplexing imaging by 20% to 30%"

38. Line 281: "and" should be "or" and there is no need for "the" before "photoswitching"
39. Line 287: Remove "an"
40. Line 294: "this" what?
41. Line 300: Delete "looking"
42. Line 307: "rationalizing our observations with an effective"
43. Line 315: "became" should be "becomes"
44. Line 318: What is "it?"
45. Line 322: Same question
46. Line 327: This is not a sentence
47. Line 330: Remove "seemingly," you are not saying its conserved among all of them necessarily
48. Line 333: "red light" not "red-light"
49. Line 336: I'd suggest starting with "There is a trade-off between spatiotemporal resolution and photostability in any live-cell-imaging application"
50. Line 337: What is "they?"
51. Line 341: "wavelength for the excitation of orange- and red-emitting fluorophores" and use "this wavelength" instead of "it"
52. Line 351: You've already defined PAA, and use it in the line above to boot. Just say "in PAA gel"
53. Line 355: "#" not "No"
54. Line 358: "characterization" not "a characterization" and "off-and-on-switching" or "off- and on-switching"
55. Line 360: Use × and appropriate spacing, as you do elsewhere in the manuscript
56. Line 371: Remove "accordingly"
57. Line 375: "with a 1-ms 405-nm"
58. Line 398: What does "1.40–0.70-NA" mean? Please elaborate.
59. Line 399: What does "modulable" mean? Please elaborate. Intensity? With time? Please clarify every time you use this word.
60. Line 400: Cobolt is a model, please acknowledge Hübner Photonics as you do for other manufacturers here and throughout the rest of the paper
61. Line 414: "in which" rather than "where"
62. Line 416: Use ×
63. Line 422: What is "MPD?" A manufacturer? Please clarify as for other pieces of equipment.
64. Line 426: "gold-bead" not "gold beads"
65. Line 432: "at a 80 MHz"
66. Line 442: What is "it?" and at very least use "its" rather than "it's"
67. Line 445: "both 810 and 900 nm"
68. Line 446: Use ×
69. Line 449: "two APDs"

70. Line 453: "into" rather than "on"

71. Line 485: "resolution"

72. Line 486: "was" rather than "were"

73. Line 490: "confined" rather than "confinement"

74. Line 491: "from the" rather than "of the"

75. Line 492: "to split the light equally into P and S polarizations"

76. Line 494: "SiO₂"

77. Line 495: What begins at the end of this line is not a sentence. Start with "The ± 1 diffraction orders"

78. Line 519: "slow image, which consisted"

79. Line 520: "the other, which we named fast image, consisted"

80. Line 546: "between adjacent foci"

81. Line 557: GMM was already defined and delete "that fits the data into a given number of Gaussian components, which you already stated."

82. Line 559: "proteins. Afterwards" and replace "Gaussian mixture model" with "GMM"

Figure 1: As mentioned above, "fatigue fraction" is never really explained adequately, and the right side of panel (a) does not help. "fatigue fraction" is pretty well hidden in the color legend. Why doesn't the color legend start at zero?

Figure 2: Right side of panel (a), normalized to what? In panel (g), the x-axis numbers do not line up with the ticks. What does it mean to have an error bar that goes below zero? Panel (j), how are the dashed lines exponential fits???

Figure 3: Last line of caption, "when the 592-nm pulse follows 1 ms after the 488-nm pulse"

Comments on the SI

I admittedly did not read this as carefully as the main paper. The font in the supporting information is quite small. I recognize that the document is already quite substantial in length, but it would be helpful to readers to use a larger font and smaller horizontal margins nonetheless.

Page 8, paragraph 4: What is a "raising time?" The dashed line is clearly not what would normally be called a stretched exponential, which is a decaying function. Please elaborate. This issue probably relates to my question on Figure 2. Also, please report the value of τ and give a physical interpretation if possible.

Page 10, Figure 6: Why don't the y-axes start at zero?

Page 11, end of first paragraph: "since this phenomenon" rather than "Since it"

Page 13, Figure 9: Please use broken scales for the y axes

Page 14, Figure 10b: Can you spread the y scale on the residuals a bit so the numbers don't overlap?

Page 16, last paragraph, first sentence: "Complementarily"

Page 18, last paragraph: Maybe just clarify slightly "The quantum yield for RISC from a given higher triplet state in a specific solvent should be conserved...."

Page 19: Same comment about the residuals in both figures.

Page 24, Figure 10: Do you really observe negative recovery at low NIR intensities?

Page 25, Figure 20: in (a) Normalized to what? Just one statement somewhere in the paper will suffice. In (b), same question regarding negative recoveries

Page 27: I don't understand the last sentence of the second paragraph

Page 33: It's probably not necessary to give all focal lengths and part numbers

My apologies for the length of this review. Most of the points raised are minor, but are still worth addressing. I do want to see the results of the checks of the kinetic modeling, but overall I like this paper quite a lot and think that it is near being ready to publish.

(Remarks on code availability)

Reviewer #2

(Remarks to the Author)

The authors have performed extensive revisions of their paper, which answer many of my previous concerns. The manuscript still largely focuses on the photophysical model that underlie the nice experimental results. I think it is adequate to substantiate these results by a restricted, meaningful photophysical model of rsEGFP2. The very extensive discussion on the photophysics is particularly interesting for photo-physicists but is somewhat overwhelming the main message of the paper about light induced recovery through RISC.

There are three major points that I believe should be addressed before the manuscript can be published, as well as a number of more minor points that I encourage the authors to consider.

Major points:

1/ Although I understand the authors viewpoint that RSFPs should be submitted to illumination schemes matching their photophysics and suitable for imaging experiments, I feel that, from the viewpoint of fundamental photophysics, the paper is still misleading: while a constant dose of 488 nm light is used, a varying dose of 405 nm light is employed. It is then difficult to compare the effect of each laser to the photofatigue decay. One main message of the paper, coming early in the manuscript, is that most of the damage is caused by the 405 nm laser (Fig 1b), while the 488 nm laser is relatively immune to photobleaching (Fig 1c). From the viewpoint of the photophysicist, the comparison is unfair, as the 488 nm laser is at constant dose, which is not the case for the 405 nm laser. Furthermore, the other main message of the paper is that RISC can be used to substantially slow down photofatigue, while the triplet state is always reached, according to the presented model, from the 488 nm laser. Thus, there is objectively very substantial photobleaching by 488 nm light, and there is an apparent contradiction. These aspects could really confuse the readers. The lack of clarity extends for example to the comparison of RSFPs with normal FPs: I think RSFPs do not react differently than FPs regarding T1 and the RISC effect: increasing 488 nm power density at constant dose with EGFP is very likely to produce the same increase in RISC efficiency, and there is no real opposite behavior between the findings here and in ref 32.

All this is mostly a question of presentation. I would recommend the authors to (i) precisely state what doses / intensities are used for each laser (at present, there is confusion at many places in the paper, e.g. in the legend of Fig 1), (ii) make sure not only the super-resolution imaging community, but also the photophysics community perceive the consistency of the paper, (iii) clarify the different sources of photobleaching (405 versus 488 nm light), based on the use photophysical model, possibly with the help of an additional figure in the main.

2/ Restricting photobleaching induced by 405 nm light to the TH species sounds inconsistent. This is because, as the 405 nm dose is increased, it is more and more the CHint species that gets populated, as can be seen on figure S3b. Since CHint strongly absorbs 405 nm light, most 405 nm photons absorbed by the (TH, CH+, CHint) trio are increasingly due to CHint absorption. In addition, if CHint is indeed fluorescent - see my next major point -, its excited state lifetime might be greater than that of TH, possibly opening the door to triplet state excursion from this state (although it is unclear whether CHint would be fluorescent under 405 nm excitation). Overall, I think photobleaching from CHint vs TH must be discussed, may be reworked, and at least the choice to only consider bleaching from TH must be carefully justified. Perhaps considering direct photobleaching from CHint in addition to from TH would not change significantly the outcome of the simulations.

3/ In evaluating the fluorescence of the same intermediate CHint (supplementary note 3), the authors ignore the effect of triplet state built up and relaxation induced by the initial 488 nm pulse. This relaxation results in a significant rise of fluorescence with a timescale corresponding to the triplet state lifetime, as detailed in supplementary note 7. The authors cannot ignore triplet state relaxation in supplementary note 3. I urge the authors to revisit the notion that CHint is significantly fluorescent. If the conclusion is maintained, and the extinction coefficient of this species at 488 nm is still high, then it must be that CHint exists as an equilibrium between an anionic and a protonated form. As this is a new finding as compared to published literature, some discussion should be added about the nature of this intermediate - which can probably not be called "CHint" -

Minor points:

L37: radii

L49-51: awkward statements: "which presents a peak around 488 nm, the same wavelength used to elicit the fluorescent photon" is in contradiction with "Given that the long-lived triplet state of FPs has a red-shifted absorption spectrum compared

to the fluorescent form”

L61: a “comprehensive understanding”: this statement is in contrast with the notion that the authors use a “minimal model”.

L80: “fluorescence signal” rather than “fluorescent signal” (this comes at several places in the manuscript)

L83/84: “Similarly, the fatigue fraction also increased with higher 405 nm illumination, while showing a minor dependency on 488 nm.” In relation to my major point 1, this sentence is misleading.

L104/105: It looks like the identified intermediate is new relative to literature. The sentence reads quite mysterious at this early stage in the manuscript. I think the point should come later, and deserves more comments in the main manuscript if fluorescence of CHint is confirmed.

L143/144: it would be fair to state right away that the penalty is an increase in the data collection time.

L163: “suggesting that the effect is driven by an absorption-mediated process”: what else could it be ? This reads like a trivial sentence.

L165: 405 nm ?

L208: to me, another possible scenario is that red-shifted wavelengths promote RISC more than photobleaching, as compared to stronger wavelengths. Naïvely speaking, lower energy photons are more gentle. (But I understand that Reviewer 1 has arguments against this possibility)

L216-219: Here the authors write in clear that 488 nm light illumination is a major cause of photobleaching through ISC. After looking at Fig 1c, readers might get lost.

L284/285: I do not understand why the majority of the triplet state population would reside in the center of each focal spot. The ratio ISC vs switching is the same everywhere, so the donut beam will also generate substantial triplet state ?

L302/303: I encourage the authors to comment about ROS generation and the possibility/absence of reduced phototoxicity upon 592 nm co-illumination, as observed in ref 32.

L327/328: please check grammar of this sentence.

L342/343: perhaps specify that special optics compatible with near-IR are then needed.

Supplementary materials:

Table S2: the on to off switching yield of rsEGFP2 from ref 6 is obsolete ... Please use the data from Adam et al, ChemPhysChem 2022. The lower switching yield in that reference means a somewhat reduced triplet state shelving effect.

Supplementary note 3: to check the effect of T1 relaxation, I would recommend repeating the experiment with no 405, as a control.

Supplementary figures 6: the red crosses are not visible

Supplementary note 4: “The addition of a photobleaching pathway was necessary to reproduce the 488 nm power density” - > 405 nm, not 488.

Supplementary note 5: the very long fitted lifetime of a putative radical state (17 s) is surprising to me. I wonder if this can result from the incomplete/nonideal photophysical scheme. In general radical states in fluorescent proteins are relatively short-lived under strong illumination, typically tens of milliseconds. Ref 25 does not report a super long lifetime radical species. Furthermore, the off switching of rsEGFP2 is always seen biexponential, which is assigned to transient excursions to a short-lived dark state (in comparison to the lifetime of the off-switched state). Finally, radical states are typically light-sensitive, so their lifetime can be shortened a lot under illumination. Please discuss the obtained results in view of this literature knowledge.

Supplementary table 6: why is the amplitude for the first phase negative ? If two phases are seen, could this be a sign of the relaxing radical with a rate somewhat slower than triplet state relaxation ? But again, here relaxation is in the dark, and radicals would relax fast maybe only under light.

Supplementary note 7: the strongly varying photobleaching yields fitted in different experiments are a bit frightening. Although FPs are definitely sensitive to their environment, a change by a factor of 10 in different (environment) replicates sounds really odd. Please comment.

(Remarks on code availability)

Version 2:

Reviewer comments:

Reviewer #1

(Remarks to the Author)

The authors have done an excellent job of responding to my comments, and I believe that this paper is ready for publication in this journal.

(Remarks on code availability)

Reviewer #2

(Remarks to the Author)

The authors have performed another round of extensive revisions, answering positively all my concerns. I congratulate the authors for a very thorough and interesting work. I have two minor suggestions, but I let the authors and the editor decide about their relevance.

1/ The first part of the results will probably be hard for the readers to immediately digest. Perhaps the authors could simplify the flow, and move some parts to the discussion ?

2/ The data now convincingly suggest that the intermediate Cint is indeed fluorescent. In fact, this is interesting: it could be that Cint is very close to the on-state, but with a different pKa due to a modification in the chromophore environment, e.g. due to hydrogen bonding changes. A very similar finding was published recently, albeit on a photoconvertible anthozoan FP (J. Am. Chem. Soc. 2025, 147, 12, 10357–10368). The authors might want to point to such an hypothesis.

(Remarks on code availability)

**Point-by-point responses to reviewers of the manuscript NCOMMS-25-05441:
All-Optical Strategies to Minimize Photo-Bleaching in Reversibly Switchable Fluorescent
Proteins**

Marin-Aguilera et al

We appreciate the comments and the suggestions of the Reviewers. All the inputs helped to improve both the structure and clarity of the manuscript.

In the revised version we integrated the comments of both reviewers to strengthen and clarify the model. This brought us to uniform the model with previous literature on triplet and fast photoswitching studies of rsEGFP2. Through simulation and experimental work, we have validated the drawn photocycle for rsEGFP2 as well as tested its expansion to incorporate additional states and the different pathways highlighted by the reviewers. In the text we have now clarified the limit and strength of the model, that is intended to be an effective model rather than a comprehensive theoretical description.

Below we provide a point-by-point response to the comments of each Reviewers. The comments of the Reviewers are written in blue and our responses in black.

REVIEWER COMMENTS

Reviewer #1 (Remarks to the Author):

Review of NCOMMS-25-05441, “All-Optical Strategies to Minimize Photo-Bleaching in Reversibly Switchable Fluorescent Proteins,” by Marin-Aguilera et al.

This paper develops optimized methods for using light to remove photoswitchable fluorescent proteins from undesirable triplet states that trap population for long time periods. This strategy not only makes the proteins less likely to photobleach, but also makes these molecules available to fluoresce more often. The experimental work in this paper is both thorough and compelling, and the work will be of great value to the community of researchers who use these proteins. The authors demonstrate that this strategy works for other green-emitting fluorescent proteins, and for a number of imaging modalities. I feel that this paper will have a high impact among those in the advanced optical imaging community, and is ultimately worthy of publication in Nature Communications.

We appreciate the reviewer’s positive evaluation of our work.

Since the concerns raised all pertain to the model, it seems best to first clarify its intended role within our study. Based on our experimental findings, we constructed a minimal, effective kinetic model to provide a mechanistic explanation. In bleaching photophysics, kinetic models are often simplified, because fluorophore bleaching typically arises from complex, oxidative reactions involving many environmental factors and chemical species. Adding too many free parameters would weaken the model’s predictive power; conversely, reporting only the experimental trends would leave readers without a framework for interpreting these inherently complex photophysical processes. Hence, we developed a minimal kinetic model: we began with the established core-state scheme for rsEGFP2 and added only those extra states and transitions required to capture our experimental observations. We made simplifications and added effective pathways, and we thank the reviewer for helping us clarify our assumptions.

There is one crucial issue that I feel needs to be addressed first, as well as a range of minor issues. Please do not take the length of the list of the issues I have identified as a negative, I would simply like to see this work have as much impact as possible.

To begin with my major concern, the model that the authors present is far from convincing to me; in fact, I am pretty certain that it is flawed, for several reasons. First, along with virtually all of the prior literature in the field of photodeactivation of triplet states, this study makes the unfounded assumption that RISC solely leads to the population of an excited singlet state. In most circumstances, a more likely outcome is the population of a highly vibrationally excited level of the singlet ground state, because the vibrational density of states of S_0 at this energy is vastly higher than that of any singlet excited state (this behavior is challenging to prove, but it can be done; see, for instance, Ref. 28 in the paper and [dx.doi.org/10.1016/j.isci.2021.103600](https://doi.org/10.1016/j.isci.2021.103600); as this journal encourages reviewers to reveal their identities, I will say at this point that this publication is one of mine.). This process, which we in the field typically (and, I would argue, incorrectly) call internal conversion (IC) in the case of T_1 , explains why phosphorescence quantum yields are not typically more than a few tens of percents for organic molecules. Indeed, the typical way to attain a high phosphorescence quantum yield in such molecules is to have the T_1 energy be well below that of S_1 , which not only prevents RISC to S_1 , but also lowers the probability of IC. The vibrational density of states of S_0 grows exponentially with increasing energy, and so the higher the triplet state, the faster RISC to the ground state will be as compared to RISC to an electronic state with an origin of similar energy to the triplet state. That does not mean that no population will undergo RISC to excited singlet states, but in many, if not most, organic systems the branching ratio for RISC to excited singlet states is likely to be small. As an example, in Ref. 29 the authors had to invoke photobleaching from the triplet state that also causes repopulation of the singlet manifold to explain the fluorescence intensity does not depend linearly on the intensity of the deactivation wavelength, but rather saturates. This behavior is a natural consequence of RISC going to the S_0 state directly, as can be shown by trivial, analytical kinetic modeling.

We acknowledge the literature complexity of the triplet to singlet transition. In outlining the used model, we have built on the cited literature, nevertheless we took the reviewer's comment as an inspiration to validate and compare the different pathways in the context of our data as well as to better ground our decision. Our intention is to have a solid effective model that is able to reproduce the experimental data of photobleaching.

The extended spectroscopical studies recently reported on rsEGFP2 (among particularly insightful for this context is Rane et al. 2023, *J. Phys. Chem. B*) allow us to have an indication of where the triplet states are in the energy ladder (Byrdin et al., 2018, *J. Am. Chem. Soc*) or which are their absorption properties (Rane et al. 2023, *J. Phys. Chem. B*, Byrdin et al., 2018, *J. Am. Chem. Soc*) (Fig. R1). However, they lack

Figure R1. Schematic of energy ladder of singlet and triplet states for rsEGFP2 in eV. The position of S_1 was determined from the maximum in the emission spectra (Grotjohann et al., 2012, *eLife*). The position of T_1 was determined from the maximum in phosphorescence emission spectra of EGFP (Byrdin et al., 2018, *J. Am. Chem. Soc*) as an approximation of the energy level of the first triplet state of rsEGFP2. The energy levels of the subsequent excited triplet states were calculated as $E(\lambda) = 1.69 + 1.24/\lambda$, with λ in μm . The colored arrows indicate which wavelength would populate each of the energy levels: 3.06 eV, 900nm; 3.78 eV, 592 nm; 4.24 eV, 488 nm; 4.75 eV, 405 nm.

conclusive evidence for which is the precise mechanism that leads to the repopulation of the singlet form via RISC. In the literature, the effects of light-induced RISC in EGFP (Ludvíková et al., 2023, *Nat. Biotechnol.*) and rsEGFP2 (Byrdin, M., et al., 2025, *Biol. Cell.*) mainly assume that any conversion of population from the triplet to the singlet will occur via excited states. The existence of such pathway in fluorescent proteins has been proven and observed by means of optically-activated delayed fluorescence (OADF) (Peng, B., 2021, *J. Phys. Chem. B.*, Lu, Y-H., 2023, *J. Phys. Chem. B.*) where a fluorescent photons are measured after illuminating the sample with red-shifted light that is delayed in time with respect to fluorescence excitation. This phenomenon is possible if the RISC mechanism involves the excited states from the singlet manifold, but as the reviewer points out this mechanism might not be the dominant RISC pathways in typical organic dyes.

As the reviewer suggests, RISC could happen from excited triplet states directly into the vibronic manifold of S_0 followed by intramolecular vibrational relaxation (IVR), this would be supported by the fact that the density of vibronic states increases exponentially with the excitation energy above a state. We recognize that, since light-induced RISC events are rare compared to the overall fluorescence emissions in a typical fatigue experiment, the final influence of landing to S_1 or S_0 after a RISC/IVR should not produce appreciable change in our observable. If light-induced RISC populates S_1 , in a few nanoseconds the great majority of the events will lead anyway to S_0 , with a very small chance of ISC ($\Phi_{ISC} \approx 10^{-3}$).

We explored whether channelling RISC $T_n \rightarrow S_1$ or RISC $T_n \rightarrow S_0$ would have a sensible impact on the numerical results of the kinetic simulation of the fatigue experiments. This was verified by simulating the light-induced fatigue recovery in two identical simulation settings, with all the parameters as already described in the main text and Supporting Information Note 1. The only change in the two conditions is the connection of the transition from the excited triplet state to the singlet states. In one case T_n converts to S_1 (as in all the remaining simulations in the manuscript) and in the second case T_n converts to S_0 . The dashed arrows in Fig. R2 represents RISC/IVR, thus including the vibrational relaxation. There are no notable changes in the computed light-induced recovery.

Figure R2. Example of light-induced photoswitching fatigue recovery computed as a function of the 592 nm power density at a fixed 488 nm power density (300 W/cm^2). (a) Condition with RISC mechanism $T_n \rightarrow S_1$. (b) Condition with RISC mechanism $T_n \rightarrow S_0$. All the simulations were carried out at 1 ms 405 nm and 100 W/cm^2 power density.

In the main manuscript, we decided to keep the original RISC $T_n \rightarrow S_1$ mechanism, because we believe it is more consistent with the cited literature and does not produce relevant changes for the results of our work compared to RISC $T_n \rightarrow S_0$. Nevertheless, in order to make the reader aware of this discussion, we included the presented discussion and reference in Supplementary Note 8, properly referring to it in the main text.

Second, the model, at least as shown schematically, treats all T_n states as having identical behavior. Given what the authors observe, this is manifestly impossible. As mentioned above, the higher the triplet state, the more likely RISC is to occur (and 488-nm light was indeed shown in Ref. 28 to cause RISC to S_0 , which is missing from the authors' model), but at the same time it may also be more likely that some sort of photobleaching event occurs. There is absolutely no reason to believe that the branching ratio for these phenomena (not to mention relaxation to lower triplet states) should be the same following triplet absorption with two disparate colors of light. The triplet absorption spectra in Ref. 27 (which predates Ref. 28 and is more relevant to the work presented here) make it quite clear that the different wavelengths used here go to different excited triplet states.

As the reviewer rightfully points out, it is uncommon that different wavelengths share the same RISC efficiency. Indeed, the different wavelengths are parameterized separately in our model, but notably we account only for the different absorption cross-section as reported in Rane et al. *J. Phys. Chem. B* 2023, but they share the effective RISC quantum yield. Such effective quantum yield is retrieved by fitting of the data.

At this point it is important to consider the nature of our experimental data: the loss in fluorescence after consecutive on/off cycling. This is not information rich, as data coming from more conventional time resolved spectroscopy methods. Considering the limited predicting capability of such data, we have fixed most of the parameters from literature and helped where possible with *ad-hoc* experiments. The RISC parameters are instead directly fitted. Overall, we believe that a finer description of RISC would not add substantial mechanistic insight, which is the primary aim of our modeling. Our RISC pathways are to be intended as effective ones and could be refined in the future if additional experiments and work will be available regarding the details of RISC in rsEFGP2.

We made clear in the main text and Supporting Note 8 that RISC from different T_n states in general has different efficiencies, and that we are working with a minimal effective model still able to reproduce satisfactorily the data.

“Similarly to recent literature on light-induced RISC in EGFP³² and rsEGFP2⁴⁵, we rationalize the repopulation of the emissive state via RISC from the triplet excited state (T_n) to a vibronic state of the singlet manifold (Supplementary Note 1 and 8). All the different laser lines of the experiment (405, 488, 592, 915 nm) will excite the triplet state³¹, stimulating both bleaching and RISC.”

Third, although I appreciate that the authors did their best to use literature values for rate constants, I do not trust much of the literature based on the points raised above. Their model contains so many processes and rate constants that it could reproduce almost anything, even though the model is missing the processes discussed in the previous two paragraphs. Indeed, in my view their model is not compatible with the results in Ref. 28. Developing a model that incorporates verifiable photophysics will take a considerable amount of additional experimental work, and I would suggest leaving that for a future paper. Others will likely have a different opinion than mine, but I feel that the modeling work detracts from the paper because, in my opinion, it is at best incomplete, and more likely is inaccurate. Although some of the experiments performed may have been motivated by the model, presenting the interesting results while having those results be motivated by a model that will elicit skepticism in readers like me proves a distraction. I suspect that this opinion will (understandably) not be popular with the authors at first (and maybe afterwards as well), but I urge them to consider my suggestions strongly. It is better to get the modeling right (or to prove the current modeling to be correct) in a subsequent paper (which I feel will also have high impact) than to dilute the impact of impressive experimental work with a model that cannot currently be substantiated.

We understand the reviewer perspective and agree on the importance of grounding the kinetic model in verifiable photophysics. In the revised main manuscript, we refocused the kinetic model's role and drastically reduced its overall importance.

The model is now presented strictly as a minimal, descriptive framework that helps readers follow the experimental trends. We have flipped the order of some paragraphs in the first two sections, first we report the data and then we follow with the comparison with the kinetic model. Furthermore, all forward-looking or predictive sections have been moved to the Supplementary Information, meaning former Fig. 2h and its discussion is now moved to Supplementary Note 9 where we expand and clarify the limitations.

We agree with the reviewer that the experimental part of the manuscript could be self-consistent. At the same time we believe that without a simple kinetic model the reader would not understand in depth the experimental results. This is primarily because rsFPs have a complex photophysical switching behavior, already without considering bleaching pathways and triplet states. A simplified effective kinetic model, although with limitations and simplifications, is a great tool for the reader to navigate the experimental results.

We have now first stated the intention of the model at the very beginning of the manuscript:

“Using this information, we built a minimal effective kinetic model for rsEGFP2 that could aid the mechanistic explanation of the observed photoswitching fatigue behavior (Supplementary Note 1)”

More clearly framed the simplification and assumption of it, this reflects both throughout the main text and in the extended and revised Supplementary Information. Also following the comments of Reviewer #2, directed to the model, we now have added the following Supplementary Note related to the model:

- Supplementary Note 5. Thermal relaxation of long lived dark in rsEGFP2
- Supplementary Note 8. Higher excited triplet state and risk mechanisms.

Finally, we conclude putting in prospective future expansions of the model to bring it outside of the live cell imaging context.

“Our results showed that a deeper understanding of the species involved in the photocycle of RSFPs as well as the mechanisms governing photobleaching can lead to longer and higher quality live-cell imaging experiments. Further expansion of the model, for example, by incorporating additional long-lived dark states (e.g., radicals) or extended pH equilibria, would generalize it to a wider range of experimental schemes beyond live cell imaging.”

We believe these changes keep the valuable mechanistic guidance the kinetic model provides, while eliminating the risk of over-interpretation, without weakening the experimental narrative.

Other, more minor points that need to be addressed, in no particular order of appearance or degree of concern:

1. The authors play fast and loose with the optical spectrum. 405-nm light is violet; there is no such color as “UV-vis.” 592-nm light is orange (and barely so, it’s almost yellow), not red. Calling light “red-shifted” is confusing, and makes it sound as if the authors are looking at a side-band of another laser they used. Please give this light a clear name, such as “light at the triplet deactivation wavelength.”

We understand the confusion that the term “red-shifted” may have caused to the reviewer. We clarified in the main text the precise meaning of the term, i.e. light of a wavelength that is shifted towards the red side of the visible spectrum compared to the primary fluorescence excitation wavelength. We also remark that “red-shifted” is a commonly used term in spectroscopic literature, that does not necessary mean that the laser light we used is of color red, but that is shifted to longer wavelengths or lower energies. In general, we have improved the main text and tried to avoid confusion regarding the term.

2. The authors use a lot of vague language in the paper, for instance talking about “several” rounds of photoswitching, and using many unclear antecedents (“it” and “this”) that made the paper hard to read for

me in some passages because I had to go back and figure out the antecedents. As I teach my students, this problem can be fixed by avoiding the use of “it” to refer to something prior, and to always follow “this” at the start of a sentence with a noun.

We improved the main manuscript by reducing the instances of antecedents. We agree that in this way the readability of the text is improved.

3. Many of the figure panels were difficult for me to parse (e.g., Figure 1d, Figure 1e, Figure 2a, Figure 2h, Figure 2 i,j). Many of the graphs are deceptive, with y-axes not starting at zero, thus exaggerating small effects. At very least, broken scales should be used in such cases. In Figure 2a in the right panel, please add a second x-axis on top that gives the intensity. 90 mJ/cm^2 (note that the 2 is not an exponent in the figure) is not an energy. To say that the model reproduces the data in Figures 2e,f is really a stretch. Within your uncertainty, the data in Figure 2f have no dependence on τ whatsoever. I also cannot read whatever is in the color bars for the heat maps in Figures 2i,j.

We have updated the mentioned figures to enhance their clarity.

- Broken axes have been added to more clearly reflect the magnitude of the change.
- Figure 2a now reports both illumination time and power density of 405 nm, and the exponent has been fixed.
- For old Figure 2f (now Figure 2g) we collected a new dataset. The updates in the model and finer sampling defined a better accordance between experimental data and simulation. Previously, the 488 nm illumination time (0.9 ms) was shorter than the 592 nm pulse (1 ms) while now both 488 nm and 592 nm pulses are 1 ms. With these experimental parameters, we can now better reproduce the data trends, especially the increase in light-induced recovery observed between 0 ms delay and 1 ms delay.
- The labels on the color map are now in black for clarity or outside the bar.

4. There are places in the paper where the authors refer to a spectrum without saying type of spectrum. An example is line 51, “...the...triplet state of PPs has a red-shifted spectrum...” I presume that they mean absorption rather than phosphorescence, but they need to be clear. Another example is in line 180.

We have now corrected the mistake in the paper.

5. There are quite a few places in which numbers that should be exponents are not.

We have now addressed it.

6. In the first sentence of the paper, the abbreviation FP is used without definition. There are other similar problems throughout the manuscript (e.g., EGFP, SNR).

We have updated accordingly.

7. In the second paragraph of the results section, it would be quite helpful if the authors explained what they mean by an “excitation cycle.” The meaning only becomes clear later on in the paper.

We have rephrased the sentence to solve its ambiguity; the text is now:

We homogeneously illuminated a region ($20 \times 20 \mu\text{m}^2$) of a thin layer of rsEGFP2 embedded in polyacrylamide (PAA) gel with consecutive on-and-off photoswitching cycles. The emitted fluorescent signal during each off-switching phase was recorded until a decrease of 80%, mimicking the imaging

conditions, and the photoswitching fatigue curve was built by repeating the photocycle hundreds of times (Fig. 1a).

8. There are many “gratuitous” hyphens following words that start with “photo.” For instance, the correct word is “photophysics,” not “photo-physics.” “Infrared” also does not have a hyphen.

We have made corrections throughout the text.

9. “Data” is plural, not singular, and Nature correctly follows that usage.

We corrected to the plural form of “data”.

10. Many people who might use the results presented here in their own work have commercial fluorescence microscopes that are not capable of simultaneous exposure at two different wavelengths and are not able to do the sort of multiplexing that the authors demonstrate. Are the authors able to test how much improvement can be achieved if only alternating scans of different colors can be used? If this idea is not easy to pursue, that is not crucial to the publication of this work.

Our results indicate that the fatigue-resistance benefit of >592 nm co-illumination remains apparent in a sequential scheme, provided the interval between the 405 nm + 488 nm excitation and the >592 nm recovery pulse is under 2 ms. In a scanning microscope, this could be implemented by interleaving the 592 nm illumination with each line or pixel scan, or by reducing the field of view to shorten the overall frame-acquisition time.

To end on a positive note, this work is quite impressive, and I hope to see the paper published in this journal. Please know that all of my comments are meant in an entirely constructive manner; the topic of your work is highly complex, and many of the issues that I have brought up are not well understood or appreciated in the community. I would like to see you publish this work in a manner that is not only exciting, but also “bullet-proof.”

Reviewer #2 (Remarks to the Author):

In this manuscript, Marín-Aguilera et al provide an in-depth study of photobleaching mechanisms in reversibly switchable fluorescent proteins (RSFPs) and introduce means of reducing photobleaching under conditions typical of RESOLFT super-resolution microscopy by the addition of red-shifted illumination during switching cycles or possibly dark periods between switching cycles. The paper is a natural follow-up of the recent work of the group of Agathe Espagne (ref 29) who showed that near infrared light is able to induce reversed intersystem crossing from the triplet state of standard green fluorescent proteins such as EGFP, shortcutting photobleaching pathways typically arising downstream this state. Extending the work to the case of RSFPs is important due to the numerous applications that these markers offer in advanced microscopy approaches.

The paper is divided into three sections: (i) a thorough experimental investigation of photofatigue decay of purified polyacrylamide-embedded rsEGFP2 as a function of illumination conditions, (ii) an experimental investigation of the level of recovery provided by red-shifted illumination, reaching up to ~15%, and finally (iii) practical realization in the context of biological samples, showing a moderate but significant effect for time-lapse experiments using either parallelized confocal or RESOLFT microscopy.

In addition, the authors introduce a novel technique for kinetic-based RSFP multiplexing based on differential photofatigue decays between different variants.

All the work is embedded in a conceptual framework based on an elaborated photophysical model of rsEGFP2, allowing the authors, through extensive simulations, to account for the experimental observations made and possibly predict optimized behavior of RSFPs in a variety of imaging experiments.

Overall, this represents a large body of interesting and timely work, clearly to the benefit of photo physicists and fluorescent protein developers, and potentially to the benefit of the large community employing RSFPs as labels in their biological experiments. The photophysical aspects are treated in the specific context of intense illuminations, typical of RESOLFT-like experiments, which is rarely addressed in the rich literature concerned with RSFPs photophysics. This makes the paper particularly interesting to photophysicists. Thus, in principle I am supportive of this work. However, I have many interrogations concerning the validity of the proposed model. To my opinion many issues remain to be solved, that may either stem from a certain lack of clarity (the paper is highly complex to read and properly digest) or be related to more serious inconsistencies. All these issues call for a major revision of the manuscript.

We thank the reviewer for the positive comments and the thorough revision that made us clarify and expand the investigation.

Major issues concerning the proposed photophysical model of Figure 1e and S1:

1/ In principle the cis anionic and the cis neutral chromophore should be in rapid equilibrium, depending on pH. This equilibrium is not represented in Figure 1e, but should be, as the cis neutral chromophore, even if it is not highly populated at physiological pH, is able to absorb 405 nm light, isomerize to TH, and probably photobleach under such illumination. I guess that in Figure 1e, CH is meant to represent the “cis on like neutral” chromophore of ref 16, not the real “CH”. If this is correct, the “equilibrium” cis neutral chromophore is missing in the model.

We apologized for the confusion on the nomenclature. We have now clarified the terminology of the states involved in photoswitching of rsEGFP2 in Supplementary Note 1 and throughout the manuscript. We have aligned the terminology to the one of Uriarte, et al. (*J. Chem. Phys. Lett*, 2022), CH^{\ddagger} in our scheme is a transient state corresponding to “Cis On-like neutral”. Fig. R3 shows the states of the model mapped over *Scheme 2* of Uriarte et al. *J Phys Chem Lett* 2022.

Figure R3. Scheme of the proposed off-to-on photoswitching mechanism of rsEGFP2, adapted from *Scheme 2* of Uriarte et al. *J Phys Chem Lett* 2022. On the top we clarify the labels implemented in the revised work.

CH^\ddagger is formed from TH by an excited state isomerization reaction upon 405 nm illumination and evolves further to C^- by consecutive relaxation steps. Given the time resolution of our experimental investigation, all the fast relaxation steps happening in the ps – ns timescale are indistinguishable, and we have collected them over CH^\ddagger . The last step of this chain relaxes in the μ s and therefore it is relevant and observable in our data (Woodhouse et al *Nat Comm* 2020). CH_{int} relaxes to the on-state (C^-) with a much slower timescale of ~ 1 ms.

The reviewer suggests that the CH state, in fast pH equilibrium with C^- , could be relevant because of its ability to absorb 405 nm light and interconvert to TH and bleach. In the kinetic model used throughout the manuscript we neglected the pH equilibrium based on the following assumptions: (i) $pK_a=5.9$ and at experimental pH of 7.5 the equilibrium concentration ratio is $[CH]/[C^-] \sim 0.025$; (ii) our experiments use cyclic illumination schemes that induces cyclic modulation of the populations of the states of rsFPs (Fig. R4b). Light at 405 nm is shined on the sample mainly on the off-state. After a waiting time of 1 ms, light at 488 nm is shined on the sample mainly on the on-state. This is a rather general properties of all our sequences, and 405 nm light is never shined on the sample mainly in the on-state C^- when also its protonated partner CH has the chance to accumulate from the fast pH equilibrium.

We decided to test these assumptions by setting up an extended version of the kinetic model including CH and the connected pathways, we will refer to this kinetic model as “with pH equilibrium” (Fig. R4a). The parametrization of the kinetic constants connected to the equilibrium C^-/CH is set up using $pH-pK_a$ and the deprotonation time of CH of 35 μ s from Woodhouse et al., *Nat. Comm* 2020. All the remaining spectroscopical parameters of CH are borrowed from CH^\ddagger , including the bleaching quantum yield. To examine if the kinetic model with pH equilibrium would predict notable changes with respect to the original kinetic model, we reproduced Figure 1d,f with using both models (Fig. R4c). We observe only minimal changes of the predicted fatigue fraction. We believe that the simplified kinetic model without the pH equilibrium is reasonable in the context of our cyclic experiments with rsFPs, we also acknowledge that this simplification might not be valid in other conditions, i.e. not cyclic experiments or if relevant 405 nm light doses are shined on the system with substantial on-state population.

2/ According to Figure S1, the triplet state is only accessed from C^- , and does not absorb 405 nm light. Yet, the authors make a very strong statement at the beginning of the manuscript (P4-L97: “the data reports the 405 nm light to be the main driving force for increased photo-switching fatigue in rsEGFP2”). If the triplet state is indeed one of the main photobleaching mechanism induced by 405 nm light absorption, either it should be accessible by ISC from a 405 nm absorbing species (a protonated species) or the triplet itself should be able to absorb this light. Otherwise triplet state bleaching must arise from 488 nm light absorption, and in fact this is underlying the discussion along the whole manuscript. This is very confusing. For example, the initial rise in the fatigue fraction seen in Figure 1f relates to 488 nm induced bleaching from

an increasingly high C^- population due to increasing 405 nm light, which is an indirect effect. In view of this introductory sentence stating that photobleaching relates to 405 nm light, and also figure 1C showing little effect of the 488 nm light, all this needs strong clarification.

Firstly, we apologize for the lack of clarity: 405 nm is indeed one of the wavelengths that promote the excitation of the first triplet state of C^- to a higher triplet state in our kinetic model. We have updated the

Figure R4 (a) Proposed kinetic models with C^-/CH pH equilibrium for the photophysics rsEGFP2, the kinetic model without pH equilibrium can be found in Supplementary Figure 1b. In the kinetic model with pH equilibrium we have included CH as an additional state that can be photoswitched to TH or bleached upon 405 and 488 nm illumination. The colored arrows describe processes that are light-driven and the color of the arrows indicate the wavelengths in the pulse scheme (405 nm, violet; 488 nm, cyan; 592 nm, orange). (b) Representation of the time evolution of TH , CH_{int} and C^- over the sequential illumination profile generally used in the experiment, i.e. 488 and 405 nm interleaved with a delay without any illumination (c) Photobleaching dependencies to 405 nm power densities. The addition of CH did not significantly change the magnitude of the fatigue fraction.

illustration of Supplementary Figure 1 to clearly show the wavelengths at which each state is optically accessible using a color legend.

The statement regarding 405 nm light being the main drive force for increased photo-switching fatigue in rsEGFP2 emerges from the experimental data. An increase of 405 nm power in our typical experiments has a higher impact on the fatigue fraction compared to an increase of 488 nm light. As the reviewer comments, there are two modes in which 405 nm light contributes to bleaching. At low power density in Fig. R5 (cyan box) there is an indirect effect: the increasingly high population of the on-state makes the sample more capable of absorbing 488 nm light stimulating the bleaching pathways from the triplet states. At high power density (Fig. R5, violet box), when the on-switching process is saturated, direct bleaching from absorption of 405 nm light becomes increasingly relevant. We clarified this aspect on the main text.

Even if there are not explicit triplet states connected to the off-states in our simplified kinetic model, we cannot rule out that some of the bleaching involves them, and the direct bleaching pathways from 405 nm absorption are to be intended as effective ones, modeling in a lumped way all the bleaching pathways. Nevertheless, the short fluorescence lifetime of the off-state should reduce the ISC quantum yields, making the population of triplet-states less probable, and overall it should not be as relevant for the overall photo-physics.

Figure R5. The fatigue fraction at increasing 405 nm power density is shown in relation of the on-switching fraction of the 405 nm light.

3/ In fact, the authors need to justify why T1 (Figure S1) would not absorb 405 nm light. This is by no means suggested by the experimental spectrum in ref 28, which shows at least as much absorption at 405 nm compared to 592 nm. It is in fact likely that T1 absorbs violet light. Have the authors considered the scenario where the off switching phase populates T1 and the immediately following on switching phase leads to reabsorption of 405 nm light by T1? Adding a delay between switching cycles would relieve this switching pathway, as observed.

As in the previous reply, the triplet state 3C is indeed optically accessible by 405, 488, and 592 nm light with the cross-section for each wavelength as in Rane et al. 2023, *J. Phys. Chem. B* (Fig. R1 and new Figure 2d in the main manuscript). Therefore, the effect that a 405 nm pulse can play on the triplet population created by the previous off-switching is included in the kinetic modeling. In the following main point 4 of the peer review, we go more in depth on the discussion about interpreting data with a delay between switching cycles, that also lets the complete relaxation of the triplet state population before shining a 405 nm light dose in the next photocycle.

4/ As mentioned by the authors, the data in Figure 1g suggests the presence of one (or several) additional dark state(s) with a lifetime significantly greater than that of T_1 (P4-L122). Such additional dark states are indeed expected, and may be radical species. They, too, may absorb 405 nm light to promote either recovery or photobleaching. Can the model of the authors reproduce the data of Figure 1g? I suspect not, because the long-lived dark states in question are not part of the model, although they are probably crucial to the behavior. The authors might want to consult the following paper: Duan et al, DOI: /10.1021/ja406860e.

Motivated by the reviewer's comment, we tested the expansion of the model to encompass long-lived dark states (Fig. 6a). This helped to rationalize the data in Figure 1g in the main text.

Following the observations in IrisFP (Duan, C et al. 2013, *J. Am. Chem. Soc.*, Roy, A. et al., 2011, *J. Am. Chem. Soc.*), we propose that in the first excited triplet state (T_1) an electron transfer reaction may occur that will form a radical state (R^\bullet). The radical state can thermally relax back to the emissive state (C) or

Figure R6. (a) Schematic representation of the photophysical scheme accounting the formation of radical states (R^\bullet) via the first triplet excited state (T_1). Similarly to the model proposed for IrisFP (Duan, C et al. 2013, *J. Am. Chem. Soc.*, Roy, A. et al., 2011, *J. Am. Chem. Soc.*), the radical state can relax back to the emissive state (Cis) or further evolved to a photobleached state. The red coloured arrows indicate the newly added photophysical pathways and the fitted parameters (τ_{R^\bullet} , $\Phi_{T_1\text{-to-radical}}$ and $\Phi_{Bleach\text{-radical}}$). (b) Experimental data and simulation of photoswitching fatigue thermal recovery. For this experiment, the 405 nm dose was 1 ms and 240 W/cm² while the 488 nm dose was 5 ms and 40 W/cm². The dotted lines represent the simulation while the shaded areas are $\pm \sigma$ of the experimental data. Below, are the residuals of each experimental curve as $data - simulation$. (c) R^\bullet at the end of each photoswitching cycle pulse scheme. For a total dwell time of 10 ms (blue curve) the population of radical species continuously increases, while for longer total dwell (25 ms – orange, 50 ms – yellow and 100 ms – magenta) times the radical state population reaches a saturation level.

evolve further into a bleached state. The kinetic model fitted on the experimental data suggests a radical state lifetime of ~ 17 seconds (Fig. R6b), in agreement with previous reports of reversible photobleaching in both organic dyes and fluorescent proteins (Ha, 2012; Schuster et al., 2005; Sinnecker et al., 2005; Berardozzi et al., 2016).

From the kinetic model, the concentration of the radical state shows a continuous build-up (Fig. R6c). With longer waiting time the radical state has more time to relax, and a lower steady state level is reached. Under the main characterization condition of the presented work, the radical lifetime (~16 s) is in the same order of magnitude of the experimental time (~ 20 s). The radical bleaching and recovery will progress slowly during the experiments, on average the radical species will be a passive player during the experiment mimicking a bleaching species. Accordingly, in the minimal kinetic model presented here, the radical pathway is effectively lumped with irreversible photobleaching. Overall, this will result on a reasonable representation of the overall kinetic behavior with some small biases on the specific kinetic constant related to the triplet branch. It is important to note that co-illumination with light above 592 nm completely bypasses radical formation by directly accessing the triplet state, the precursor of any long-lived dark state.

Because of the strong correlation of the triplet and radical parameters, attempting to fit the expanded kinetic model simultaneously will converge difficulty, weakening the presented effective kinetic model. In this regard, the lifetime of the radical that we have estimated could be deduced only once all the other parameters have been fixed.

Considering the radical lifetime could provide practical guidelines in imaging experiments. In widefield microscopy, slower acquisition speeds (< 0.05 Hz) will minimize the build-up of substantial steady-state radical population, whereas faster speeds (> 0.05 Hz) will maximize it. In confocal imaging, the scanning inherently introduces dark intervals of 20–100 ms or more, with parallelized confocal and RESOLFT systems providing shorter dark times than point-scanning approaches. However, the radical lifetime in biological samples could be shorter than our experiments in PAA and difficult to predict a priori. Moreover, in live-cell experiments, acquisition timing is dictated by the dynamics of the biological process rather than by RSFP photophysics.

This investigation has been now added in a new Supplementary Note 5 and referenced accordingly in the main text.

5/ The authors, based on Supplementary Note 3, conclude that the Int intermediate absorbs light at 488 nm and is substantially fluorescent. How do they reconcile this finding with the fact that Int has been reported in several publications to be a protonated species, hence with little absorption at 488 nm and, in principle, very low fluorescence? The authors report an extinction coefficient of Int at 488 nm of 28,000, based on their data and also based on ref 2 (supplementary table 2). This is very unlikely for a fully protonated species, and ref 2 does not support such value (the authors have probably misread table 1 in ref 2, the extinction coefficient of ~28000 of the fully neutral species is at 403 nm, not 488 nm).

We thank the reviewer for pointing out the miss reference of the extinction coefficient value from reference 2 in the old Supplementary Table 2. We removed the reference in the updated version of the Supporting Information. The value of the extinction coefficient of CH_{Int} was deduced primarily and in a self-consistent manner from our experimental data according to our updated Supplementary Note 3. The nature of CH_{Int} is assigned from the best knowledge available in literature, i.e. from Uriarte et al. *J Phys Chem Lett* 2022.

The extinction coefficient of $28000 \text{ M}^{-1}\text{L}^{-1}$ best reproduces our transient fluorescence experiments in Figure 2c of the main manuscript. The only assumption was to assign the same quantum yield of fluorescence to CH_{Int} and C^* , and assign the difference in fluorescence signal only to a difference in extinction coefficient. We clarified these assumptions in Supplementary Note 3. A proof that CH_{Int} is substantially fluorescence comes from the experiments with impulsive on-switching light at 405 nm in Supplementary Figure 4, which show a rise of fluorescence emission with approximately 5 microseconds time constant. The non-negligible

extinction coefficient value at 488 nm of CH_{Int} is also supported by fast-spectroscopy data (Woodhouse *et al Nat Comm* 2020, Figure 2), that shows the initial shift and growth of the main transition band of the on-species in a time-scales compatible with the lifetime of CH_{Int} .

It is difficult, based on supplementary figure 3A, to understand how supplementary figure 3B is obtained: at early delays, the integrated fluorescence from this figure is about half of the full signal at late delays. But this is not what is suggested by looking at panel A, where the area under the blue curve (5 μ s) seems much less than the area under the red curve (51 ms), not even considering the baseline due to residual background. So please explain how supplementary figure 3B is obtained. If a residual substantial amount of the on-state is present at the beginning of the second 488 nm pulse, this will contribute to the fluorescence signal in all curves of panel A, so how do the authors make sure that this amount is negligible?

The experiment in Supplementary Figure 3a consists of a first long 488 nm pulse that will trigger the complete off-switching transition. Given that there is a non-zero probability of on-switching at 488 nm, there will be a residual signal at the end of that illumination pulse. Immediately after, a burst of 405 nm will induce the on-switching transition and after a variable delay a 488 nm illumination dose is used for fluorescence read-out and to bring most of the molecules back to the off-state. If there were no 405 nm illumination, the signal at the 2nd 488 nm pulse would remain constant at equilibrium value of the end of the 1st 488 nm pulse (assuming no significant relaxation processes in the first few μ s). The background level at the end of the first 488 nm pulse is subtracted from the signal of the overall time trace. Therefore, we established that any signal that we observe after the 405 nm burst will come from the on-switching transition and the dynamics we observe in the integrated signal of Supplementary Fig. 3a,b stem from the relaxation of a ms-lived state that appears after on-switching. The computation of the values shown in Supplementary Figure 3b is done with a trapezoidal numerical integration method.

We enriched and clarified the explanation in the relative Supplementary Note 3.

This being said, I agree that the microsecond rise in the fluorescence could point at Int being fluorescent. All in all, the authors should carefully clarify this point. If they maintain that indeed Int must be fluorescent, they should state it clearly in the main manuscript, as this is a very important finding in view of the available literature, and they should provide a possible mechanistic scenario.

We clarified in the main manuscript that the argument supporting the fluorescent nature of CH_{Int} comes from the microsecond rise of the fluorescence in experiment with impulsive 405 nm light on-switching. The nature of state CH_{Int} is interpreted as described above according to recent spectroscopic literature. We thank the reviewer for stressing that this is an interesting finding that deserves to be highlighted more in the main text.

Furthermore, if Int is indeed fluorescent, why would it not undergo ISC to the triplet state in the proposed model? Is there any good reason to neglect this possibility?

We have not included the formation of triplet state from CH_{Int} in our simplified kinetic model. We agree with the reviewer that if a lot of fluorescence would be emitted by CH_{Int} , the formation of its triplet state would become a relevant pathway. We believe that the simplification in the proposed kinetic model holds for the description of our bleaching and imaging experiments because we always wait 1 ms between 405 nm and 488 pulses. During the 1 ms waiting time, most of the CH_{Int} fraction is converted into C, significantly reducing the fluorescence signal generated from CH_{Int} and the formation of its associated triplet population. This aspect is represented more clearly in the sketch of Figure R4b and Supplementary Note 1.

6/ On which basis do the authors assume that C⁻ may convert to Int as a ground state process ? What is “tau_protonation-C” in Supplementary Table 1 ? This parameter is not reported in Supplementary Table 2. Is it used in the actual simulations ?

Following the interpretation of the states according to Uriarte et al. *J Phys Chem Lett* 2022 in Fig. R3a and the refined state model as in Supplementary Figure 2 of the updated Supporting Information, we corrected the kinetic model. CH_{Int} relaxes to C^- as maturation of the on-state. Conversely C^- does not convert to CH_{Int} . C^- could interconvert to CH through a ground state pH-equilibrium, that is not included in the simplified kinetic model used throughout the manuscript. We have now made this clarification in Supporting Information. Accordingly, the parameters table is now updated removing the parameter connected to the kinetics $C^- \rightarrow CH$. This discussion was expanded above in major point 1, where we argument on the relevance of the pH equilibrium for our experimental conditions.

7/ The authors invoke a direct bleaching pathway from TH, which apparently does not involve the triplet state. The authors should explain why the triplet state is not considered here ? Is this because of the short lifetime of the nonfluorescent TH? This would make sense, but if the triplet state is not involved, then, photobleaching must arise from the excited singlet state manifold. The authors should provide more details here, notably as RISC is not going to help at all in this case, meaning that, according to the author’s view, light induced recovery by red shifted light would not directly relate to 405 nm induced photobleaching (see point 2 above) ... in stark contradiction with the statement that photobleaching mainly arises from excitation at this wavelength. Could it still be that the triplet state and downstream radical species be reached from neutral chromophore species? The authors might be interested to read: Wulffele et al, DOI 10.1021/acs.jpcllett.2c00933.

According to ultrafast spectroscopic data, the lifetime of the off-state is a few picoseconds, e.g. from Uriarte et al, *J Phys Chem Lett*, 2022. As the reviewer suggests, this implies a lower chance of populating the triplet states through ISC from the off-state. In our simplified kinetic model we have a pathway for direct photobleaching from the off-states, mainly driven by 405 nm light. This is an effective pathway necessarily to reproduce the experimental data and account for the direct photobleaching with 405 nm light that happens when the on-switching process is saturated, i.e. when the population is driven almost fully to the on-state at every cycle. We cannot rule out the complete absence of a small fraction of triplet state from the off-states, nor the formation of very long-lived dark states with lifetime much slower than our experiments. Nevertheless, bleaching from the off-states is to be taken as a lumped process that models in a simplified way the direct bleaching.

As discussed above in major point 2, increasing the 405 nm power density when below saturation will indirectly increase the population of C^- at every cycle, and it will stimulate the formation of an increasingly high amount of triplet state population and the connected pathways (RISC/bleach). Thus, even if we notice an increase in fatigue with increasing 405 nm power density, in the regime below saturation, triplet mediated bleaching is still the most relevant channel and rescuable with RISC. We clarified in the main text that our statement regarding the trend of the fatigue fraction with 405 nm increasing power density does not mean that the bleaching only arises from the direct excitation with this wavelength. The experimental data suggests that direct bleaching from excitation of the off-states is not effectively rescuable by the red-shifted light. It is reasonable to assume that the possible intermediates mediating bleaching are not absorbing substantially at the red-shifted wavelengths used for rescuing, or that rescue mechanisms from those states are not very probable.

8/ The authors find that increased power density 488 nm results in higher light induced recovery (figure 2e). This is unexpected in view of the fact that a high-level of 488 nm illumination should by itself promote RISC, decreasing the additional impact of red shifted light. In fact this was noted in refs 28 and 29. I feel

that the data shown in figure 2e are misleading, as here the dose of illumination at 488 nm is kept constant, that is, the data at 0.42 kW/cm² are collected with a shorter pulse of 488 nm light than the data at 0.2 kW/cm², while at the same time, if I understand correctly, the 592 nm illumination is kept at 1 ms duration. In this situation, the observation can be easily explained, as, as soon 488 nm illumination terminates, the additional time until the next cycle becomes free for either thermal or light induced relaxation from T1 to S0. These aspects should be clarified to avoid this possible misunderstanding that the more 488 nm illumination the more 592 nm light induced recovery by RISC.

We are aware of the opposite behavior of the studied rsFPs to 488 nm power ramp respects to what reported in the mentioned reference. This is due to the photoswitchable nature of the rsFPs respect to the fluorescent proteins under investigation by Ludvikova et al., 2023, *Nat Biotechnol*. Our aim is to investigate negative rsFPs in conditions close to imaging settings, and this demands working at constant 488 nm energy dose. The user would only deliver the necessary light dose to the sample, meaning that to reach the same off-switched fraction at higher power densities, one would illuminate for a shorter period. This constraint does not apply to conventional microscopy, where more illumination does bring to a steady increase of fluorescence. In super-resolution microscopy with negative rsFPs, the optimal spatial confinement is obtained at a fixed energy dose, i.e. when the illumination time roughly matches the off-switching time. In this scenario, the shorter the 488 nm light the lower the chance for the population in the triplet state to do RISC by 488 nm illumination. This defines the opposite behavior with respect to a conventional fluorescent protein where the 488 nm can always be increased to obtain a higher signal, and indeed this was the illumination condition explored by Ludvikova et al., 2023, *Nat. Biotechnol*.

We have now clarified this aspect in the main text.

9/ As shown by the authors the RISC effect can as well be obtained by near-infrared light, at 810 or 900 nm. In fact, near infrared light should be the privileged choice, as absorption of the triplet state around 900 nm has been shown to be maximum in green fluorescent proteins. The authors have used 592 nm light probably because this was technically easier (already implemented on their microscope). Interestingly, they found that the achievable effect with 592 nm surpasses that at 810 nm, even with the huge power densities they used in the near infrared. It would be nice to give a precise interpretation of what might be going on. At the end of supplementary note 7, the authors claim that the comparison is difficult based on the fact that the “spatial distribution of light-induced kinetic processes is radically different”. This statement is unclear to me, it would be nice if the authors could elaborate more.

In all cases it looks like, from supplementary figure 13, that no matter what intensity is used, the level of light induced recovery at 900 nm cannot be reached at 810 nm. Would the authors have a clue whether this is true and if yes, why ?

At the end, using visible red light to promote RISC is nice, as this wavelength is available right away on many microscopes. However, the authors should comment on possible penalties e.g. for 2-color experiments requiring 592 nm light to excite a second red fluorescent marker.

The difficulty in comparing 810 nm and 592 nm stands from the different experimental setups used for those two measurements: point-scanning with pulsed lasers on one side and widefield (~30 μm in diameter) with continuous wave lasers on the other. For this reason, in the work, we have decided to compare 810 nm with 900 nm, using the 810 nm as the control wavelength where we can assume a cross section for T1 similar to the one at 592 nm.

We agreed that it would have been valuable for the reader to have a direct comparison, therefore we have performed a new experiment where we directly compare 592 nm and 915 nm (as it is laser line available in commercial CW laser) on the widefield microscope. As can be seen in Fig. R7 at 915 nm the recovery is higher than at 592 nm. This aligned with Rane et al., *J Phys Chem B*, 2023, where the triplet state absorbs

more efficiently 915 nm than 592 nm light. At the same time, it is counterintuitive with the expectation of higher RISC probability for higher excitation energies (see also major point 1 of reviewer #1).

Similarly to what observed in the confocal data comparison, there is higher recovery when using 900/915 nm light. At the same wavelengths the triplet state absorbs more efficiently according to Rane et al, *J Phys Chem B*, 2023. The absorption cross-section of the triplet seems to be the dominant effect, compared to RISC efficiency and other pathways addressed by the red shifted light, following the trends reported in Ludvikova et al, *Nat Biotechnol* 2023, where the higher RP effect matches the peak of the triplet absorption spectrum of EGFP.

The new data is now available in Figure 2j and Supplementary Note 10.

Figure R7. Light-induced recovery as a function of the 592 nm and 915 nm power density in the widefield microscope. We observed a more efficient recovery as well as a higher maximum recovery value with 915 nm co-illumination than at 592 nm. The dashed lines correspond to an exponential function ($y = a(1 - \exp(x/k))$) fit performed on the data with $k_{915} = 0.42 \text{ kW/cm}^2$ and $k_{592} = 1.33 \text{ kW/cm}^2$.

10/ It would be nice to see the experimental data of figure 1F on a linear scale and without the fitted model data superimposed. It looks like these experimental data may follow a simple exponential growth on the used logarithmic scale, meaning a linear growth on a linear scale. Many readers could feel that using such a complicated photophysical model to fit essentially linear data is largely unjustified. Could the authors comment on this? What are the factors that prevent from only relying on a linear photobleaching effects by the 405 nm light?

We changed the plot of Figure 1f to linear scale, and changed the label to make it more clear.

We apologize for the lack of clarity in the main figure. Figure 1f of the main manuscript shows a section of the experimental data (dark shaded data of Figure 1d) together with the expected trends computed from the kinetic model. Supplementary Note 4 explains the procedure to fit the bleaching quantum yields of the kinetic model. Briefly we fit only two parameters for the kinetic model with only bleaching from the triplet states ($\Phi_{\text{Bleach-triplet}}$, $\Phi_{\text{Bleach-triplet-excited}}$) and three parameters for the kinetic model with also bleaching from the off-states ($\Phi_{\text{Bleach-triplet}}$, $\Phi_{\text{Bleach-triplet-excited}}$, $\Phi_{\text{Bleach-trans-neutral}}$). For the actual fitting we use a subset of the experimental data as shown in Supplementary Figure 7b of the revised Supplementary Information. Thus, Figure 1f of the main manuscript shows the extrapolated trend predicted from the kinetic model for a range of 405 nm power density. The computed fatigue fraction contains all the bleaching contributions included in the kinetic model.

11/ I am puzzled by supplementary figure 4: the data in this figure show the rapid growth of the integrated fluorescence which then levels off without any sign of further decay. On the contrary, the model predicts a decay, as Int is more and more redirected towards TH when the 405 nm power density is increased, and this decay is clearly visible in the fitted data. The experimental data here clearly seem to deviate from the model. Furthermore, the data of figure 1B seem to go in the same direction, as the initial point in the photofatigue decay curves appear to be significantly higher at 0.7 kW/cm² as compared to 0.22 kW/cm², as opposed to what the model would predict. In their experiment of figure S4, could the authors increase further the intensity of the 405 nm illumination to start seeing the expected decay (maybe using their confocal microscope)? Could it be that Int may also be photoconverted to C- by violet light, so that the relative amount of Int molecules sent to either C- or TH be relatively independent of 405 nm illumination ? It is known that chromophore deprotonation in FPs can be accelerated by light. These aspects should be discussed.

We investigated the apparent drop in the simulated integrated fluorescence curve by extending the experimental investigation at higher 405 nm photoactivation dose. We increased the 405 nm illumination time as we were limited in the power density we can deliver at the sample plane with the widefield configuration of the microscope. As shown in Figure R8a, increasing the illumination time yields a faster rise of the fluorescence signal at lower 405 nm, however, there is a decrease of the fluorescence at higher doses, and that drop is more significant for longer illumination times. Similarly, the normalized fluorescence in Figure Rb decreases at > 1 J/cm². Taking this into consideration, we rationalize the discrepancy between the simulated and experimental curve in the previous Supplementary Figure 4 as an overestimation of the delivered power density to the sample.

When we calculate the power density for a given condition, we assume that all the power measured at the back aperture of the objective will be delivered in the sample plane across an area with a diameter equal to FWHM of the beam profile (~ 30 - 40 μm). We choose our experimental region of interest (ROI) small enough so that the power delivered in the ROI is constant. We estimate that the discrepancies observed between simulations and the data stem from these considerations: i) there might be a loss of laser power between the point of measurement and the sample plane, ii) the extension of the beam profile that is relevant to the calculation of the power density in the sample plane is larger than FWHM, iii) the power across the experimental ROI is not constant which will lead to a reduction the photoactivation efficiency away from the center of the beam.

Figure R8. (a) Normalized fluorescence as function of the 405 nm energy. The data are coming from three dataset acquired over the same interval of power densities but increasing illumination time (1, 2 and 3 ms). (b) On-switching curve data and fit. Each data point is an average of 9 complete photocycle repetitions. The simulation consisted of a 5-state model without triplet and bleaching pathways. The parameter α scales the 405 nm power density such as the simulation matches the experimental data points.

To quantify the discrepancy between the experimental observations and the simulation output, we recompute the on-switching curve with an additional correction factor α (0.642 ± 0.06) that scales the power density such as the simulation output best matches the experimental values. We deemed that the best way to account for this source of experimental error is to study the correlation between the different parameters using the simulation tool. The resulting $\Phi_{INT \rightarrow TH}$ and ϵ_{405}^{INT} from the model fitting in Supplementary Figure 5 are to be understood in the context of a family of solutions where certain experimental parameters that greatly influence the model fit have an implicit uncertainty and correlation among each other.

This correction in power density is now applied in Figure 1a, 1b, 1d, 1f and 2a as well as Supplementary Figure 5 and 7 where we studied the dependency of the 405 nm power density and updated the experimental 405 nm power density values in Supplementary Table 10. The correction is introduced in Supplementary Note 3.

Concerning the question about a light driven pathway connecting CH_{int} and C , we have ourselves faced this question when building the kinetic model. In all the recent spectroscopic literature on negative RSFPs that we have consulted, the light always induces the cis/trans isomerization through a conical intersection relaxation. CH_{int} is a protonated species (Woodhouse et al *Nat Comm* 2020) thus we have assumed a cis/trans isomerization toward TH or a similar species. Of course, we would be happy to consult the literature that the reviewer is suggesting that could help the interpretation of the photoinduced processes of CH_{int} .

12/ The scheme in Fig1e is difficult to read: the triplet state between C- and B should be shown in clear, notably as the trans states (T) could be confused by some readers with the triplet state (T1). The protonated chromophore state in equilibrium with C- should be shown as CH, and the short-lived intermediates preceding Int along the on switching pathway should be given a different name.

We have updated Figure 1e with a simplified scheme that primarily reports on the two bleaching pathways identified in the study: triplet-mediated and off-state-mediated bleaching, with the wavelengths that mainly drive these pathways. An updated full scheme is now presented in Supplementary Figure 1b, where we have clarified and corrected the nomenclature and identities of the different states considered for building the kinetic model.

Overall, I strongly encourage the authors to revisit their photophysical model in depth. They should also much improve the clarity of their manuscript so that the essential message carried by this model can be captured by the white audience of Nature Communication. At present, what I can extract from the manuscript is that: (i) the effect of 405 nm illumination on photofatigue either relates to the triplet state only via the amount of the on-state population at the beginning of off switching phases, or goes through a singlet state photobleaching mechanism that is nonrecoverable by RISC and (ii) recoverable photobleaching through T1 only occurs upon 488 nm illumination of the on-state.

The comments of both reviewer #1 and #2 helped strengthen the model and define its working conditions. As we mentioned also at the beginning of this point-to-point rebuttal letter, we conceive the model as a descriptive tool rather than a predictive one. Overall, the model acts in support of the main observations of the work that are exactly what the reviewer highlights.

One possible strategy that the authors could consider would be to investigate whether their photophysical model predicts a pH dependence of photofatigue and to carry out corresponding experiments to see whether these predictions are realized. For example, at low pH the fraction of protonated chromophores will increase, likely increasing direct bleaching by 405 nm light. It could also be expected that relaxation of Int

to C- slows down, making the initial drop faster. In view of the complexity of the problem, those pH dependent experiments could help to narrow down the photophysical model.

Following the reviewer suggestion, we performed a series of photoswitching fatigue experiments at different pHs (6, 6.5, 7.5 and 8) and with varying 405 nm power densities. Across the 2-dimensional space of parameters we monitored the fatigue fraction observed in our data and compared it to the prediction of the kinetic model. Overall, we see a very good agreement between our experimental data and the simulated fatigue fraction, especially notable at low 405 nm power densities. At higher 405 nm power densities, our data shows a less pronounced pH dependency of the fatigue fraction than what is predicted by the kinetic model. To check whether the pH of the rsEGFP2-PAA embedded gel was correct we monitored the mean initial fluorescence intensity as function of the pH and we see a sigmoidal shape as would be expected.

Our photophysical investigation aims at moving close to live cell imaging conditions and therefore guides optimized imaging. For this reason, the majority of our experiments were carried out at the cell's physiological pH of 7.4, and we see a good agreement between experimental data and the model within a unit pH around the physiological pH.

Figure R9. (a) Fatigue fraction as a function of the pH and the 405 nm power density. Overall, the simulated photobleaching matches the experimental data across the different explored pH. At higher 405 nm power densities, our data shows a less pronounced pH dependency than the prediction by the kinetic model. (b) Fatigue fraction data and simulation at different pHs. At pH = 6.0 our data shows a lower fatigue fraction than predicted by the simulation.

Other major points

13/ The authors nicely apply their methodology to both prokaryotic and eukaryotic samples. Figure 4A and 4C suggest that there is a real benefit of applying 592 nm co-illumination. Yet, it is not clear whether the protecting effect shown in the photofatigue decay curves of figure 4A is as pronounced as that observed in vitro. Could the authors comment on this ? Do they see a possible effect of polyacrylamide in augmenting the effect, maybe about oxygen level and/or diffusion, and thus triplet state lifetime, and even maybe propensity for radical formation ? In view of those photofatigue decay curves in figure 4A, which after all predict a relatively faint improvement, how representative are the series of images shown underneath ? The authors may want to show the photofatigue decay of this very sample in the SI. In passing, the first frame of the “with co-illumination” series appears less intense than the first frame of the “without co-illumination” series. Does this correspond to a true, reproducible effect ? If yes, what happens ?

To help the transversal comparison among the different systems studied, from synthetic PAA layers of the RSFPs to cells, we have compared the fatigue in each sample under identical illumination conditions and the recovered fraction under 592 nm co-illumination (Fig. R10).

We set the imaging conditions to maximize the temporal resolution of the acquisition with 4 ms total dwell time and 60 nm scanning steps and monitored the fluorescence over 50 imaging frames. As shown in Figure 10a, different live-cell samples (bacteria, vimentin, actin and mitochondria) have their characteristic bleaching curve. Additionally, we incorporated a sample of rsEGFP2 embedded in a PAA matrix at the same concentration as in the characterization experiments. We observed photobleaching between 20 to 40 % in eukaryotic cells, with actin labelled rsEGFP2 showing the higher bleaching resistance (29% normalized fluorescence at the last frame in magenta) compared to rsEGFP2 tagged to vimentin and mitochondria (15% and 16% normalized fluorescence at the last frame, in yellow and green, respectively). In prokaryotic cells, we observed the highest photobleaching resistance (52% normalized fluorescence at the last frame in orange) which we attributed to the specific geometry of the sample, i.e. a thick pellet of bacterial cell rather than a semi-2D sample like for the other samples. Similarly, we measured the photobleaching with 592 nm co-illumination (15 kW/cm² and 0.5 ms) added in the pulse scheme and we observed a decrease in photobleaching in all samples, specifically higher in bacteria (21% recovery in orange) and in the PAA layer (13% in blue). An interpretation of these results can be linked to a slower diffusivity of molecular oxygen both in biofilms (De Beer, D., et al, 1994, *Biotechnol. Bioeng.*, Stewart, P.S., 2003, *J. Bacteriol.*) and in the PAA matrix (Hepworth, S.J., et al, 1999, *Phys. Med. Biol.*, Ju, L.-K., et al, 1986, *Chem. Eng. Sci.*, Marek, P., et al., 2014, *J. Sens. Sens. Syst.*), which can prolong the lifetime of the triplet and reduce its reactivity downstream to generate irreversible photobleaching. In this context, given that live-cells are imaged in aqueous media, is plausible that the reactivity of the triplet state of rsEGFP2 generates more photoproducts that can lead into photobleaching and the observed fluorescence loss.

Regarding the difference in the first frame of Fig. 4a, the reason lies in the variability of expression levels associated with transient transfection, explicitly shown in Supplementary Figure 25c as the bleaching constant against the initial signal-to-background ratio. The average raw and normalized bleaching curves have now been added to the same Supplementary Figure 25 for completeness.

The presented comparative study is now reported in Supplementary Note 14.

Figure R10. (a) Intensity traces for a variety of samples containing rsEGFP2 imaged in parallelized confocal mode. In blue, a rsEGFP2-embedded polyacrylamide layer; in orange, a biofilm of *Escherichia Coli* expressing rsEGFP2; in yellow, rsEGFP2-vimentin in U2OS cells; in magenta, rsEGFP2-ActinChromobody in U2OS cells, and in green, rsEGFP2-OMP2 in U2OS cells. The lines represent the mean intensity traces, and the shaded area shows $\pm \sigma$ from the mean. (b) Light-induced photobleaching recovery with 592 nm co-illumination in the aforementioned samples. The 592 nm illumination dose were 0.5 ms at 15 kW/cm². The bars represent the mean light-induced recovery and the error bars shows $\pm \sigma$ from the mean.

The Mona Lisa data are also nice and seem to show a significant gain. Would there be a way to more quantitatively assess the spatial resolution loss along multiframe data collection with and without co-illumination, rather than showing just one profile at a user chosen position (which is the classical old procedure often criticized for potential bias) ? Possibly, a number (~ 10) of random locations across multiple fibers could be picked and the profiles presented in the SI ?

Following the suggestion of the reviewer we have extended the characterization of the SNR and spatial resolution at different time points within MoNaLISA timelapse recording of U2OS cells tagged with LifeAct-rsEGFP2 (Figure 4c in the main text). We traced $\sim 20 - 40$ line profiles of actin filaments per image and performed a Lorentzian fitting with a custom-written MATLAB script. We calculated the SNR as well as the FWHM of the Lorentzian peak of each line profile in the first and last frame of each timelapse recording. As shown in Figure R11a, the SNR with or without 592 nm co-illumination is higher in the initial frame (median values 14.01 with 592 nm, 10.88 without), however we observed a greater reduction in SNR without 592 nm co-illumination ($\sim 40\%$ decrease in SNR after 20 RESOLFT imaging frames) than when the 592 nm co-illumination is applied ($\sim 27\%$ decrease in SNR after 20 RESOLFT imaging frames). Along with the intensity traces shown in Figure 4c of the main text, the SNR quantification highlights the benefits of incorporating the 592 nm co-illumination for prolonged timelapse imaging in intensity-demanding techniques such as MoNaLISA. At the same time, we quantified the FWHM measured on > 100 actin filaments (over 4 – 5 cells) per condition in Figure R11b. Considering that our determination depends on a fitted parameter, what we see as a result of the reduced SNR is a broadening of the FWHM distribution (Figure R11b) and an increase on failed fitting score (Figure R11c) at the 20th frame without co-illumination.

This further quantification is now reported in detail in Supplementary Note 14 and Supplementary Figure 26.

Figure R11. (a) SNR of U2OS-LifeAct-rsEGFP2 measured from > 100 actin filaments per condition. The size of the box represents the 25th-75th percentiles of the data, and the lines represent the median value. The whiskers extend to 1.5 x the interquartile distance below or above the 25th-75th percentiles. The reduction in SNR at the 20th frame is less significant when 592 nm co-illumination is applied (27 %) than when is not (40 %). (b) FWHM from Lorentzian fit of U2OS-LifeAct-rsEGFP2 measured from > 100 actin filaments per condition. The size of the box represents the 25th-75th percentiles of the data, and the lines represent the median value. The whiskers extend to 1.5 x the interquartile distance below or above the 25th-75th percentiles. The distribution of sizes of the measured filaments in the 20th frame without 592 nm co-illumination is skewed towards broader filaments indicating a loss of resolution. (c) Percentage of fits that lie within 3σ of the median. Applying the 592 nm co-illumination increases the number of representatives fits at 20th frame.

More minor issues

1/ The authors investigated the effect of modulating the on switching light (essentially adding dark times, or reducing power density at constant dose, both resulting in increased data collection time) in the context of their photophysical studies, but they have not implemented such strategies for data collection of biological samples. Thus L19 in the abstract does not really reflect the content of the paper.

Following the reviewer's suggestion, we verified the direct influence of on-switching time modulation on imaging. In Figure R12 (now Supplementary Figure 14 and Figure 2b), actin filaments labeled with rsEGFP2 were imaged using on-switching times of 20 ms and 0.25 ms for a power density of 0.14 and 1.05 kW/cm² respectively; the displayed image represents the pixel-wise ratio of these two acquisitions. Because fluorescence increased in the image recorded with a longer 405 nm illumination duration at lower intensity, the histogram of the resulting ratio is shifted above 1.

Figure R12. (a) *Slow image* pulse scheme. The 405 nm illumination dose was set to 20 ms and 14 W/cm², the fluorescence was captured with a 488 nm read-out pulse (500 W/cm² and 1 ms). (b) *Fast image* pulse scheme. The 405 nm illumination dose was set to 0.25 ms and 1.05 kW/cm², the fluorescence was captured with a 488 nm read-out pulse (500 W/cm² and 1 ms). (c) Ratiometric image (*Slow image / Fast image*). The pseudocolor of each pixel in the actin fibres shows the value of the intensity ratio. (d) Histogram of the pixel values in (c). The colour bar on top represents the pseudocolor in associated to each bin in the histogram.

2/ P2-L45/46: this sentence calls for references

We have now added a reference to a review article in the above-mentioned position.

3/ P3-L64/65: maybe replace “dark state” by “triplet state”: as is, it looks like the authors are not sure of the nature of the state, whereas their photophysical model clearly identifies the triplet state.

We have now updated the text accordingly, clearly addressing the triplet state since the beginning of the text.

4/ P3-L75: the authors might be interested to read: Adam et al, <https://doi.org/10.1002/cphc.202200192> It would be interesting to know whether they noticed potential signs of the off-state heterogeneity described in this paper, in view of the ~kHz switching rates that they apply (e.g. non-mono exponential switching kinetics).

We thank the reviewer for the insight. With the focus of the work being on the photoswitching fatigue, we only considered the overall off-switching rate of rsEGFP2 to define the on/off cycle timing for the fatigue measurements, and did not examine the detailed heterogeneity of its off-switching kinetics.

5/ P4-L96/97: there seems to be a contradiction between this statement and all the discussion in supplementary note 9 on the different RSFPs tested. In this note, unless I am mistaken, the difference in switching fatigue between the variants is not assigned to 405 nm induced photobleaching, rather to a balance between off switching and ISC induced by 488 nm light. Please explain and improve consistency of the text in this regard.

We apologize for the confusion regarding the roles of 405 nm and 488 nm light in bleaching. Different RSFPs variants exhibit varying fatigue resistance, which is inversely proportional to their off-switching times. In our comparison, we deliver the same 405 nm light to all proteins, since their on-switching profiles are consistent, which means that the only variable that changes between the fatigue recording is the duration of 488 nm illumination. When we then investigated the effect of 592 nm co-illumination against this heterogeneous pool, we observed that not all rsFPs showed the same fatigue rescue percentage. Same behaviour when we increase the wait time between cycles to allow for thermal recovery. Because the 405 nm illumination remained constant, it is reasonable to attribute these differences to the component of bleaching that 592 nm co-illumination and thermal recovery can reverse, i.e. the triplet-mediated bleaching.

We believe that the restructuring of the main text, especially the first two paragraphs of the results, and the old Supplementary Note 9 (now 11) helped solve this confusion.

6/ P4-L122/125: increasing the dark time will not prevent T1-O2 interactions, resulting in ROS production. There are definitely several photobleaching mechanisms at play, please see Duan et al, DOI: /10.1021/ja406860e.

We have now clarified in the text that the increased dark time will allow the long-lived dark state originated from the triplet (such as radicals) to relax back to the ground state. We have also added the reference to the in-depth analysis of the thermal relaxation of long-lived dark states in Supplementary Note 5.

The text now reads:

The time dependence of such relaxation exceeds the expected triplet state lifetime, both reported³¹ and measured in our data, suggesting that the triplet state acts as a precursor for longer-lived photoproducts that lead to reversible and irreversible photobleaching (Supplementary Note 5). Increasing the interval between successive photocycles prevents the accumulation of these intermediate dark states.

7/ P5-L136: “we investigated how the modulation of the on-switching dose, given a fixed total energy, influences the fluorescence output”: this sentence does not make sense, the authors tend to make a confusion between power density and dose throughout the manuscript. Dose = total energy.

We apology for the mistake and we have now corrected it throughout the text.

8/ P7-L195/201 (and Table S2): is it reasonable to assume that the quantum yield for photobleaching from the excited triplet is wavelength independent? As different levels of excited triplet states might be reached as a function of illumination wavelength, the fate could be very different, for example with blue shifted light generating more photobleaching and red shifted light generating more RISC. This is really an open question but the issue should be discussed, and at least the assumptions made clearly stated. A related question is: is it reasonable to assume a much weaker quantum yield for photobleaching from the excited triplet state than from the ground triplet state (supplementary table 2) ? It would be expected indeed that the excited triplet state be more reactive than the ground triplet state.

In response to Reviewer #1 point 2 and to address Reviewer #2 major point 9, we have partially discussed this aspect. We agree that wavelength-independent photobleaching from the excited triplet states is a simplification. And as Reviewer #1 suggests, RISC should become more probable at blue shifted wavelengths. As stated in a few discussions above, this simplification follows the needs of the current work. On one side, there is no available literature able to provide the parameters for the pathways involving T_n at every laser line we used, and on the other side, our data would not be able to sustain more free parameters when fitting the fatigue fractions without adding large uncertainty to correlated parameters.

Within the simplification of our kinetic model, in rEGFP2 our data indicates that photobleaching from the excited triplet state is not the dominant contribution and RISC is more probable, allowing the rescue of fluorescence when 592 nm light is applied. Moreover, in the updated Supplementary Note 9, we discuss and explore the behaviors of the kinetic model as a function of the ratio of the bleaching quantum yields from T_l and T_n . We show that within our simplified model, rescue is possible only when bleaching from T_n is not the dominant fraction, and when $\Phi_T / \Phi_{Tn} > 600$ (Supplementary Figure 18b).

9/ P9-L276/277: why only dim samples?

The effect is indeed not limited to only dim samples, we have therefore better rephrase the sentence in the updated main text.

10/ P9-L282: “extends the absolute observation window”: by how much ? Also around 15% ? Please discuss this point.

We quantify the improvement in photobleaching with 592 nm co-illumination to be 20-30 % as that is the perceived decrease in fluorescence loss shown in the violin plots of Figure 4b; this is now clearly stated in the text.

11/ P10-L294/295: it is not clear to me why co-illumination should only be applied during the readout step. Why not during the switch off step, ideally using a doughnut-shaped red beam ? Isn't it during this step that most photobleaching occurs ?

Ideally, 592 nm co-illumination would be applied throughout the entire RESOLFT pulse sequence since triplet-state population can occur at any illumination, during both sequential illumination with 405 and 488 nm and 488 nm illumination only. In practice, however, we restricted 592 nm overlap to the excitation phase to suit our parallelized RESOLFT geometry. In our setup, off-switching is delivered in a dense 312.5

nm grid, while on-switching and 488 nm readout share a more relaxed 625 nm periodicity. Because only rsFPs within the on-switched regions undergo the majority of the cycling event, and are thus most susceptible to triplet-mediated bleaching, we patterned the 592 nm light through a matching microlens array that aligns with the on-switching and readout grid.

Figure R13. Illustration of the time sequence and spatial geometry for RESOLFT imaging in its parallelized implementation. The on-switching 405 nm micro-lens foci selectively defines the population that will undergo the following off-switching; therefore to maximize the rescue of the fatigue the co-illumination with 592 nm light needs to follow this same spatial geometry.

12/ P10-L312/313: in fact, a protective effect could be expected, as shown in ref 29. Could such a decrease in phototoxicity be observed by looking at mitochondria mobility?

Phototoxicity is a major concern when applying additional illumination to biological samples. To assess the phototoxicity induced by 592 nm co-illumination, we monitored mitochondrial mobility as an indicator of cellular health. Supplementary Figure 27 (and the related Supplementary Note 14) shows the mobility events recorded in time-lapse sequences with and without 592 nm co-illumination; no differences were observed between the two conditions. Mitochondria are highly sensitive organelles and are often used as phototoxicity indicators, since they rapidly respond to stress by increasing the rate of fission and formation of punctiform and fragmented mitochondria (Karbowski et al 2003, doi:10.1038/sj.cdd.4401260).

Furthermore, Ludvíková et al. (2023) extensively evaluated phototoxic effects of 900 nm illumination at intensities comparable to those used here (10-15 kW/cm²), reporting minimal damage under equivalent conditions. In our previous work (Pennacchiotti et al. 2018), we compared 592 nm co-illumination with shorter wavelengths (e.g., 488 nm) in the context of RESOLFT microscopy and similarly found that 592 nm caused less phototoxicity. Together, these observations support that the used 592 nm co-illumination protocol does not induce significant phototoxic effects.

13/ P10-L318/321: yes but no experimental demonstration was provided (see point 1 above). The downside of this strategy should also be mentioned (increase in data collection time).

As mentioned in Minor Point 1, we have now reported the experimental demonstration of the on-switching modulation in Figure 2b and commented accordingly in the text.

14/ P11-L326: again, “a dark state” rather than the triplet state is invoked here (see point 3 above). This really reads bizarre, as the triplet state is clearly identified throughout the results section.

We have now updated the nomenclature throughout the text.

15/ P11-L346: the authors should recapitulate the benefits of using either visible red light or near infrared light. Have they tried to implement 900 nm illumination on their RESOLFT microscope ? This would be a very nice experiment to do, to check the pros and cons.

As noted in Major Point 9, we have implemented 915 nm illumination in our RESOLFT microscope, although so far only for photophysical measurements. We have now expanded the discussion to consider the feasibility of using 915 nm light as co-illumination. Compared to 592 nm light, NIR illumination offers reduced phototoxicity and simplifies multicolor imaging by minimizing spectral overlap, although it is a more unusual wavelength in commercial systems.

The text now read:

By using 592 nm light, a wavelength already available on most commercial systems, to drive triplet-state recovery, this strategy can be seamlessly integrated into diverse imaging modalities. However, because 592 nm is a common wavelength for orange/red fluorophores, it may impair multicolor acquisitions. This limitation can be addressed by exploring other wavelengths within the triplet absorption spectrum of rsEGFP2, which peaks around 900 nm³¹.

16/ The authors have developed the nice rsFusionRed proteins. Readers will naturally wonder whether they have assessed the potential photofatigue enhancement of those red rsFPs with near-IR light. I encourage the authors to answer this question.

After incorporating 900 nm illumination, we tested the co-illumination strategy on rsFusionRed3, replying to the reviewer and our curiosity. We evaluated the fatigue under 405 nm, 592 nm, and 915 nm light but observed no significant increase in fatigue decay. Given the complexity we previously noted in green negative rsFPs, we cannot generalize this finding to the entire portfolio of negative red rsFPs; however, we believe this is beyond the scope of the current investigation.

Figure R14. Light-induced recovery in rsFusionRed3. (a) Light-induced recovery as a function of the 405 nm power density. The 405 nm on-switching dose consisted of 7 ms illumination time with variable 405 nm power density (13 – 550 W/cm²); the 592 off-switching dose consisted of 4 ms illumination time at 8.5 kW/cm² and the 915 nm dose consisted of 4 ms at 3.3 kW/cm² simultaneous with off-switching. (b) Light-induced recovery as a function of the 592 nm power density. The 405 nm on-switching dose consisted of 7 ms illumination time at power density 85 W/cm²; the 592 off-switching dose consisted of variable illumination times (50 – 4 ms) at 0.6 - 8.5 kW/cm² and the 915 nm dose consisted of 4 ms at 3.3 kW/cm² at the onset off-switching. (c) Light-induced recovery as a function of the 915 nm power density. The 405 nm on-switching dose consisted of 7 ms illumination time at power density 85 W/cm²; the 592 off-switching dose consisted of 4 ms illumination time at 8.5 kW/cm² and the 915 nm dose consisted of 4 ms at variable power density (0.17 – 4.1 kW/cm²) simultaneous with the off-switching.

Similarly to the investigation with rsEGFP2, we monitored the photoswitching fatigue of rsFusionRed3 after 2000th ON/OFF cycles and incorporated 915 nm co-illumination as the photoswitching fatigue wavelength. The experiments were performed in the microscope of Supplementary Figure 28 (update version of MoNaLISA set-up) and, unlike in rsEGFP2, we measured the photoswitching fatigue in biofilms containing *Escherichia Coli* expressing rsFusionRed3. We explored the photoswitching fatigue and the possibility of recovery from 915 nm co-illumination across a domain of illumination power densities for each wavelength. Briefly, the pulse scheme of the experiment consisted of a 405 nm on-switching dose followed by a 592 nm off-switching dose. The 592 nm illumination time was adjusted to each 592 nm power density so that the fluorescence was reduced to 20% of the maximum. The 915 nm was added in the pulse scheme simultaneously with the off-switching pulse. When exploring the dependency of the recovery with 592 nm power density, the 915 nm illumination time was selected as the shorter 592 nm pulse in the ramp (corresponding to the higher 592 nm power density). Below we display the dependencies of the measured light-induced photoswitching fatigue recovery to the different illumination wavelengths, from left to right: (a) 405 nm power density ramp, (b) 592 nm power density ramp and (c) 915 nm power density ramp. From our investigation, we cannot see any light-induced recovery from co-illuminating rsFusionRed3 with 915 nm light.

17/ It would be nice in the discussion to consider the potential effect of the level of oxygen on the observed light induced recovery. Less oxygen is typically beneficial for photostability because of decreased ROS production, but also increases the triplet state lifetime (this was shown also in FPs). Under anaerobic conditions do the authors think that RISC by red shifted light will still help?

Yes, we consider that under anaerobic conditions the photoswitching fatigue recovery would in fact improve. As the reviewer points out, a condition with depleted oxygen would reduce the probability of quenching the triplet state limiting the production of chemically reactive species that can react with the chromophore and reduce the fluorescence output. Ringemann, et al., 2008, *ChemPhysChem*, showed a stabilization of the fluorescence signal in some fluorophores under anaerobic conditions. Moreover, an increase in the triplet state lifetime would in principle also be beneficial when considering the photoswitching fatigue recovery since a lower illumination dose could be used to depopulate the triplet state assuming that the RISC efficiency does not change in an oxygen depleted atmosphere. We used the simulation tool to recreate these conditions and approximated a reduction of the oxygen concentration as

Figure R15. Light-induced recovery as a function of the 592 nm power density and the triplet lifetime. We used different triplet lifetimes as a proxy for different oxygen concentrations and simulated the expected light-induced recovery for rsEGFP2 at multiple 592 nm power densities. Increased triplet lifetimes yield a more efficient light-induced recovery, meaning less 592 nm kW/cm² are necessary to access the same recovery levels.

an increase in the triplet lifetime while maintaining all other parameters the same. In this context we monitored the light-induced recovery as a function of the 592 nm power density and the triplet lifetime. The simulation shows that with a triplet state lifetime of 5 ms, $\sim 20 \text{ kW/cm}^2$ are necessary to achieve $\sim 20\%$ recovery, while only 5 kW/cm^2 would be necessary to achieve the same recovery with a triplet state lifetime of 25 ms.

18/ As RISC by red shifted light increases the singlet state population, not only photostability but also effective brightness should increase. Does this show up in the model and have the authors noticed this experimentally? Discussing this point would be important.

From the modelling perspective, since we treat the RISC process similarly to Ringemann, et al. we do observe an increase in integrated photon budget of around 14% at maximum 592 nm co-illumination (Figure R16). Such change is difficult to detect experimentally since we always measure the integrated fluorescence over different ROIs and therefore it will depend on the local concentration.

Figure R16. Normalized integrated fluorescence in the 1st ON/OFF cycle for rsEGFP2 as a function of the 592 nm co-illumination power density. Adding the 592 nm co-illumination can depopulate the triplet state formed in the 1st ON/OFF cycle enhancing the integrated photon budget.

19/ Using photofatigue as a distinguishing feature for multiplexing is a nice idea, but it should be better explained in what situation it is beneficial to use this technique rather than exNEEMO or TMI. Is this to collect the data faster? What are the potential consequences in terms of photo/cytotoxicity? Please discuss these issues.

We acknowledge that, although we thoroughly analyzed various multiplexing strategies, we did not fully integrate these considerations into the main text and instead left them in Supporting Note 10. We have now incorporated this broader context into the main text, addressing, as the reviewer pointed out, the challenges that arise when attempting to achieve kinetic multiplexing with high spatiotemporal resolution.

The text now reads:

When comparing RSFPs, each exhibits a unique switching fatigue profile, therefore we set to explore whether fatigue can serve as an additional multiplexing dimension¹⁰⁻¹³. Multiplexing strategies generically refer to any method that aims to distinguish RSFPs that emit in the same spectral range by analyzing fluorescence signal changes under specific illumination sequences. Methods like LIGHTNING¹¹ and TMI¹² resolve the kinetic rates of light-induced transitions, while approaches such as exNEEMO^{10,13} capture fluorescence intensity changes in response to varying on-switching sequences. Bringing both approaches to high spatiotemporal resolution (e.g., confocal imaging with excitation densities $\sim 100 - 500 \text{ W/cm}^2$) is difficult, since they both rely on the ability to clearly distinguish the off-switching kinetics, either as a fully

sampled decay for LIGHTNING and TMI, or as integrated fluorescence for exNEMO (Supplementary Note 12).

To overcome this limitation, we tested photobleaching dynamics as an additional dimension to aid multiplexing strategies, resolving ambiguities under high illumination irradiance.

20/ The provided software appears to work but is based on Python scripts without any manual. It will be very difficult to use unless such manual is made available. Ideally a Python GUI would be desirable to facilitate the use of the software. I am curious to know whether the authors considered using the SMIS simulator (DOI: Bourgeois, 10.1038/s42003-023-04432-x) as an alternative to their software (and then the authors will not have much doubt about the identity of the reviewer ...) ?

The SMIS simulator looks like a comprehensive and impressive piece of software for simulating many fluorescence microscopy techniques and experiments. We didn't consider the software at the beginning of our project. The main reason is that the simulation of fatigue required very specific cycles of laser illumination sequences that we don't think are easily implementable in other available pieces of software. We are of course open to considering the use of SIMS in the future, especially if there is already implemented a straightforward way to simulate our fatigue experiment, or if simulation could be run with simple hacks. We will be happy to receive any input in that direction.

21/ There are quite a few typos within the manuscript that should be checked ...

We apologize for the typos and have corrected as many as we could identify.

22/ Figure 2f is very noisy: please perform a statistical test (ANOVA) to check that the fitted decay is significant.

The fit is now greatly improved because we adjusted the experimental parameters to better highlight the time dependence of the process. In particular, we have made the illumination times of 488 and 592 nm the same (1 ms) while in the previous dataset, the 488 nm was slightly shorter which introduced more uncertainty in the first two datapoints. The graph has been updated accordingly, now Figure 2g.

23/ Figure 2h legend: I cannot see any green shaded area.

We have now changed the legend accordingly.

24/ Figure 3A legend: fluorescence loss after how many cycles ? Why was the chosen off switching 488 nm dose different for each RSFP, while during a real demixing experiment, all RSFPs will be subjected to the same dose. Please explain. "Different multi-foci" -> "different foci" ? Note that some readers may confuse these "foci" with different "focal planes" and have a hard time to realize that they are all in the xy plane.

We have updated the legend of Figure 3A to report on the number of cycles.

It is important to note that Figure 3A is designed to compare different RSFPs under similar conditions, specifically 488 nm illumination maintained until 80% of the initial fluorescence is switched off, thereby empirically correlating fatigue with off-switching kinetics.

By contrast, Figures 3B and 3C explore fatigue as a multiplexing dimension: here, all proteins are illuminated for the same fixed duration of 488 nm light, regardless of their individual off-switching rates.

Because bleaching profiles vary strongly from sample to sample (see Major Point 13 and Supplementary Figure 24), we perform a fresh calibration under the exact measurement conditions at the start of each multiplexing experiment.

25/ Figure 4b, lower right panel: why showing the mix with and without 592 nm illumination ? It would be more informative to show the two RSFPs with co-illumination.

Figure 4b shows a TMI experiment where the dimension used for unmixing is the off-switching decay of the two proteins. The effect of the co-illumination globally enhances the fatigue resistance of both proteins; this is what the orange cloud wants to highlight.

26/ Supplementary table 4: bleaching yields are in % ? Please check consistency with table S2.

The change has been implemented

27/ Supplementary figure 7b: what is the unit on the Y scale ?

We have now updated the figure.

28/ Supplementary table 6: bleaching quantum yields from ground and excited triplet seem inverted relative to those in table S2 and S4 ?

We have now addressed the mistake in the table.

29/ I am not sure of some of the arguments presented in supplementary note 9: (i) that photobleaching by slower switching RSFPs is expected to be less recoverable by the RISC effect. If there is more triplet formed relative to off switching, the benefit of RISC should be higher, not lower ? Furthermore, non-switchable Green FPs have been shown to be nicely stabilized by RISC (ref 29);(ii) that the ratio between bleaching from the ground or excited triplet state is the key determinant: the ratio between RISC and photobleaching upon photon absorption by the triplet state is also a key determinant.

Regarding point (i) the original discussion about photobleaching and photo-switching kinetics was done considering no 592 nm co-illumination. As stated above, we have now modified the main text to limit predictive comments based on kinetic modeling. Nevertheless, with the proposed kinetic model it can be shown that RSFPs with the same spectroscopical properties and bleaching parameters as rsEGFP2, but with different off-switching quantum yields (QY_{OFF}) display higher photoswitching fatigue at the same 488 nm power density for slower switchers. We simulated this behavior by computing the fatigue fraction after 2000 cycles at 300 W/cm^2 for rsEGFP2-like proteins with different QY_{OFF} . In each simulation the illumination time was adjusted so that the fluorescence was reduced to 80% of the maximum in each cycle, i.e. more light was needed for slower switchers.

The statement regarding RISC and slower switchers can be further clarified by considering Ludvíková et al. (*Nat Biotech*, 2023) on non-switchable fluorescent proteins. Ludvíková et al. identify the cut off for red-light recovery at 470 nm illumination at intensities below approximately 1 kW/cm^2 for green FPs (Fig. 1g in the referenced publication). Nevertheless, we clearly observe recovery in rsEGFP2 and other green rsFPs under power densities around kW/cm^2 and upper in our kind of experiments. The reason lays in the different nature of experiments, i.e. in our cyclic sequences off-switching shields part of the population from the triplet state. In this simplified picture, slow-switching rsFPs converge toward non-switchable FP behavior, which can partially explain why red-light recovery becomes less pronounced as switching slows.

Regarding point (ii), in line with Reviewer #1's suggestion, we have refocused the manuscript on the descriptive aspects of our model. We have therefore removed some of the previously included considerations that were more predictive than descriptive.

Figure R16. Fatigue fraction as a function of the off-switching quantum yield (QY_{OFF}). We simulated the photoswitching fatigue response of rsEGFP2 with different QY_{OFF} while keeping all other spectroscopic parameters the same. The fatigue fraction increases when reducing the QY_{OFF} ($QY_{OFF} < 1.65\%$, from El Khatib, M., 2016, *Sci. Rep.*) as a higher fraction of triplet state is formed during the off-switching illumination time. Moreover, the triplet state can greatly absorb 488 nm light (Rane et al. 2023, *J. Phys. Chem. B*) which induces further photobleaching effects. On the contrary, increasing QY_{OFF} ($QY_{OFF} > 1.65\%$) reduces the 488 nm illumination time necessary to off-switch a significant fraction, reducing the expected photoswitching fatigue fraction.

30/ supplementary figure 16: do these data support the notion that the long-lived dark state downstream the triplet is possibly also bleached by 405 nm or 488 nm light ?

Supplementary Figure 16 is consistent with the observation made for rsEGFP2, which we have now described in the response to Major Point 4 and the newly added Supplementary Note 5.

Reviewer #2 (Remarks on code availability):

Below I paste point 20/ in my review:

The provided software appears to work but is based on Python scripts without any manual. It will be very difficult to use unless such manual is made available. Ideally a Python GUI would be desirable to facilitate the use of the software.

In conclusion for this part, at this stage I consider that the code is not a usable resource for the community.

We thank the Reviewer for his comments regarding the software. We agree that the provided Python scripts are not intended as a general-purpose tool comparable to other publicly maintained software available. Rather, the code is shared primarily for transparency and reproducibility of the analyses presented in this work. At this stage, the development of a dedicated graphical interface lies beyond the scope of our current work. That said, we recognize that brief documentation could help others using the scripts, and we have now added a user manual to the repository.

**Point-by-point responses to reviewers of the manuscript NCOMMS-25-05441:
All-Optical Strategies to Minimize Photo-Bleaching in Reversibly Switchable Fluorescent
Proteins**

Marin-Aguilera et al

We thank the reviewer for the extensive feedback and the constructive discussion provided in this second round of revision. We have included further validation and refinement of the kinetic model, together with clearer explanations and additional references in both the main text and the Supplementary Information. In particular, we have examined the role of specific pathways involved in the RISC transition, rationalized the principal mechanisms underlying off-to-on bleaching, and provided additional experimental evidence supporting the fluorescent nature of the intermediate state incorporated in our minimal kinetic model. The text has been revised following the corrections suggested by both reviewers, and all changes are highlighted in green in the main text.

In the following section, we provide a detailed point-by-point reply to the reviewers' comments, with our responses in black and the reviewer comments in blue.

Reviewer #1 (Remarks to the Author):

Review of NCOMMS-25-05441A, "All-Optical Strategies to Minimize Photo-Bleaching in Reversibly Switchable Fluorescent Proteins," by Marin-Aguilera et al.

I thank the authors for their generally positive responses to my prior comments, which I feel have improved this interesting paper substantially. In particular, I applaud their approach of creating a generalized model that does not rely heavily on literature parameters. This model in general makes the paper considerably more convincing.

We thank the reviewer again for the careful and thorough analysis of our work, and for providing important points of reflection that have helped us to strengthen the manuscript.

There is one major issue that I think still needs more work. First, as to whether the RISC drives the system to S_0 or a higher singlet state, I either misread reference 29 of the original paper or gave the wrong reference number. Rob Dickson's group has clearly found molecules for which RISC to an excited singlet state is the dominant pathway. That being said, in the work of his that the authors cite to substantiate this effect in fluorescent proteins, the protein that was used was specifically engineered to have this behavior, so I do not find this argument to be compelling. I am further concerned that the model used cannot differentiate between RISC to S_0 and a higher singlet state. These two processes must have different outcomes in the scheme of the authors. The fluorescence quantum yield of this protein is 0.3, according to Grotjohann et al. (DOI 10.7554/eLife.00248). Of course, some of the other 70% comes from nonradiative relaxation, but presumably a substantial fraction arises from ISC. If RISC repopulates the ground state, then no additional fluorescence can be generated until the next excitation pulse. If RISC populates a higher singlet state, then some population will be cycled back to the triplet state, where it can again have

a chance to undergo RISC while the redder light pulse is on. I would therefore expect that the amount of fluorescence observed would be substantially higher for RISC to an excited singlet state than for RISC to the ground state. This picture supports what the authors propose, I think, and could be used as an argument for it. But the fact that their model does not indicate that the two different pathways could be distinguished is worrisome. It could simply be related to their range of parameters. Still, the authors should at very least test some kinetic parameters that ought to show a difference and make sure that a difference is seen. For instance, with a 5% quantum yield for fluorescence and a 95% quantum yield for ISC, RISC to a higher singlet state should give far greater total fluorescence than should RISC to the ground state.

We agree with the reviewer that RISC to S_0 or to S_1 can introduce differences in systems where this pathway has substantial quantum yields. In the range of parameters of rsEGFP2, RISC to S_0 or to S_1 do not influence meaningfully the measurable fluorescence, mainly because ISC and RISC are not very probable when compared to the other pathways.

The model is indeed able to reproduce the different effects that RISC to S_0 or S_1 would produce, for example the optically activated delayed fluorescence (OADF), i.e. fluorescence induced by excitation of the triplet state with consecutive RISC to S_1 . That is highlighted in a OADF experiment similarly to the one proposed by Peng et al. (*J Phys Chem B*, 2021), a burst of high intensity 488 nm (5 μ s, 50 kW/cm²) will generate substantial triplet state, right after, the 592 nm illumination starts (\sim 1 ms, 50 kW/cm²), and as shown in the simulation in Figure R1 fluorescence signal is only detectable if RISC happens to S_1 . The magnitude of the OADF fluorescence signal is $\sim 10^4$ times smaller than the fluorescence signal coming from the excitation wavelength (Figure R1a). If, on the other hand, we simulate a rsEGFP2-like protein with extremely high ISC, i.e. 95%, the magnitude of the OADF fluorescence photons become much more significant, and these photons would only be detectable in the case of RISC to S_1 (Figure R1b). In the case of rsEGFP2, the ISC yield is ~ 0.3 % (Rane et al., *J. Phys. Chem. B.*, 2023) which makes determining the pathway for RISC through OADF rather difficult.

Figure R1. OADF measurements with a short 488 nm pulse followed by a longer 592 nm excitation and tracking of the fluorescent signal throughout the pulse. (a) rsEGFP2 model investigated in the paper and (b) FP optimal for OADF with 95% ISC. The RISC pathway to S_0 (blue line) or S_1 (orange line) does not sensibly affect overall fluorescence in rsEGFP2.

In our experiments, we use very different pulse schemes to those optimized for revealing OADF, for instance, ~ 300 W/cm² and 0.9 ms of 488 nm to off-switch rsEGFP2 in continuous illumination, and 592 of 1 ms and 50 kW/cm² as the recovery illumination pulse. Under these experimental conditions, the OADF photons that would appear from RISC to S_1 are negligible when compared

to the bulk of fluorescence photons, i.e. fluorescence traces from RISC to S_0/S_1 look indistinguishable (Figure R2a). In our pulse scheme, as we are continuously illuminating with 488 nm light and the majority of the fluorescence is stimulated by direct excitation of S_1 with 488 nm light. The transient population of S_1 and T_1 are also minimally affected by RISC to S_0 or S_1 as shown by Figure R2b.

Figure R2. Off-switching-like measurements with 592 nm co-illumination and longer duration respect to the OADF-like experiment. (a) Evolution of the signal for RISC to S_0 (blue curve) and S_1 (orange curve). (b) The time evolution of S_1 (magenta curve) and T_1 (green curve) are extracted for RISC to S_0 (dotted line) and RISC to S_1 (solid line). The inset report on a small interval of the decay to highlight the magnitude of the difference between the two cases.

Some more minor points:

1. The authors continue to make errors of nomenclature that will potentially cause readers to doubt other aspects of the work unnecessarily. For instance, they still call 405-nm light “ultraviolet.” This designation is an improvement over UV-vis, but 405-nm light is violet (or indigo, if one wishes). The UV spectrum starts at 400 nm. This designation should be fixed. They have also continued to misuse the term “red-shifted.” I agree with them that this term is used often in spectroscopy, but it is used primarily to describe spectra, and secondarily to describe tunable lasers, and never to describe lasers with different wavelengths. It is not synonymous with having a longer wavelength. At a recent conference I queried a half dozen spectroscopists, in a neutral manner, regarding whether they would call a laser with a longer wavelength than another “red-shifted.” The unanimous answer was no, and I got several puzzled looks for asking this question. I strongly recommend that the authors change this language. So the use in line 51 of the paper, for instance, is fine, but most of the other uses are not.

We have now changed both instances in the text and supplementary.

2. There are still numerous unclear antecedents in the paper that make for more difficult reading:

Line 29: “them,” should be “RFSPs”

Fixed.

Line 55: “it” should be something like “this technique”

Fixed.

Line 58: “the one” do you mean “those?”

Fixed.

Some other small things I found as I was reading the manuscript:

3. Line 56: Please use something more accurate than “red-induced photobleaching”

We have now aligned the definition to the one used by the authors of the referenced paper, Ludvikova et al.

4. The word “fluorescent” is used often when the correct word is “fluorescence.” There is no such thing as a “fluorescent photon,” for instance. Instead of “fluorescent signal,” “fluorescence” is more accurate.

This issue should be fully addressed now.

5. Line 78: “a decrease of 80%” in what?

Fixed.

6. Line 83: “fatigue fraction” is not defined

It is defined in Line 81-82. “a steady decrease over thousands of cycles, which we refer to as the fatigue fraction”.

7. Line 84: “488 nm” should be “488-nm illumination.”

Fixed.

8. Line 86: “405 nm light” should just be “405 nm”

Fixed.

9. Line 91: “crystallography” should be “crystallographic”

Fixed.

10. Line 115: “low” has no quantitative meaning, just say “for 405-nm power densities below 300 mW/cm²”

Fixed.

11. Line 127: “RFSP” not “RFSPs”

Fixed.

12. Line 128: What is “it?”

Fixed.

13. Line 134: “trapped in TH” not “trapped in the TH”

Fixed.

14. Line 136: Checking whether delivering the same dose over different times is known as a reciprocity study. What you are observing is a form of what is known as low-intensity reciprocity failure (LIRF), which is a clear sign of nonlinearity in a chemical system. I mention this in case you want to cite some relevant literature, but that is up to you.

We thank the reviewer for this suggestion.

15. Line 137: “delivered by progressively” not “delivered progressively”

Fixed.

16. Line 145: This sentence is hard to read. I’d recommend starting with “Reverse intersystem...” and ending with “by preventing photobleaching”

We have now rephrased the sentence.

17. Line 149: “compared” not “respect”

Fixed.

18. Line 151: “in” rather than “on”

Fixed.

19. Line 153: Remove “rate” because you later in the sentence talk about a timescale, which is the inverse of a rate

Fixed.

20. Line 163: “increasing by up to” not “increasing up to”

Fixed.

21. Line 179: Laser lines by definition come from the same laser; for instance, an argon ion laser has several different lines. Use “all the different laser wavelengths explored” instead of all the different laser lines of the experiment”

Fixed.

22. Line 180: There is no such thing as the ground state of a triplet unless the ground state of the molecule is a triplet (molecular oxygen, for instance). Say something like “both T1 and higher triplet states contribute”

Fixed.

23. Line 200: “reaches a plateau at 488 nm of more than 600”

Fixed.

24. Line 201: “power densities of more than 10”

Fixed.

25. Line 215: “This” what?

Fixed.

26. Line 217: Doesn’t this sentence say the same thing twice? Strong absorption promotes excitation? I would remove “promotes triplet-state excitation and” and replace “strong triplet absorption” with “strong absorption of the T1 state” for a little bit of additional clarity

Fixed.

27. Line 226: What is “they?”

Fixed.

28. Line 230: “illumination irradiance” is redundant. The adjective “high” goes better with irradiance, but if you want to use illumination couple it with “strong”

Fixed.

29. Line 239: Define FOV as field of view, i.e. “over an extended field of view (FOV)”

Fixed.

30. Line 248: What is “their?”

Fixed.

31. Line 249: “This” what?

Modified.

32. Line 256: Replace “such” with “this”

Fixed.

33. Line 260: “criterion” not “criteria”

Fixed.

34. Line 265: What is a photobleaching constant? If this was defined I missed it.

Fixed.

35. Line 267: I do not know what “Attaining these results” means

Modified.

36. Line 272: Remove “a timelapse of”

Fixed.

37. Line 274: “extends the absolute observation window for the multiplexing imaging by 20% to 30%”

Fixed.

38. Line 281: “and” should be “or” and there is no need for “the” before “photoswitching”

Fixed.

39. Line 287: Remove “an”

Fixed.

40. Line 294: “this” what?

Fixed.

41. Line 300: Delete “looking”

Fixed.

42. Line 307: “rationalizing our observations with an effective”

Fixed.

43. Line 315: “became” should be “becomes”

Fixed.

44. Line 318: What is “it?”

Modified.

45. Line 322: Same question

Modified.

46. Line 327: This is not a sentence

Modified.

47. Line 330: Remove “seemingly,” you are not saying its conserved among all of them necessarily

Fixed.

48. Line 333: “red light” not “red-light”

Fixed.

49. Line 336: I’d suggest starting with “There is a trade-off between spatiotemporal resolution and photostability in any live-cell-imaging application”

Modified.

50. Line 337: What is “they?”

Modified.

51. Line 341: “wavelength for the excitation of orange- and red-emitting fluorophores” and use “this wavelength” instead of “it”

Fixed.

52. Line 351: You’ve already defined PAA, and use it in the line above to boot. Just say “in PAA gel”

Fixed.

53. Line 355: “#” not “No”

Fixed.

54. Line 358: “characterization” not “a characterization” and “off-and-on-switching” or “off- and on-switching”

Fixed.

55. Line 360: Use × and appropriate spacing, as you do elsewhere in the manuscript

Fixed.

56. Line 371: Remove “accordingly”

Fixed.

57. Line 375: “with a 1-ms 405-nm”

Fixed.

58. Line 398: What does “1.40–0.70-NA” mean? Please elaborate.

We had reported both numerical aperture of 1.4 and an iris diaphragm of 0.7, we now corrected for a more conventional notation.

59. Line 399: What does “modulable” mean? Please elaborate. Intensity? With time? Please clarify every time you use this word.

We have now replaced it with “modulated”, since we meant that the lasers are capable of high-speed direct modulation of the output power.

60. Line 400: Cobolt is a model, please acknowledge Hübner Photonics as you do for other manufacturers here and throughout the rest of the paper

Fixed.

61. Line 414: “in which” rather than “where”

Fixed.

62. Line 416: Use \times

Fixed.

63. Line 422: What is “MPD?” A manufacturer? Please clarify as for other pieces of equipment.

Fixed.

64. Line 426: “gold-bead” not “gold beads”

Fixed.

65. Line 432: “at a 80 MHz”

Fixed.

66. Line 442: What is “it?” and at very least use “its” rather than “it’s”

Fixed.

67. Line 445: “both 810 and 900 nm”

Fixed.

68. Line 446: Use \times

Fixed.

69. Line 449: “two APDs”

Fixed.

70. Line 453: “into” rather than “on”

Fixed.

71. Line 485: “resolution”

Fixed.

72. Line 486: “was” rather than “were”

Fixed.

73. Line 490: “confined” rather than “confinement”

Modified.

74. Line 491: “from the” rather than “of the”

Fixed.

75. Line 492: “to split the light equally into P and S polarizations”

Fixed.

76. Line 494: “SiO₂”

Fixed.

77. Line 495: What begins at the end of this line is not a sentence. Start with “The ± 1 diffraction orders”

Modified.

78. Line 519: “slow image, which consisted”

Fixed.

79. Line 520: “the other, which we named fast image, consisted”

Fixed.

80. Line 546: “between adjacent foci”

Fixed.

81. Line 557: GMM was already defined and delete “that fits the data into a given number of Gaussian components, which you already stated.

Fixed.

82. Line 559: “proteins. Afterwards” and replace “Gaussian mixture model” with “GMM”

Fixed.

Figure 1: As mentioned above, “fatigue fraction” is never really explained adequately, and the right side of panel (a) does not help. “fatigue fraction” is pretty well hidden in the color legend. Why doesn’t the color legend start at zero?

As in point 6, fatigue fraction is defined in the text in lines 81-82. We have now modified the panel mentioned. The color legend does not start from zero to not flatten and compress that axis.

Figure 2: Right side of panel (a), normalized to what? In panel (g), the x-axis numbers do not line up with the ticks. What does it mean to have an error bar that goes below zero? Panel (j), how are the dashed lines exponential fits???

On Figure 2a, the signal was normalized to an initial pulse with only 488 nm illumination which represents the true concentration of fluorophores in the interrogated volume. We report it in the methods section. The other aspects are now corrected in the figure and legend.

Figure 3: Last line of caption, “when the 592-nm pulse follows 1 ms after the 488-nm pulse”

Fixed.

Comments on the SI

I admittedly did not read this as carefully as the main paper. The font in the supporting information is quite small. I recognize that the document is already quite substantial in length, but it would be helpful to readers to use a larger font and smaller horizontal margins nonetheless.

We have adapted the supplementary to a more readable formatting style.

Page 8, paragraph 4: What is a “raising time?” The dashed line is clearly not what would normally be called a stretched exponential, which is a decaying function. Please elaborate. This issue probably relates to my question on Figure 2. Also, please report the value of τ and give a physical interpretation if possible.

As a reply to reviewer #2 we have also updated the fitting.

Page 10, Figure 6: Why don't the y-axes start at zero?

The axis reports the range of parameters that we have explored, that happen to start at 0.05.

Page 11, end of first paragraph: “since this phenomenon” rather than “Since it”

Modified.

Page 13, Figure 9: Please use broken scales for the y axes

Fixed.

Page 14, Figure 10b: Can you spread the y scale on the residuals a bit so the numbers don't overlap?

Addressed.

Page 16, last paragraph, first sentence: “Complementarily”

Fixed.

Page 18, last paragraph: Maybe just clarify slightly “The quantum yield for RISC from a given higher triplet state in a specific solvent should be conserved...”

Fixed.

Page 19: Same comment about the residuals in both figu res.

Fixed.

Page 24, Figure 10: Do you really observe negative recovery at low NIR intensities?

The recovery values are obtained by subtracting the signal with and without red light. The large experimental error for this specific dataset, collected in a confocal setup, results in apparent negative values. The background subtraction procedure also explains why negative values are possible in the dataset discussed in the following point.

Page 25, Figure 20: in (a) Normalized to what? Just one statement somewhere in the paper will suffice. In (b), same question regarding negative recoveries

When we report the fatigue curves the normalization is generally to the signal of the first cycle as we describe in the material, otherwise it is stated in the caption of the figure.

Page 27: I don't understand the last sentence of the second paragraph

We have now rephrased the sentence.

Page 33: It's probably not necessary to give all focal lengths and part numbers

We follow the conventional way we report optical set-ups.

My apologies for the length of this review. Most of the points raised are minor, but are still worth addressing. I do want to see the results of the checks of the kinetic modeling, but overall I like this paper quite a lot and think that it is near being ready to publish.

We sincerely appreciate the time and effort you dedicated to reviewing our paper. Your constructive feedback has helped us improve its depth and clarity.

Reviewer #2 (Remarks to the Author):

The authors have performed extensive revisions of their paper, which answer many of my previous concerns. The manuscript still largely focuses on the photophysical model that underlie the nice experimental results. I think it is adequate to substantiate these results by a restricted, meaningful photophysical model of rsEGFP2. The very extensive discussion on the photophysics is particularly interesting for photo-physicists but is somewhat overwhelming the main message of the paper about light induced recovery through RISC.

There are three major points that I believe should be addressed before the manuscript can be published, as well as a number of more minor points that I encourage the authors to consider.

We thank the reviewer for the positive feedback and further insights to strengthen and clarify the presented work.

Major points:

1/ Although I understand the authors viewpoint that RSFPs should be submitted to illumination schemes matching their photophysics and suitable for imaging experiments, I feel that, from the viewpoint of fundamental photophysics, the paper is still misleading: while a constant dose of 488 nm light is used, a varying dose of 405 nm light is employed. It is then difficult to compare the effect of each laser to the photofatigue decay. One main message of the paper, coming early in the manuscript, is that most of the damage is caused by the 405 nm laser (Fig 1b), while the 488 nm laser is relatively immune to photobleaching (Fig 1c). From the viewpoint of the photophysicist, the comparison is unfair, as the 488 nm laser is at constant dose, which is not the case for the 405 nm laser. Furthermore, the other main message of the paper is that RISC can be used to substantially slow down photofatigue, while the triplet state is always reached, according to the presented model, from the 488 nm laser. Thus, there is objectively very substantial photobleaching by 488 nm light, and there is an apparent contradiction. These aspects could really confuse the readers. The lack of clarity extends for example to the comparison of RSFPs with normal FPs: I think RSFPs do not react differently than FPs regarding T1 and the RISC effect: increasing 488 nm power density at constant dose with EGFP is very likely to produce the same increase in RISC efficiency, and there is no real opposite behavior between the findings here and in ref 32.

All this is mostly a question of presentation. I would recommend the authors to (i) precisely state what doses / intensities are used for each laser (at present, there is confusion at many places in the paper, e.g. in the legend of Fig 1), (ii) make sure not only the super-resolution imaging community, but also the photophysics community perceive the consistency of the paper, (iii) clarify the different sources of photobleaching (405 versus 488 nm light), based on the use photophysical model, possibly with the help of an additional figure in the main.

We thank the reviewer for pointing out this confusion. The intention of the work is not to define a dominant wavelength but rather to provide a comprehensive analysis of the bleaching profile of rsFPs and we do not want to restrict this work only to the imaging community but rather using this to open a discussion across the communities that revolve around FPs and RSFPs photophysics. Therefore, we have followed the suggestions of the reviewer as follows.

(i) We have removed any statement that would misleadingly present the work as a dichotomy of 405 nm and 488 nm. We now put emphasis on the pathways. When the system is in the on-state, bleaching will predominantly arise from triplet mediated pathways (i. triplet bleaching in Figure

1e), whereas the system in the off-state will induce bleaching mainly from the cascade of states in the off-to-on process (ii. Off-to-on bleaching in Figure 1e). 405 and 488 nm illuminations are involved in both on- and off-switching processes and the two simplified bleaching pathways, although with different efficiency. The main changes are in the Main Text, especially in the introduction and first section of the results (highlighted in green).

(ii) We have updated figures and figure legends to state or make clearly visible the experimental conditions with the aim of minimizing confusion. The drawing in Fig. 1a has been extended to provide more depth on the dependency between illumination wavelength and RSFP's response.

(iii) We have performed and included a symmetrical screening of the bleaching profile for the 405 and 488 nm power densities, to provide the reader with a clear comparison, now Supplementary Figure 1. While the 405 nm power ramp is identical to the one reported in Fig. 1b for the 488 nm we have kept the illumination time constant at 1 ms and varied the power density across two orders of magnitude 0.02 to 2 kW/cm². It is important to note that the decision on the illumination time will always be arbitrary, in this case we have decided for 1 ms as the time that provide an 80 % decrease of fluorescence for the middle power recorded and therefore incomplete off for the power

Figure R3. Photoswitching fatigue characterization. (a) Schematic of the photoswitching fatigue experiment. rsEGFP2-embedded PAA gel is repeatedly illuminated with 1 ms 405 nm and 1 ms of 488 nm light, with an interval of 1 ms between them. An interval of two orders of magnitude in power density is explored for both wavelengths, from ~ 0.01 to ~ 1 kW/cm². (b) Moving the 405 nm illumination across this range result in the complete on-switch of the RSFPs fraction, (c) while the increased 488 nm speed up the off-switch kinetics. The grey area in the graph corresponds to the illumination time and consequently camera integration time used for the photoswitching fatigue recording. The set time of 1 ms matches the 80 % decrease in fluorescence for the middle power density. Photoswitching fatigue recorded respectively at varying (d) 405 nm and (e) 488 nm (mean $\pm \sigma$ of at least 3 measurements). The upper graphs are normalized to the fluorescence of the first cycle, while the bottom one is normalized to the second cycle to better visualize and compare the fatigue fraction upon the different illuminations.

below and longer than the off time for the one above. A different integration time would surely modify this overall trend. The 488 nm dependency has been explored at a level of 405 nm for which we have seen no severe bleaching (0.05 kW/cm^2), in the attempt of isolating the two effects, nevertheless as investigated throughout the work, the two are intrinsically dependent, since the irradiation at 405 nm will define the concentration of RSFPs excited at 488 nm.

2/ Restricting photobleaching induced by 405 nm light to the TH species sounds inconsistent. This is because, as the 405 nm dose is increased, it is more and more the CHint species that gets populated, as can be seen on figure S3b. Since CHint strongly absorbs 405 nm light, most 405 nm photons absorbed by the (TH, CH+, CHint) trio are increasingly due to CHint absorption. In addition, if CHint is indeed fluorescent - see my next major point -, its excited state lifetime might be greater than that of TH, possibly opening the door to triplet state excursion from this state (although it is unclear whether CHint would be fluorescent under 405 nm excitation). Overall, I think photobleaching from CHint vs TH must be discussed, may be reworked, and at least the choice to only consider bleaching from TH must be carefully justified. Perhaps considering direct photobleaching from CHint in addition to from TH would not change significantly the outcome of the simulations.

We agree with the reviewer that we needed to better clarify the nature of bleaching from 405 nm light. In this regard, tracking the population of the intermediate (C^{Int}) and TH during 405 nm illumination suggests the predominance of bleaching from the intermediate state C^{Int} , as pointed out by the reviewer (Fig. R4, now Supplementary Figure 8). During 405 illumination the TH fraction is reduced as consequence of the off-to-on transition, however, due to the long lifetime of

Figure R4. (a) Evolution of the ground state population during the 405 nm illumination period. (b) Integrated population of the excited state TH* and C^{Int}* during the 405 nm illumination and at increasing 405 nm. (c) Comparison of the revised model (green dotted line) respect to the previous model (yellow dotted line).

the intermediate state C^{Int} , this state is very prone to excitation by violet light. Given the fluorescent nature of C^{Int} , we expect a long excited state lifetime similar to the one of the on-state (~ 1 ns C^{Int*} vs 20 ps TH^*), this is also mapped in the higher expected excited state population during 405 nm illumination for C^{Int} compared to TH^* .

We have now revised the model to incorporate a constant rate of irreversible bleaching from the excited states of TH , CH^{\dagger} and C^{Int} . In this way, the longer the lifetime of a state the more induced bleaching, effectively making C^{Int} the major source of bleaching from the on-to-off 405 nm illumination. As the reviewer hinted, this does not change the outcome of the simulation (we report an example in Fig. R4c) but better describes the involvement of the different states according to the experimental observation.

The new changes are reported in:

- Supplementary Note 1.
- It influences also plots in Supplementary Note 4 (specifically Supplementary Fig. 7 and Table 4), Note 5 (specifically Supplementary Fig. 10 and Table 5) and Note 7 (specifically Supplementary Fig. 15, 16 and Table 7).
- In the Main text we have updated Figure 1d, 1f, 2f and 2g.

3/ In evaluating the fluorescence of the same intermediate CH_{int} (supplementary note 3), the authors ignore the effect of triplet state built up and relaxation induced by the initial 488 nm pulse. This relaxation results in a significant rise of fluorescence with a timescale corresponding to the triplet state lifetime, as detailed in supplementary note 7. The authors cannot ignore triplet state relaxation in supplementary note 3. I urge the authors to revisit the notion that CH_{int} is significantly fluorescent. If the conclusion is maintained, and the extinction coefficient of this species at 488 nm is still high, then it must be that CH_{int} exists as an equilibrium between an anionic and a protonated form. As this is a new finding as compared to published literature, some discussion should be added about the nature of this intermediate - which can probably not be called “ CH_{int} ” –

We performed the experiment reported in Figure R5 (Supplementary Figure 4) in absence of 405 nm illumination to investigate the relevance of the recovery from triplet state relaxation. Although a slower and smaller recovery is observed (blue line and experimental data point in Fig. R5a), it does not account for the full recovery observed in the experiment with the 405 nm light pulse. Remarkably, the experiments without 405 nm light do not exhibit any fluorescence over background for short waiting times (up to about 0.1 ms), whereas when adding the 405 nm light pulse, substantial fluorescence is observed even at the smallest waiting times (red, orange, green curves when delays are less than 0.1 ms). Fluorescence at short waiting times suggests that an intermediate state formed right after the 405 nm light pulse is fluorescence. A biexponential fit was used to reproduce the observed recovery with a slower component matching the recovery without 405 nm light, and a faster component of about 800 μ s (Fig. R5b).

To disentangle more clearly the recovery of the triplet population from the maturation of the fluorescent states arising in the on-switching pathway, we report a second experiment recorded on our custom confocal microscopy system. The experiment is performed on a solution of rEGFP2. The pulse sequence uses a long circularly polarized pulse of 488 nm light (3 ms, 55 kW/cm^2). After 1.5 ms the fluorescence signal reaches a stable baseline and a short pulse of linearly polarized

405 nm light (250 ns, 235 kW/cm²) is delivered. The sequence is repeated several times on the same spot. In this experimental conditions (i) the triplet state population will reach a steady state population, slowly decreasing only because of bleaching, (ii) there is no waiting time in between on-off pulses and the triplet population cannot recover from the steady state population as in experiments with dark waiting times, (iii) a rise of the fluorescence signal is observed right after the 405 nm light pulse (time constant of about 5 μs).

Figure R5. (a) Integrated fluorescence signal as a function of the delay between 405 and 488 nm pulses at different 405 nm power densities. Even with no 405 nm illumination (blue line), there is a raise in signal with a characteristic time $t_2 = 2.37 \pm 0.40$ ms, that we assign to the triplet state. The data is fitted with a raising monoexponential function, $y_2 = a_2(1 - \exp(-t/t_2))$. As the 405 nm illumination is introduced, the signal at shorter delays increases and a faster component also appears in the raising signal. (d) Normalized fluorescence captured as function of delay between 405 and 488 nm pulses at a 405 nm power density of 40 kW/cm². For visualization purposes, the data and corresponding fits have been normalized between 0 and 1. A biexponential function was necessary to fit the data. The slower component in blue, corresponds to the relaxation of the triplet state, while the faster component we attribute to C^{Int} . The characteristic time of the faster component is $t_1 = 0.82 \pm 0.45$ ms, close to the slow relaxation processes observed in TA measurements (Woodhouse et al, *Nat Comm*, 2020 Uriarte et al, *J. Phys. Chem. Lett*, 2022).

Figure R6. Fast dynamics of the fluorescence signal upon on-switching for rsEGFP2 in solution. Over a constant 3 ms 488 nm illumination, 405 nm is delivered to the sample as a 250 ns pulse and the fluorescence continuously recorded with a time resolution of 1 μs.

Figure R7. Fluorescence anisotropy signal for rsEGFP2 fixed on glass. The laser sequence and temporal resolution correspond to the one depicted in Fig. R6.

Furthermore, using a polarization sensitive detection with two APDs and a polarizing beam splitter we measured the anisotropy of the fluorescence after the impulsive 405 nm light on protein anchored on coverglass (static and rotational diffusing). Recovery of triplet state generated by the 488 nm light would not have an orientational preference because the 488 nm light is circularly polarized. Instead, we observe substantial anisotropy of the fluorescence above background that rises after the 405 nm light pulse (Fig. R7). This adds additional proof of the causality relationship between the 405 nm light pulse and the fast rise of the fluorescence signal.

In the main text we changed the label of the intermediate state from CH_{Int} to C^{Int} , following the statement of the reviewer suggesting that a protonated cis-intermediate state would be most likely be non-fluorescent. Thus, the more generic label does not make strong assumptions on the nature of the state. A statement on the possible anionic nature of the fluorescent intermediate state was also added to the main text and read as “*The detectable fluorescence suggests it relates to an anionic form of the chromophore and places it as the major gateway for bleaching pathways that stems from the on-switching process*”.

Minor points:

L37: radii

Corrected in the text.

L49-51: awkward statements: “which presents a peak around 488 nm, the same wavelength used to elicit the fluorescent photon” is in contradiction with “Given that the long-lived triplet state of FPs has a red-shifted absorption spectrum compared to the fluorescent form”

We have now restructured the sentence to avoid contradiction.

L61: a “comprehensive understanding”: this statement is in contrast with the notion that the authors use a “minimal model”.

We addressed the contrast by changing the adjective for “mechanistic”.

L80: “fluorescence signal” rather than “fluorescent signal” (this comes at several places in the manuscript)

We addressed this throughout the main text and supplementary information.

L83/84: “Similarly, the fatigue fraction also increased with higher 405 nm illumination, while showing a minor dependency on 488 nm.” In relation to my major point 1, this sentence is misleading.

We have now restructured the full paragraph.

L104/105: It looks like the identified intermediate is new relative to literature. The sentence reads quite mysterious at this early stage in the manuscript. I think the point should come later, and deserves more comments in the main manuscript if fluorescence of CHint is confirmed.

We have now reported more assertively the fluorescent nature of the C^{Int} and expanded in the discussion how this finding relates to current literature.

L143/144: it would be fair to state right away that the penalty is an increase in the data collection time.

In the Result section we limit ourselves to experimentally prove the modulation of the initial drop of the fluorescence signal, while the cost of the strategy is clearly addressed in the Discussion section when this phenomenon is reported in the context of imaging. We believe it would add a repetition.

L163: “suggesting that the effect is driven by an absorption-mediated process”: what else could it be ? This reads like a trivial sentence.

We have removed this sentence.

L165: 405 nm ?

This sentence describes the experiment with the 592 nm co-illumination; therefore 592 nm is the wavelength that is removed in the control experiment.

L208: to me, another possible scenario is that red-shifted wavelengths promote RISC more than photobleaching, as compared to stronger wavelengths. Naïvely speaking, lower energy photons are more gentle. (But I understand that Reviewer 1 has arguments against this possibility)

We agree with the reviewer that this could also be a valid interpretation guided by experimental common sense. We are also supported by the view and literature suggested by the first reviewer, which points to reduced RISC efficiency at longer wavelengths. The sentence in the main text read “*The increased recovery at ~ 900 nm is attributed to both stronger triplet-state absorption and*

reduced crosstalk with other photobleaching pathways”, and it implies that less photobleaching is expected by 900 nm light. We think it does not conflict with the statement of the reviewer.

L216-219: Here the authors write in clear that 488 nm light illumination is a major cause of photobleaching through ISC. After looking at Fig 1c, readers might get lost.

We believe the changes introduced in response to the first major comment now clarify this aspect. Photobleaching is not caused by one wavelength alone, but rather by the specific balance of 405 nm and 488 nm illumination that determines the population of the different states. At high 405 nm intensity, most RSFPs are in the on-state, and thus 488 nm excitation significantly populates the triplet state compared to conditions where only a small fraction of RSFPs are on.

In the section highlighted by the reviewer, we discuss the experimental trend observed when comparing bleaching rates of faster and slower RSFPs. In this comparison, the 405 nm dose was always adjusted to the saturation level of the specific protein. Therefore, the observed differences arise from the distinct off-switching yields of the proteins and the resulting competition between off-switching and triplet formation.

L284/285: I do not understand why the majority of the triplet state population would reside in the center of each focal spot. The ratio ISC vs switching is the same everywhere, so the donut beam will also generate substantial triplet state?

The underlying reason lies in the Gaussian profile of the 405 nm on-switching beam and its spatial displacement relative to the 488 nm off-switching maxima. When operating below saturation, the 405 nm Gaussian profile imprints a corresponding on-switch spatial distribution (Fig. R8). This defines the region where the triplet state will be more populated, indeed for low 405 nm illumination the bleaching fraction is greatly reduced. Consequently, an overlapping Gaussian profile at wavelengths above 592 nm is sufficient to depopulate the triplet state and thereby rescue the majority of RSFPs from bleaching during imaging.

Figure R8. Spatial profile of the laser beams and on-switch fraction. (a) On-switch curve where intensity is mapped on the relative on-switch fraction and (b) point spread function for the key wavelengths involved in the imaging sequence: on-405 nm, off-488 nm and rescue beam-592 nm. The spatial profile of the on-switch fraction is plotted relative to the level identified in (a).

L302/303: I encourage the authors to comment about ROS generation and the possibility/absence of reduced phototoxicity upon 592 nm co-illumination, as observed in ref 32.

Our main concern regarding phototoxicity was whether 592 nm co-illumination, being closer to the visible range than 900 nm, could induce additional bleaching. We did not observe such an effect under the conditions tested. However, our assays were not specifically designed to evaluate phototoxicity relief, i.e. our control mitochondria mobility experiment is designed to be in a condition of low phototoxicity/healthy state, thus we argue that relief would not be easily detected. Since our hypothesis involves the same pathways and mechanisms described in ref. 32, where phototoxicity was directly assessed through dedicated experiments that have as control phototoxic conditions, we believe their conclusions are applicable to the power scheme and intensities used in our study. For this reason, we extensively cite and refer to the work of Ludvikova et al.

L327/328: please check grammar of this sentence.

We have now rephrased this sentence.

L342/343: perhaps specify that special optics compatible with near-IR are then needed.

We incorporated this specific aspect.

Supplementary materials:

Table S2: the on to off switching yield of rsEGFP2 from ref 6 is obsolete ... Please use the data from Adam et al, *ChemPhysChem* 2022. The lower switching yield in that reference means a somewhat reduced triplet state shelving effect.

We have now acknowledged in Supplementary Table 3 that the value for $\Phi_{C \rightarrow T_1}$ is updated in the literature according to Adam et al. (*ChemPhysChem*, 2022). In our parameterization of the photoswitching fatigue of rsEGFP2, a reduced $\Phi_{C \rightarrow T_1}$ would result in an incomplete transfer of population to the off-state at the same 488 nm illumination times. Similarly, as the reviewer points out, a lower off-switching quantum yield increases the bleaching stemming from this channel, which given our current parametrization, would be represented as an overestimation of the simulated photobleaching compared to the experimental data. To recreate the experimental photoswitching curves, the bleaching quantum yield from the T_1 and T_n would be reduced in the fitting, nonetheless, preserving the magnitude of the photoswitching fatigue recovery induced by the 592 nm light. The study and characterization of photoswitching fatigue required multiparametric kinetic modelling, therefore, throughout this work we emphasis has been on reporting trends and behaviors which go beyond the precision of single parameter.

Supplementary note 3: to check the effect of T1 relaxation, I would recommend repeating the experiment with no 405, as a control.

We have repeated the experiment removing the 405 nm and also gradually increasing it. The data are now reported in the reply to the major point 3, since they represent a further validation of C^{int} being fluorescent.

Supplementary figures 6: the red crosses are not visible

We have now corrected the legend.

Supplementary note 4: “The addition of a photobleaching pathway was necessary to reproduce the 488 nm power density” -> 405 nm, not 488.

Corrected.

Supplementary note 5: the very long fitted lifetime of a putative radical state (17 s) is surprising to me. I wonder if this can result from the incomplete/nonideal photophysical scheme. In general radical states in fluorescent proteins are relatively short-lived under strong illumination, typically tens of milliseconds. Ref 25 does not report a super long lifetime radical species. Furthermore, the off switching of rsEGFP2 is always seen biexponential, which is assigned to transient excursions to a short-lived dark state (in comparison to the lifetime of the off-switched state). Finally, radical states are typically light-sensitive, so their lifetime can be shortened a lot under illumination. Please discuss the obtained results in view of this literature knowledge.

The incorporation of a long-lived dark state was necessary to reproduce the data of Figure 1g and the reversible photobleaching observed therein. In our modelling of the observed relaxation, we borrow the photophysical scheme proposed for IrisFP (Duan et al., 2013, *J. Am. Chem. Soc.* and Roy et al., 2011, *J. Am. Chem. Soc.*) where a radical state that is formed from the triplet state does not interplay with the illumination light. The recovery of photobleaching with increasing dark times per cycle observed in our data is consistent with other accounts of reversible photobleaching with a time scale of seconds (Sinnecker et al., 2005, *Biochemistry* and Berardozzi, 2016, *J. Am. Chem. Soc.*). We agree with the reviewer’s view that radical states are typically described as light-sensitive, for example, in Berardozzi et al., 2016, *J. Am. Chem. Soc.*, the authors show how the reversible photobleaching can be speed up with 405 nm illumination, nonetheless, the energy doses employed in that study are much higher than in our experiments (3 W/cm^2 for $\sim 20\text{-}30\text{ s}$, while we use 240 W/cm^2 for 10^{-3} s). Similarly, the biexponential behavior of rsEGFP2 off-switching has been shown in illumination conditions very different to our experiments (0.27 W/cm^2 488 nm light for 85 s in Adam et al., 2022, *ChemPhysChem*), where the off-switching curves can be well reproduced with a simple 4-state model. It is true that if the putative radical state is accessible to 405 nm light, its lifetime would be shortened, however, establishing the spectroscopic characteristics of that state is not possible with our data. We now acknowledge in Supplementary Note 5 that in view of the literature knowledge on long-lived dark state of green FPs our description is a simplification of the pathways behind the reversible photobleaching recovery we observe.

Supplementary table 6: why is the amplitude for the first phase negative ? If two phases are seen, could this be a sign of the relaxing radical with a rate somewhat slower than triplet state relaxation ? But again, here relaxation is in the dark, and radicals would relax fast maybe only under light.

We thank the reviewer for pointing out this inconsistency. There is indeed no reason why two exponentials should be considered to model the triplet lifetime, therefore we repeated the fitting and now reported the results for a mono-exponential fit.

Supplementary note 7: the strongly varying photobleaching yields fitted in different experiments are a bit frightening. Although FPs are definitely sensitive to their environment, a change by a factor of 10 in different (environment) replicates sounds really odd. Please comment.

We acknowledge that the absolute bleaching yields vary strongly across experiments, and we agree that this can appear striking. This variability is precisely why we do not report absolute bleaching rates but instead focus on relative dependencies that we could consistently reproduce across multiple days of measurements. We identified several experimental factors that influence absolute bleaching behavior, including the age of the PAA gel, oxygen content, and room temperature. While we do not yet have a complete rationale for the full extent of this variability, our conclusions are based on reproducible trends in the bleaching behavior rather than on absolute bleaching yields.

**Point-by-point responses to reviewers of the manuscript NCOMMS-25-05441B:
All-Optical Strategies to Minimize Photo-Bleaching in Reversibly Switchable Fluorescent
Proteins**

Marin-Aguilera et al

We thank both reviewers for their critical feedback throughout all the process of revision. Every input has brought the paper to mature in both experimental investigation and reader accessibility.

In the following section, we address the last two comments from reviewer #2 with our responses in black and the reviewer comments in blue.

Reviewer #2

The authors have performed another round of extensive revisions, answering positively all my concerns. I congratulate the authors for a very thorough and interesting work. I have two minor suggestions, but I let the authors and the editor decide about their relevance.

1/ The first part of the results will probably be hard for the readers to immediately digest. Perhaps the authors could simplify the flow, and move some parts to the discussion?

After careful consideration, we have decided to keep the structure as it is laid out. We believe that the present structure reflects the logic of the paper. We want to follow a bottom-up approach, with all the building block of the photophysical behavior investigated separately and only later discussed in the context of the imaging application.

2/ The data now convincingly suggest that the intermediate C_{int} is indeed fluorescent. In fact, this is interesting: it could be that C_{int} is very close to the on-state, but with a different pK_a due to a modification in the chromophore environment, e.g. due to hydrogen bonding changes. A very similar finding was published recently, albeit on a photoconvertible anthozoan FP (J. Am. Chem. Soc. 2025, 147, 12, 10357–10368). The authors might want to point to such an hypothesis.

We agree with reviewer #2 that the fluorescent nature of C_{int} is an interesting property and possibly instrumental for microscopy applications. We are also aware that to uniquely identify its nature further spectroscopical analysis will be crucial. Specifically to the suggested hypothesis, we are unsure how the proposed pH equilibrium of the two ON species will play out in the μ s timescale relaxation of our measurements and in the typical model arising from ultrafast experiments (such as Woodhouse et al 2020, *Nat Comm*), where the relaxation cascade is sequential process and the proton transfer is one of the steps in the tens-hundreds μ s.

For this reason, we have decided to limit ourselves to report only the objective observed behaviour.